# FINITE-TIME ANALYSIS OF ACTOR-CRITIC METHODS WITH DEEP NEURAL NETWORK APPROXIMATION

**Xuyang Chen, Fengzhuo Zhang, Keyu Yan, Lin Zhao** [*]
Department of Electrical and Computer Engineering
National University of Singapore
`{xuyang.chen,yky,fzzhang,elezhli}@nus.edu.sg`

## ABSTRACT

Actor–critic (AC) algorithms underpin many of today's most successful reinforcement learning (RL) applications, yet their finite-time convergence in realistic settings remains largely underexplored. Existing analyses often rely on oversimplified formulations and are largely confined to linear function approximation. In practice, however, nonlinear approximations with deep neural networks dominate AC implementations, leaving a substantial gap between theory and practice. In this work, we provide the first finite-time analysis of single-timescale AC with deep neural network approximation in continuous state-action spaces. In particular, we consider the challenging time-average reward setting, where one needs to simultaneously control three highly-coupled error terms including the reward error, the critic error, and the actor error. Our novel analysis is able to establish convergence to a stationary point at a rate $\widetilde{\mathcal{O}}(T^{-1/2})$, where $T$ denotes the total number of iterations, thereby providing theoretical grounding for widely used deep AC methods. We substantiate these theoretical guarantees with experiments that confirm the proven convergence rate and further demonstrate strong performance on MuJoCo benchmarks.

## 1 INTRODUCTION

Actor-critic (AC) methods have achieved substantial success in many challenging applications (Silver et al., 2017; Vinyals et al., 2019; Lazaridis et al., 2020). In particular, it becomes instrumental in enabling highly robust and agile robot motion control involving continuous state-action spaces, such as quadruped locomotion control (Miki et al., 2022; Hoeller et al., 2024), humanoid whole-body control (Radosavovic et al., 2024), drone racing (Kaufmann et al., 2023), etc. These successes are largely driven by the use of powerful function approximators, such as deep neural networks, to represent control policies (actors) and value functions (critics).

Despite substantial empirical success, the theoretical understanding of AC methods remains under-developed, especially in the most practical settings. Existing studies often restrict attention to finite state–action spaces and adopt simplified algorithmic variants to ease analysis. For instance, *double-loop* methods perform multiple critic updates per fixed actor (Yang et al., 2019; Kumar et al., 2023; Agarwal et al., 2021; Xu et al., 2020b), which improves value estimation and thereby yields a more accurate policy gradient for that actor. This enables a clean, decoupled analysis of the actor and critic, but at the cost of impractically high sampling complexity. Similarly, *two-timescale* methods (Wu et al., 2020; Xu et al., 2020c; Chen et al., 2023) impose a smaller step size on the actor than the critic, with their ratio vanishing as iterations grow (i.e., $\lim_{t\to\infty} \alpha_t/\beta_t = 0$). This asymptotically decouples the actor and critic, mimicking multiple critic updates per actor. However, this artificial slowing down of the actor is undesirable and rarely adopted in practice.

In contrast, the canonical form widely used in practice is the *single-timescale* AC algorithm, where both actor and critic are updated simultaneously with proportional step sizes at each iteration (i.e., $\alpha_t/\beta_t = c$). However, analyzing its convergence is more challenging than for the aforementioned simplified variants, as the actor and critic updates are strongly coupled. The aforementioned decoupled analysis is over-conservative and cannot establish convergence of the single-timescale AC.

---

[*]Corresponding author

Table 1: Comparison of related works on single-timescale actor-critic algorithm analysis.

| Reference | MDP | | Sampling | | Approximation | Convergence | |
|---|---|---|---|---|---|---|---|
| | Continuous State Space | Continuous Action Space | Markovian for Actor | Markovian for Critic | Neural Network Function Class | Experiment Validation | Convergence Rate |
| Chen et al. (2021) | ✓ | ✗ | ✗ | ✗ | ✗ | ✗ | $\mathcal{O}(T^{-0.5})$ |
| Olshevsky & Gharesifard (2023) | ✗ | ✗ | ✗ | ✗ | ✗ | ✗ | $\mathcal{O}(T^{-0.5})$ |
| Chen & Zhao (2024) | ✓ | ✗ | ✓ | ✓ | ✗ | ✗ | $\tilde{\mathcal{O}}(T^{-0.5})$ |
| Tian et al. (2024) | ✗ | ✗ | ✗ | ✓ | ✓ | ✗ | $\tilde{\mathcal{O}}(T^{-0.5} + m^{-0.5})$ |
| Ours | ✓ | ✓ | ✓ | ✓ | ✓ | ✓ | $\tilde{\mathcal{O}}(T^{-0.5})$ |

Recent efforts to study the convergence of the single-timescale AC algorithm include Chen et al. (2021), Olshevsky & Gharesifard (2023), Chen & Zhao (2024), and Tian et al. (2024), with their respective settings summarized in Table 1. Among these works, only Tian et al. (2024) considers single-timescale AC with neural network approximation. Nevertheless, it suffers from two fundamental limitations. First, it is restricted to finite state–action spaces, where linear function approximation already suffices. This renders the neural network perspective redundant and undermines the practical significance of the analysis. In contrast, real-world reinforcement learning problems typically involve continuous state–action spaces and rely on neural networks for expressive function approximation. Second, as shown in Table 1, the convergence rate in Tian et al. (2024) is $\tilde{\mathcal{O}}(T^{-0.5} + m^{-0.5})$, where $T$ denotes the number of iterations and $m$ the neural network width. While the $T$-dependence is natural for finite-time analysis, the $m$-dependence is problematic. In practice, $m$ is fixed during training and does not scale with $T$, leaving a constant $\tilde{\mathcal{O}}(m^{-0.5})$ error term that fails to capture the true convergence behavior. Moreover, neural tangent kernel theory (Jacot et al., 2018) shows that neural networks become increasingly linear as $m \to \infty$, thereby degrading their representational power. Intuitively, the convergence of the algorithm should not hinge on such $m$-limiting behavior. The observed dependence on $m$ in prior results is merely a consequence of analytical technicalities, rather than a fundamental property of the algorithm.

Motivated by these gaps, we provide the first finite-time convergence guarantee for single-timescale AC in continuous state–action spaces under the time-average reward setting. Our analysis rigorously incorporates both deep neural network approximation and Markovian sampling for the actor and the critic. We prove the convergence of the reward error (Eq. (8a)), critic error (Eq. (8b)), and actor error (Eq. (8c)) at rate $\tilde{\mathcal{O}}(T^{-1/2})$, without requiring the network width $m$ to diverge. As summarized in Table 1, our results compare favorably with prior studies across key dimensions that are critical for practical applicability.

From a technical perspective, this improvement is enabled by a series of technical innovations. To sharpen the convergence rate, we show that the smoothness-induced error—arising uniquely from neural networks and absent in the linear setting—is intertwined with the critic error. Unlike Tian et al. (2024), which conservatively bound the critic error by a constant, we prove that its mean path diminishes, thereby removing the prior requirement of $m \to \infty$ (see the mean-path update analysis in Eq. (26)). To address the challenges of continuous state–action spaces, we introduce an operator-based framework (see Eq. (1)) capable of handling uncountable domains. To mitigate error propagation caused by deep neural networks (DNNs) approximation of the value function, we establish a set of important regularity properties of DNNs in Lemma 1. Moreover, the interplay between DNN approximation error and Markovian sampling noise poses greater challenges than those encountered in the linear function approximation (Chen & Zhao, 2024) or the i.i.d. sampling setting (Olshevsky & Gharesifard, 2023; Tian et al., 2024). To control such complex error dynamics, we develop refined analyses in Lemma 7–Lemma 10 (Markovian noises). A high-level overview of our proof ideas and newly developed techniques is provided in the Proof Sketch in Appendix C.

This work is further distinguished by extensive empirical validation that corroborates our theoretical results (see Section 5). Although all prior studies listed in Table 1 attempt to bridge the gap between theory and practice, none included empirical evaluations. A key reason is that many of their assumptions are unrealistic for practical deployment. For instance, the algorithms analyzed in prior works cannot even be applied to simple tasks such as `gymnasium Pendulum-v1`(Towers et al., 2024), since they are restricted to finite action spaces and depend on sampling from the stationary state distribution. Moreover, AC with linear function approximation is generally incapable of controlling

standard benchmarks. In particular, our experiments show that the linear critic fails to approximate the value function even for the simple pendulum (Figure 1). These observations highlight the substantial gap between the simplifying assumptions underlying existing theoretical analyses and the complexities inherent in practical RL applications. In contrast, we empirically verify both the convergence and the proven convergence rate of our algorithm on pendulum, and further demonstrate its effectiveness on more challenging Gym MuJoCo benchmarks, where neural AC consistently outperforms its linear approximation counterpart. This underscores the importance of analyzing neural AC algorithms, which are both practically relevant and theoretically more challenging.

**Notation.** We use san-serif letters to denote scalars and use lower and upper case bold letters to denote vectors and matrices respectively. For two sequences of real numbers $(x_n)$ and $(y_n)$, we write $x_n = O(y_n)$ if there exists $C < \infty$ such that $|x_n| \leq C|y_n|$ for all $n$ sufficiently large. We use $\tilde{\mathcal{O}}(\cdot)$ to further hide logarithmic factors. The total variation distance of two probability measure $\mu$ and $\nu$ is defined by $d_{TV}(\mu, \nu) := 1/2 \int_{\mathcal{X}} |\mu(dx) - \nu(dx)|$.

## 2 PRELIMINARIES

**Markov Decision Process.** We consider the standard Markov Decision Process (MDP) characterized by $(\mathcal{S}, \mathcal{A}, \mathcal{P}, r)$, where $\mathcal{S}$ is the state space and $\mathcal{A}$ is the action space. The spaces $\mathcal{S}$ and $\mathcal{A}$ are allowed to be either finite sets or real vector spaces, i.e., $\mathcal{S} \subset \mathbb{R}^{d_s}$ and $\mathcal{A} \subset \mathbb{R}^{d_a}$. The transition kernel is denoted by $\mathcal{P}(s_{t+1}|s_t, a_t) \in \mathbb{R}_{\geq 0}$ and the reward function is $r : \mathcal{S} \times \mathcal{A} \to [-u_r, u_r]$. A policy $\pi_{\boldsymbol{\theta}}$ parameterized by $\boldsymbol{\theta} \in \mathcal{X}_{\Theta}$ maps a given state to a probability distribution over the action space, i.e., $\boldsymbol{a}_t \sim \pi_{\boldsymbol{\theta}}(\cdot|\boldsymbol{s}_t)$. In this work, we consider the time-average reward setting (Sutton et al., 1999; Yang et al., 2019; Wu et al., 2020; Chen & Zhao, 2024), which aims to find a policy $\pi_{\boldsymbol{\theta}}$ that maximizes the following infinite-horizon time-average reward:

$$J(\boldsymbol{\theta}) := \lim_{T \to \infty} \mathbb{E}_{\boldsymbol{\theta}}\left[\frac{1}{T}\sum_{t=0}^{T-1} r(s_t, a_t)\right] = \mathbb{E}_{(s,a) \sim (\mu_{\boldsymbol{\theta}}, \pi_{\boldsymbol{\theta}})}\big[r(s, a)\big].$$

In the above equation, the expectation $\mathbb{E}_{\boldsymbol{\theta}}$ is taken over the states and actions generated by following the policy $\pi_{\boldsymbol{\theta}}$ and the transition kernel $\mathcal{P}$. Additionally, $\mu_{\boldsymbol{\theta}}$ denotes the stationary state distribution induced by $\pi_{\boldsymbol{\theta}}$ and $\mathcal{P}$. The existence of this stationary distribution is guaranteed by the uniform ergodicity of the underlying MDP, which is a common assumption (See Assumption 6 in the sequel). Hereafter, we refer to $J(\boldsymbol{\theta})$ as the time-average reward (and exchangeably, *performance function*), which can be evaluated by the expected reward over the stationary distribution $\mu_{\boldsymbol{\theta}}$ and the policy $\pi_{\boldsymbol{\theta}}$. The state-value function is used to evaluate the overall rewards starting from a state $s$, following policy $\pi_{\boldsymbol{\theta}}$ and transition kernel $\mathcal{P}$ thereafter, which is defined as

$$V_{\boldsymbol{\theta}}(s) := \mathbb{E}_{\boldsymbol{\theta}}\left[\sum_{t=0}^{\infty}\big(r(s_t, a_t) - J(\boldsymbol{\theta})\big)\bigg|s_0 = s\right].$$

Similarly, we define the action-value (Q-value) function to evaluate the overall rewards starting from $s$, taking action $a$, and following transition kernel $\mathcal{P}$ and policy $\pi_{\boldsymbol{\theta}}$ thereafter:

$$Q_{\boldsymbol{\theta}}(s, a) := \mathbb{E}_{\boldsymbol{\theta}}\left[\sum_{t=0}^{\infty}\big(r(s_t, a_t) - J(\boldsymbol{\theta})\big)\bigg|s_0 = s, a_0 = a\right] = r(s, a) - J(\boldsymbol{\theta}) + \mathbb{E}\big[V_{\boldsymbol{\theta}}(s')\big],$$

where the last expectation is taken over $s' \sim \mathcal{P}(\cdot|s, a)$.

To tackle the technical challenges associated with neural network function approximation over continuous state and action spaces, we introduce two auxiliary operators. Let $\mathcal{F} := \{f \mid f : \mathcal{S} \to \mathbb{R}\}$ denote the class of real-valued functions on $\mathcal{S}$. For a policy $\pi_{\boldsymbol{\theta}}$, define the operators $D_{\boldsymbol{\theta}} : \mathcal{F} \to \mathcal{F}$ and $P_{\boldsymbol{\theta}} : \mathcal{F} \to \mathcal{F}$ as

$$(D_{\boldsymbol{\theta}}f)(s) := \mu_{\boldsymbol{\theta}}(s)\,f(s), \quad (P_{\boldsymbol{\theta}}f)(s) := \int_{\mathcal{S}}\int_{\mathcal{A}} f(s')\,\mathcal{P}(s' \mid s, a)\pi_{\boldsymbol{\theta}}(a \mid s)dads'. \tag{1}$$

Here, $D_{\boldsymbol{\theta}}$ multiplies a function $f$ by the stationary distribution $\mu_{\boldsymbol{\theta}}$, whereas $P_{\boldsymbol{\theta}}$ maps $f$ to its one-step look-ahead under the Markov chain induced by $\pi_{\boldsymbol{\theta}}$ and $\mathcal{P}$, i.e., $(P_{\boldsymbol{\theta}}f)(s) = \mathbb{E}_{\boldsymbol{\theta}}[f(s_{t+1}) \mid s_t = s]$. The inner product on $\mathcal{F}$ is given by

$$\langle f, g \rangle = \int_{\mathcal{S}} f(s)\,g(s)\,ds, \tag{2}$$

and the induced norm of a function $f$ is $\|f\|^2 = \langle f, f \rangle$.

**Actor-Critic.** In AC, the actor corresponds to the policy, while the critic typically estimates the actor's value function via temporal-difference learning. The actor then updates its policy parameters through stochastic gradient ascent to maximize the performance function. The policy gradient theorem (Sutton et al., 1999) offers a closed-form expression for the gradient of the performance function $J(\boldsymbol{\theta})$ with respect to the policy parameters $\boldsymbol{\theta}$, which is given by

$$\nabla_{\boldsymbol{\theta}} J(\boldsymbol{\theta}) = \mathbb{E}_{s \sim \mu_{\boldsymbol{\theta}}, a \sim \pi_{\boldsymbol{\theta}}} \big[ Q_{\boldsymbol{\theta}}(s, a) \cdot \nabla_{\boldsymbol{\theta}} \log \pi_{\boldsymbol{\theta}}(a|s) \big]. \tag{3}$$

Equivalently, the policy gradient can be written as

$$\nabla J(\boldsymbol{\theta}) = \mathbb{E}_{s \sim \mu_{\boldsymbol{\theta}}, a \sim \pi_{\boldsymbol{\theta}}} [(Q_{\boldsymbol{\theta}}(s, a) - b(s)) \nabla_{\boldsymbol{\theta}} \log \pi_{\boldsymbol{\theta}}(a|s)],$$

where $b(s)$ is called the baseline function, which is employed to reduce the variance of the gradient estimate. A popular choice of baseline is the state-value function, which leads to the following so-called advantage-based policy gradient

$$\nabla_{\boldsymbol{\theta}} J(\boldsymbol{\theta}) = \mathbb{E}_{s \sim \mu_{\boldsymbol{\theta}}, a \sim \pi_{\boldsymbol{\theta}}} [\Delta_{\boldsymbol{\theta}}(s, a) \nabla_{\boldsymbol{\theta}} \log \pi_{\boldsymbol{\theta}}(a|s)], \tag{4}$$

where $\Delta_{\boldsymbol{\theta}} := Q_{\boldsymbol{\theta}}(s, a) - V_{\boldsymbol{\theta}}(s)$ is known as the advantage function.

In deep RL, the policy (actor) and value functions (critic) are typically parameterized by deep neural networks due to their strong representation capabilities (Mnih et al., 2015; Lillicrap et al., 2015). However, the convergence of training deep neural networks are less understood, especially in RL. In this paper, we establish conditions and provide a finite-time analysis for single-timescale AC algorithms utilizing deep neural network approximations for both the actor and the critic.

## 3 THE SINGLE-TIMESCALE NEURAL ACTOR-CRITIC ALGORITHM

In this section, we present the single-timescale neural AC algorithm to be analyzed in the sequel, incorporating key components commonly found in practical implementations.

### 3.1 PARAMETERIZATION OF THE VALUE FUNCTION AND POLICY

We consider a deep neural network for estimating the true state-value function $V_{\boldsymbol{\theta}}(s)$ under a policy $\pi_{\boldsymbol{\theta}}$. The network $\widehat{V}(\boldsymbol{\omega}; \boldsymbol{s})$ has a general form of a deep neural network with a linear output layer:

$$\boldsymbol{s}^{(0)} = \boldsymbol{s}, \quad \boldsymbol{s}^{(k)} = \frac{1}{\sqrt{m_k}} \sigma(\boldsymbol{W}^{(k)} \boldsymbol{s}^{(k-1)}), k = 1, 2, \cdots, K, \quad \widehat{V}(\boldsymbol{\omega}; \boldsymbol{s}) = \frac{1}{\sqrt{m_K}} \boldsymbol{b}^\top \boldsymbol{s}^{(K)}, \tag{5}$$

where $K$ is the total number of hidden layers, state $\boldsymbol{s} \in \mathbb{R}^{d_s}$ is the input to the neural network, $\sigma$ is an element-wise activation function, $\boldsymbol{b}$ is a fixed coefficient vector for the output layer, and $\boldsymbol{\omega} \in \mathcal{X}_\Omega$ stands for the trainable parameter of the neural network. The latter is a column vector formed by stacking the weights of different layers, $\boldsymbol{\omega} := \{\boldsymbol{W}^{(k)} \in \mathbb{R}^{m_k \times m_{k-1}}\}_{k=1}^K$, where $m_k \in \mathbb{N}$ is the width of the $k$-th layer and $m_0 = d_s$ is the input dimension. Without loss of generality, we assume all the hidden layers have the same width $m$, i.e., $m_k = m$ for $k \in \{1, 2, \cdots, K\}$. It is for the ease of presentation only. As shown in the proof, our analysis also applies to $m_k \geq m$. Note that the above definition is general enough to encompass standard multilayer perceptrons (MLPs), convolutional neural networks (CNNs), and residual networks (ResNets) as special cases (Liu et al., 2020).

The policy $\pi_{\boldsymbol{\theta}}$ is allowed to have a general parameterization, including linear functions (Yang et al., 2019), deep neural networks (Wang et al., 2019), and energy-based policies (Fu et al., 2020). For the deep neural network approximation case, the actor can be parameterized similarly to Eq. (5), where all the trainable parameters will be stacked into the column vector $\boldsymbol{\theta} \in \mathcal{X}_\Theta$.

### 3.2 ALGORITHM DESIGN

In this subsection, we first aim to update the parameter of the neural network (the critic) $\boldsymbol{\omega}$ so that $\widehat{V}(\boldsymbol{\omega}; s)$ can approximate the true value function $V_{\boldsymbol{\theta}}(s)$ of a policy $\pi_{\boldsymbol{\theta}}$. Concretely, at step $t$,

---

**Algorithm 1** Single-Timescale Neural Actor-Critic

---

1: **Input** initial actor parameter $\boldsymbol{\theta}_0$, initial critic parameter $\boldsymbol{\omega}_0$, initial reward estimator $\eta_0$, stepsizes $\alpha$ for actor, $\beta$ for critic, and $\gamma$ for reward estimator.
2: Draw $s_0$ from some initial distribution
3: **for** $t = 0, 1, 2, \cdots, T - 1$ **do**
4:     Take action $a_t \sim \pi_{\boldsymbol{\theta}_t}(\cdot \,|\, s_t)$
5:     Observe next state $s_{t+1} \sim \mathcal{P}(\cdot \,|\, s_t, a_t)$ and reward $r_t = r(s_t, a_t)$
6:     $\delta_t = r_t - \eta_t + \widehat{V}(\boldsymbol{\omega}_t; \boldsymbol{s}_{t+1}) - \widehat{V}(\boldsymbol{\omega}_t; \boldsymbol{s}_t)$
7:     $\eta_{t+1} = \eta_t + \gamma(r_t - \eta_t)$
8:     $\boldsymbol{\omega}_{t+1} = proj_{\mathcal{B}_{\boldsymbol{\omega}_0}}(\boldsymbol{\omega}_t + \beta\delta_t\nabla\widehat{V}(\boldsymbol{\omega}_t; \boldsymbol{s}_t))$
9:     $\boldsymbol{\theta}_{t+1} = \boldsymbol{\theta}_t + \alpha\delta_t\nabla_{\boldsymbol{\theta}} \log \pi_{\boldsymbol{\theta}_t}(a_t \,|\, s_t)$
10: **end for**

---

we implement Stochastic Gradient Descent (SGD) methods to adjust the critic in the direction that would most reduce the mean square value error $[V(s_t) - \widehat{V}(\boldsymbol{\omega}_t; s_t)]^2$:

$$
\begin{aligned}
\boldsymbol{\omega}_{t+1} =& \boldsymbol{\omega}_t - \frac{1}{2}\beta\nabla\big[V(s_t) - \widehat{V}(\boldsymbol{\omega}_t; s_t)\big]^2 \\
=& \boldsymbol{\omega}_t + \beta\big[V(s_t) - \widehat{V}(\boldsymbol{\omega}_t; s_t)\big]\nabla\widehat{V}(\boldsymbol{\omega}_t; s_t),
\end{aligned}
\tag{6}
$$

where $\beta$ is the stepsize (learning rate). Since $V(s_t)$ is unknown, the semi-gradient TD(0) method approximates it by replacing $V(s_t)$ with the current target $r_t - J(\boldsymbol{\theta}) + \widehat{V}(\boldsymbol{\omega}_t; s_{t+1})$. To further estimate the unknown time-average reward $J(\boldsymbol{\theta})$, we use the following exponential moving average update of $\eta_t$,

$$
\eta_{t+1} = \eta_t + \gamma(r_t - \eta_t),
$$

where $\gamma$ is the stepsize. Hereafter, we will refer to it as the *reward estimator*. This additional estimation of the time-average reward $J(\boldsymbol{\theta})$ introduces more analysis complexity compared to the discounted setting (Olshevsky & Gharesifard, 2023; Tian et al., 2024). Now, by denoting the TD error as

$$
\delta_t := r_t - \eta_t + \widehat{V}(\boldsymbol{\omega}_t; \boldsymbol{s}_{t+1}) - \widehat{V}(\boldsymbol{\omega}_t; \boldsymbol{s}_t),
$$

we can rewrite the update of the critic in Eq. (6) as

$$
\boldsymbol{\omega}_{t+1} = \boldsymbol{\omega}_t + \beta\delta_t\nabla\widehat{V}(\boldsymbol{\omega}; \boldsymbol{s}_t).
$$

For the neural network specified in Section 3.1, we require its width $m$ to be a large constant such that the neural network is in the overparameterization regime. In this regime, the optimal solution typically resides in the neighborhood of the initialization (Du et al., 2019; Chen et al., 2021; Tian et al., 2024). Therefore, in Line 8 of Algorithm 1, we constrain the update of the critic parameter within a ball of constant radius around its initial condition, which ensures the boundedness without overlooking the optimal solution. Specifically, $proj_{\mathcal{B}_{\boldsymbol{\omega}_0}}$ stands for the projection onto a ball with a constant radius around the initial condition of the critic, i.e., $\mathcal{B}_{\boldsymbol{\omega}_0} = \{\boldsymbol{\omega}\|\|\boldsymbol{\omega} - \boldsymbol{\omega}_0\| \leq u_{\boldsymbol{\omega}}\}$, where $u_{\boldsymbol{\omega}}$ is a constant.

For the actor update, it is standard to use the TD error ($\delta_t$) as an approximation of the advantage function (Sutton & Barto, 2018). Therefore, based on the policy gradient theorem, the corresponding update rule for the actor can be written as

$$
\boldsymbol{\theta}_{t+1} = \boldsymbol{\theta}_t + \alpha\delta_t\nabla_{\boldsymbol{\theta}} \log \pi_{\boldsymbol{\theta}_t}(a_t|s_t),
$$

where $\delta_t\nabla_{\boldsymbol{\theta}} \log \pi_{\boldsymbol{\theta}_t}(a_t|s_t)$ is an approximation of the policy gradient defined in Eq. (4). The parallel updates of the critic and actor in Lines 8 and 9 aim to drive the actor towards the direction that increases the time-average reward $J(\boldsymbol{\theta})$.

We summarize the above-described AC algorithm in Algorithm 1, which follows the classic AC architecture studied in prior works under various settings, as listed in Table 1. The "single-timescale" refers to the fact that the stepsizes $\alpha, \beta, \gamma$ are only constantly proportional to each other. We consider

the more challenging neural network approximation for both the actor and the critic, which is referred to as the "neural actor-critic". Moreover, we consider the more practical Markovian sampling, starting from an initial state $s_0$, with subsequent states and actions generated according to the transition kernel and the policy, respectively. The consecutive transition tuples $(s_0, a_0, s_1, a_1, s_2, \cdots)$ form a single trajectory, thereby circumventing the time-consuming re-sampling procedure (i.i.d. sampling) mandated in prior works (Chen et al., 2021; Tian et al., 2024). More importantly, we aim to address the challenging settings of continuous state and action spaces that are prevalent in applications (see Table 1 for a detailed comparison). The finite-time convergence in such contexts is of significant interest to the community but remains unresolved.

## 4 ANALYSIS OF SINGLE-TIMESCALE NEURAL ACTOR-CRITIC

In this section, we begin by outlining several standard assumptions and then present our main finite-time convergence results for the algorithm.

### 4.1 ASSUMPTIONS

**Assumption 1** (Neural architecture and initialization). *The neural network defined in Eq. (5) satisfies the following properties:*

*(a) (Input assumption) Any input to the neural network satisfies $\|\boldsymbol{s}^{(0)}\| \leq 1$.*

*(b) (Activation assumption) $\sigma$ is $l_\sigma$-Lipschitz and $h_\sigma$-smooth. i.e., $\forall x_1, x_2 \in \mathbb{R}$, (i) $|\sigma(x_1) - \sigma(x_2)| \leq l_\sigma|x_1 - x_2|$; (ii) $|\sigma'(x_1) - \sigma'(x_2)| \leq h_\sigma|x_1 - x_2|$ where $\sigma'$ is the derivative of $\sigma$.*

*(c) (Initialization assumption) Each entry of the vector $\boldsymbol{b}$ satisfies $|b_i| \leq 1, \forall i$, and the weights of the neural network $\boldsymbol{W}_0^{(k)}$ are randomly initialized from a normal distribution $\mathcal{N}(0, 1)$, with each entry being independently sampled.*

This assumption mainly states the initialization and analytic properties of the neural network. We note that these assumptions are widely satisfied in various applications. For the input norm constraint, we could normalize the state space to guarantee this assumption. Regarding the activation function, we emphasize that many commonly used activation functions, such as sigmoid and GeLu, satisfy this condition. The initialization assumption, furthermore, can be easily implemented during neural network training. We also note that the above assumptions are common in the theoretical analysis of neural networks (Liu et al., 2020; Tian et al., 2024).

As shown in Lemma F.4 of (Liu et al., 2020), with Assumption 1, the following assumption holds with high probability, which we state as an assumption in our work for ease of presentation.

**Assumption 2.** *The absolute value of each entry of $\boldsymbol{s}^{(k)}$ (the output of layer $k$ of the neural network) is $\widetilde{O}(1)$ at initialization. The initial weights satisfy $\|\boldsymbol{W}_0^{(k)}\| \leq \mathcal{O}(\sqrt{m})$ for all $k$.*

For the value function $V_{\boldsymbol{\theta}}(s)$ of a given policy $\boldsymbol{\theta}$, its best approximation using the neural network (Eq. (5)) is defined via

$$\epsilon_{\text{app}}(\boldsymbol{\omega}^*(\boldsymbol{\theta})) := \inf_{\boldsymbol{\omega}} \sqrt{\mathbb{E}_{s \sim \mu_{\boldsymbol{\theta}}}\left[(\widehat{V}(\boldsymbol{\omega}; s) - V_{\boldsymbol{\theta}}(s))^2\right]}, \tag{7}$$

where $\boldsymbol{\omega}^*(\boldsymbol{\theta})$ is referred to as the *optimal critic* that yields the minimal (optimal) approximation error $\epsilon_{\text{app}}(\boldsymbol{\omega}^*(\boldsymbol{\theta}))$. In this paper, we assume the optimal approximation errors for all potential policies are uniformly bounded, that is,

$$\forall \boldsymbol{\theta}, \ \epsilon_{\text{app}}(\boldsymbol{\omega}^*(\boldsymbol{\theta})) \leq \epsilon_{\text{app}},$$

for some constant $\epsilon_{\text{app}} \geq 0$. The error $\epsilon_{\text{app}}$ is zero if $V_{\boldsymbol{\theta}}$ can be exactly approximated by the neural network (Eq. (5)). Naturally, it is expected that the learning errors of Algorithm 1 depend on $\epsilon_{\text{app}}$, which represents the approximation capacity of the critic.

The assumption of a uniformly bounded approximation error is common in the literature (Chen et al., 2021; Zhang et al., 2023b; Olshevsky & Gharesifard, 2023; Chen & Zhao, 2024; Tian et al., 2024). It is more restrictive for the linear function approximation than for the neural network setting. If

the true value function is not linear, which is typically the case in practice, the approximation error $\epsilon_{\text{app}}$ can be significantly large. In contrast, the neural network approximation can arbitrarily closely approximate any continuous function according to the universal Approximation Theorem (Hornik, 1991), and therefore can potentially keep the approximation error arbitrarily small.

We then make the following assumption for the optimal critic.

**Assumption 3** (Smoothness of optimal critic). *For any* $\boldsymbol{\theta}_1, \boldsymbol{\theta}_2 \in \mathcal{X}_\Theta$*, we have*

$$\|\boldsymbol{\omega}^*(\boldsymbol{\theta}_1) - \boldsymbol{\omega}^*(\boldsymbol{\theta}_2)\| \leq l_\omega \|\boldsymbol{\theta}_1 - \boldsymbol{\theta}_2\|,$$
$$\|\nabla \boldsymbol{\omega}^*(\boldsymbol{\theta}_1) - \nabla \boldsymbol{\omega}^*(\boldsymbol{\theta}_2)\| \leq h_\omega \|\boldsymbol{\theta}_1 - \boldsymbol{\theta}_2\|,$$

*where* $l_\omega$ *and* $h_\omega$ *are finite positive constants.*

The above assumption states that the optimal critic is $l_\omega$-Lipschitz and $h_\omega$-smooth. This assumption is commonly employed for the single-timescale AC with neural network approximation (Tian et al., 2024). In the case of linear function approximation, the above assumption is trivially implied by the linearity of the value function (Olshevsky & Gharesifard, 2023; Chen & Zhao, 2024).

Furthermore, we specify the regularity of the neural network.

**Assumption 4** (Regularity of the neural network). *For the neural network defined in Eq. (5), there exists some constant* $\lambda_1 > 0$ *such that*

$$\|\widehat{V}(\boldsymbol{\omega}) - \widehat{V}(\boldsymbol{\omega}^*(\boldsymbol{\theta}))\| \geq \lambda_1 \|\boldsymbol{\omega} - \boldsymbol{\omega}^*(\boldsymbol{\theta})\|, \; \forall \boldsymbol{\theta} \in \mathcal{X}_\Theta, \boldsymbol{\omega} \in \mathcal{X}_\Omega,$$

where the norm of a function is defined based on the inner product given in Eq. (2), which involves the product of function values integrated over $s$. Assumption 4 states the regularity of the neural network in terms of learning the optimal value. Intuitively, it requires that the perturbation of the critic parameter around the optimal one will cause a non-zero change of the critic neural network output. From the point of view of the optimization landscape of the neural network, it merely assumes that optimal and suboptimal points are distinguished. This is also a standard assumption of other analysis of AC methods with neural network approximation (Tian et al., 2024).

The next assumption pertains to the exploration of the policy $\pi_{\boldsymbol{\theta}}$ in continuous settings.

**Assumption 5** (Exploration). *There exists a constant* $\lambda_2 > 0$ *such that* $\langle \widehat{V}(\boldsymbol{\omega}), D_{\boldsymbol{\theta}}(I - P_{\boldsymbol{\theta}})\widehat{V}(\boldsymbol{\omega}) \rangle \geq \lambda_2 \|\widehat{V}(\boldsymbol{\omega})\|^2$*, for any* $\boldsymbol{\theta} \in \mathcal{X}_\Theta$ *and neural network* $\widehat{V}(\boldsymbol{\omega}) \in \mathcal{F}$*, where* $D_{\boldsymbol{\theta}}$*,* $P_{\boldsymbol{\theta}}$ *are operators defined in Eq. (1), I denotes the identity operator, and the inner product is defined in Eq. (2).*

To demonstrate its connection to exploration, we show that if exploration is insufficient, the assumption fails to hold. First note that the operator $D_{\boldsymbol{\theta}}$ essentially multiplies the stationary distribution $\mu_{\boldsymbol{\theta}}$ to the function on its left (see the definition in Eq. (1)). If the policy $\pi_{\boldsymbol{\theta}}$ does not sufficiently explore, there exists a subset of the state space $A \subset \mathcal{S}$ such that $\mu_{\boldsymbol{\theta}}(A) = 0$. Furthermore, we can choose $\widehat{V}(\boldsymbol{\omega})$ such that $\widehat{V}(\boldsymbol{\omega}; s) = 0, \forall s \in \mathcal{S} \setminus A$ and $\widehat{V}(\boldsymbol{\omega}; s) > 0, \forall s \in A$. With this choice, the left-hand side of the inequality evaluates to 0, while the right-hand side becomes positive. This violates the condition stated in Assumption 5. Thus, the contrapositive holds: if Assumption 5 is satisfied, it ensures sufficient exploration of the state space under the policy $\pi_{\boldsymbol{\theta}}$. This sufficient exploration assumption is standard in the literature of analyzing the convergence of AC algorithms (Wu et al., 2020; Chen et al., 2021; Chen & Zhao, 2024; Tian et al., 2024).

**Assumption 6** (uniform ergodicity). *For a Markov chain generated by the policy* $\pi_{\boldsymbol{\theta}}$ *and transition kernel* $\mathcal{P}$*, let* $\mathbb{P}$ *denote the corresponding state transition probability. Then there exists* $\kappa > 0$ *and* $\rho \in (0, 1)$ *such that the total variation distance between the state distribution at time* $\tau$ *and the stationary distribution* $\mu_{\boldsymbol{\theta}}$ *satisfies:* $d_{TV}(\mathbb{P}(s_\tau \in \cdot | s_0 = s), \mu_{\boldsymbol{\theta}}(\cdot)) \leq \kappa \rho^\tau$*, for all* $\tau \geq 0$*,* $s \in \mathcal{S}$*.*

Assumption 6 assumes the Markov chain is geometrically mixing, which is implied by the uniform ergodicity of the chain. It is commonly employed to characterize the noise induced by Markovian sampling in RL algorithms (Bhandari et al., 2018; Zou et al., 2019; Wu et al., 2020; Chen et al., 2021; Zhang et al., 2022; Olshevsky & Gharesifard, 2023; Chen & Zhao, 2025).

**Assumption 7** (Regularity of the policy). *Let* $\pi_{\boldsymbol{\theta}}(a|s)$ *be a bounded policy parameterized by* $\boldsymbol{\theta} \in \mathcal{X}_\Theta$*. There exists positive constants* $u_\pi, h_\pi$ *and* $l_p$ *such that for any* $\boldsymbol{\theta}$*, s, and a, it holds that: (i)* $\|\nabla \log \pi_{\boldsymbol{\theta}}(a|s)\| \leq u_\pi$*; (ii)* $\|\nabla \log \pi_{\boldsymbol{\theta}_1}(a|s) - \nabla \log \pi_{\boldsymbol{\theta}_2}(a|s)\| \leq h_\pi \|\boldsymbol{\theta}_1 - \boldsymbol{\theta}_2\|$*; (iii)* $|\pi_{\boldsymbol{\theta}_1}(a|s) - \pi_{\boldsymbol{\theta}_2}(a|s)| \leq l_p \|\boldsymbol{\theta}_1 - \boldsymbol{\theta}_2\|$*.*

Assumption 7 states the regularity of the policy, which is standard in the literature of actor-critic methods (Wu et al., 2020; Chen et al., 2021; Chen & Zhao, 2024; Tian et al., 2024). These conditions are sufficiently general to be satisfied by a wide range of distributions, including the uniform distribution, the truncated Gaussian distribution, and the Beta distribution with $\alpha, \beta > 1$.

## 4.2 FINITE-TIME ANALYSIS

We define the integer $\tau_T := \min\{i \geq 0 \mid \kappa\rho^{i-1} \leq T^{-1/2}\}$, where $T$ is the total number of iterations, $\kappa$ and $\rho$ are the same constants defined in Assumption 6. The integer $\tau_T$ represents a certain mixing time of an ergodic Markov chain, which will be used to control the Markovian noise in the analysis. In our main results, we require that $T \geq 2\tau_T$ to ensure that the Markov chain is well-mixed and the Markovian noise is effectively bounded. We can estimate that $\tau_T = \frac{\log \kappa\rho^{-1}}{\log \rho^{-1}} + \frac{\log T}{2 \log \rho^{-1}} = \mathcal{O}(\log T)$ which results in $\kappa\rho^{\tau_T - 1} \leq \frac{1}{\sqrt{T}}$.

We quantify the *learning errors* by defining $y_t := \eta_t - J(\boldsymbol{\theta}_t)$, which is the difference between the reward estimator and the true time-average reward $J(\boldsymbol{\theta}_t)$ at time $t$. For the critic, we define $\boldsymbol{z}_t := \boldsymbol{\omega}_t - \boldsymbol{\omega}_t^*$ with $\boldsymbol{\omega}_t^* := \boldsymbol{\omega}^*(\boldsymbol{\theta}_t)$ to measure the error between the critic and its target value at iteration $t$. The following theorem summarizes our main results.

**Theorem 1.** *Consider Algorithm 1 with $\alpha = \frac{c}{\sqrt{T}}, \beta = \frac{1}{\sqrt{T}}, \gamma = \frac{1}{\sqrt{T}}$, where $c$ is a constant depending on problem parameters. Suppose Assumption 1-7 hold, for $T \geq 2\tau_T$, we have*

$$\frac{1}{T - \tau_T} \sum_{t=\tau_T}^{T-1} \mathbb{E}[y_t^2] = \mathcal{O}\left(\frac{\log^2 T}{\sqrt{T}}\right) + \mathcal{O}(\epsilon_{\text{app}}), \tag{8a}$$

$$\frac{1}{T - \tau_T} \sum_{t=\tau_T}^{T-1} \mathbb{E}\|\boldsymbol{z}_t\|^2 = \mathcal{O}\left(\frac{\log^2 T}{\sqrt{T}}\right) + \mathcal{O}(\epsilon_{\text{app}}), \tag{8b}$$

$$\frac{1}{T - \tau_T} \sum_{t=\tau_T}^{T-1} \mathbb{E}\|\nabla J(\boldsymbol{\theta}_t)\|^2 = \mathcal{O}\left(\frac{\log^2 T}{\sqrt{T}}\right) + \mathcal{O}(\epsilon_{\text{app}}). \tag{8c}$$

Theorem 1 establishes the finite-time convergence of Algorithm 1. Given that the problem is inherently non-convex in general, it is common to prove convergence to a stationary point. The error term $\mathcal{O}(\epsilon_{\text{app}})$ represents the critic approximation error that commonly appears in the analysis of AC methods (Wu et al., 2020; Chen & Zhao, 2024; Tian et al., 2024). If the critic approximation error $\epsilon_{\text{app}}$ is zero, the critic and the actor errors all vanish at a rate of $\widetilde{\mathcal{O}}(T^{-1/2})$. The $\widetilde{\mathcal{O}}$ notation hides the polynomials of all other problem parameters that do not depend on $T$ and $\epsilon_{\text{app}}$. The additional logarithmic term with respect to $T$ arises from the mixing time of the Markov chain, which can be further eliminated if considering the i.i.d. sampling scheme (Chen & Zhao, 2024). As summarized in Table 1, we establish convergence of single-timescale AC under the most practical settings.

The main challenge of our analysis lies in controlling the coupled reward error (Eq. (8a)), the critic error (Eq. (8b)), and the actor error (Eq. (8c)). We begin by deriving implicit and coupled bounds for the time-average reward error, the critic error, and the actor error, respectively. We then view the propagation of these errors as an interconnected system (Chen & Zhao, 2024) and analyze them holistically. To better appreciate the merit of our analysis, we sketch the main proof steps of Theorem 1 in the Proof Sketch in Appendix C.

## 5 EXPERIMENTS

### 5.1 APPROXIMATION CAPABILITY OF THE NEURAL CRITIC

We evaluate Algorithm 1 on the `Gymnasium Pendulum-v1` task. This is a canonical control task with a continuous state space described by $\boldsymbol{s} = (\cos\theta, \sin\theta, \dot{\theta})$ (minimal coordinates $(\theta, \dot{\theta})$) and a continuous torque action. The critic is parameterized by a DNN of the form Eq. (5), and the actor is a Gaussian policy whose mean and variance are produced by a DNN with the same architecture. For comparison, we employ a linear critic parameterized by a fixed 6-term RBF feature map, $\widehat{V}(\boldsymbol{s}) = \boldsymbol{\omega}^\top \boldsymbol{\phi}(\boldsymbol{s})$ with $\boldsymbol{\omega} \in \mathbb{R}^6$. The feature vector consists of Gaussian RBFs defined on $(\cos\theta, \sin\theta, \dot{\theta})$: $\phi_i(\boldsymbol{s}) = \exp\left(-\frac{\|\boldsymbol{s}-\boldsymbol{c_i}\|_2^2}{2\sigma^2}\right), i = 1, \ldots, 6$, where the centers $\{\boldsymbol{c_i} \in \mathbb{R}^3\}$ are placed uniformly and $\sigma$

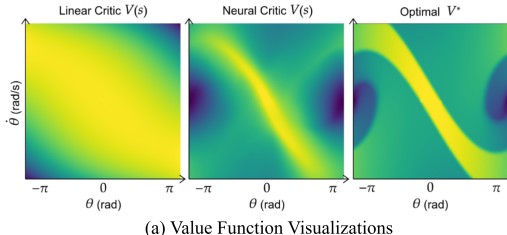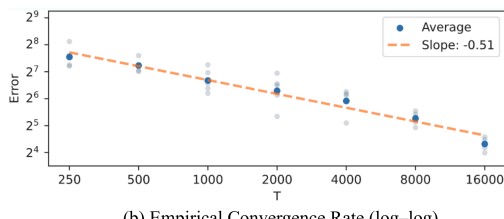

(a) Value Function Visualizations      (b) Empirical Convergence Rate (log–log)

Figure 1: **(a)** Comparison of learned value functions: (left) linear critic, (middle) neural critic, and (right) optimal $V^*$ obtained via time-average reward relative value iteration. Heatmaps are rendered with the Viridis colormap, where blue indicates low values, green intermediate values, and yellow high values. **(b)** Log–log plot of $E_T := \frac{1}{T-\tau_T} \sum_{t=\tau_T}^{T-1} \mathbb{E}\|\nabla J(\boldsymbol{\theta}_t)\|^2$ (Eq. (8c)) versus $T$, with $\tau_T = 125$ (Theorem 1 holds for $T \geq 2\tau_T$). Colored markers represent five independent runs, and the dashed line shows their mean with a linear regression fit.

Table 2: Final average reward under different configurations (mean $\pm$ std over 5 seeds). Width sweep uses fixed depth = 2; depth sweep uses fixed width = 128.

| Config | Ant | HalfCheetah | Hopper | Humanoid | Swimmer | Walker2d |
|---|---|---|---|---|---|---|
| Linear | 797.1±66.0 | 299.2±61.9 | 61.4±25.2 | 186.9±14.7 | 35.9±4.7 | 810.9±290.6 |
| Width-64 | 1120.0±140.3 | 590.7±135.6 | 108.5±16.3 | 264.0±56.1 | 132.5±78.4 | 1215.3±192.6 |
| Width-128 | **1587.4**±183.2 | 1425.8±161.7 | 533.8±64.7 | 291.1±63.9 | 220.5±41.8 | **1400.9**±461.2 |
| Width-256 | 1245.2±126.7 | **2250.1**±187.9 | **725.3**±165.0 | 365.2±64.3 | **251.3**±8.8 | 1390.9±324.9 |
| Width-512 | 949.2±75.4 | 1691.6±245.8 | **749.3**±304.6 | **448.9**±48.4 | 222.7±22.7 | 996.5±180.9 |
| Depth-1 | 961.2±8.0 | 1205.8±293.5 | 174.6±34.4 | 219.0±24.3 | 173.6±101.1 | 1118.4±39.5 |
| Depth-2 | 1587.4±183.2 | 1425.8±381.7 | **533.8**±64.7 | 291.1±63.9 | 201.2±54.2 | **1400.9**±461.2 |
| Depth-4 | **1824.9**±147.0 | **2144.2**±229.6 | 465.6±95.6 | 385.0±50.0 | 182.6±26.8 | 865.1±196.5 |
| Depth-8 | 1021.0±58.3 | 1699.2±285.4 | 210.8±68.2 | **546.4**±63.7 | **230.9**±57.7 | 1136.9±45.0 |

is determined by a standard width rule (Konidaris et al., 2011). For visualization, the ground-truth baseline is computed via time-average reward relative value iteration (RVI) (Bertsekas, 1998). As illustrated in Figure 1(a), the neural critic aligns more closely with the ground-truth value.

## 5.2 EMPIRICAL VALIDATION OF THEORETICAL CONVERGENCE RATE.

In this experiment, we follow the same setting as in Section 5.1. We empirically estimate the convergence rate of Algorithm 1 (Eq. (8c)) to examine its consistency with the theoretical rate of $\widetilde{\mathcal{O}}(T^{-1/2})$. As shown in Figure 1, after an initial warm-up period of about 250 iterations (recall that Theorem 1 applies to $T \geq 2\tau_T$), the curve exhibits a clear linear trend. Fitting a single slope to the mean trajectory yields $-0.51$, which aligns closely with the theoretical value of $-0.5$. This agreement provides direct empirical support for our theoretical convergence rate.

## 5.3 ALGORITHM EVALUATION ON MUJOCO BENCHMARKS

We further evaluate Algorithm 1 on challenging continuous-control benchmarks from Gym Mu-JoCo, including *Ant*, *HalfCheetah*, *Hopper*, *Humanoid*, *Swimmer*, and *Walker2d*. We conduct ablations along three axes: (i) linear critic, (ii) neural critic with varying depths, and (iii) neural critic with varying widths. The linear critic is identical to that used in Section 5.1. Table 2 reports the final average rewards over five seeds, while Figure 2 illustrates the learning curves of two selected entries from the table. Overall, the linear critic underperforms substantially across all tasks. These experiments validate the effectiveness of our considered algorithm on practical tasks and also reinforce the importance of analyzing realistic neural AC settings.

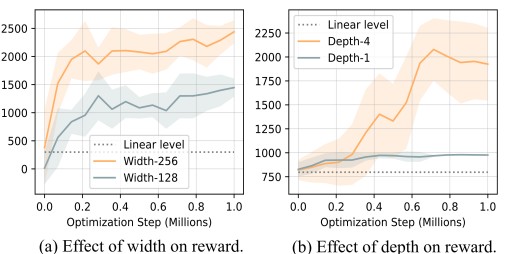

(a) Effect of width on reward.    (b) Effect of depth on reward.

Figure 2: Reward curves under different capacities: *HalfCheetah* (width sweep) and *Ant* (depth sweep). Mean $\pm$ std over 5 seeds; dashed = linear baseline.

## 6 CONCLUSION AND DISCUSSION

In this paper, we provide the first finite-time analysis of single-timescale AC with deep neural network approximation in continuous state–action spaces under the time-average reward setting. Our results surpass those of existing works by effectively addressing continuous state and action spaces, utilizing Markovian sampling, and employing deep neural network approximations for both critic and actor. We also conduct extensive experiments to validate the convergence guarantees of the analyzed algorithm.

## ACKNOWLEDGEMENTS

This work was supported by the Singapore Ministry of Education Tier 1 Academic Research Fund (A-8001174-00-00) and Tier 2 Academic Research Funds (T2EP20123-0037, T2EP20224-0035).

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

# Supplementary Material

## Table of Contents

## A  RELATED WORK

**Actor-Critic methods.** The AC algorithm was initially proposed by Konda & Tsitsiklis (1999). Subsequently, Kakade (2001) extended it to the natural AC algorithm. The asymptotic convergence of AC algorithms has been well established under various settings, as demonstrated in works by Kakade (2001), Bhatnagar et al. (2009), Castro & Meir (2010), and Zhang et al. (2020b). More recently, many studies have focused on the finite-time convergence of AC methods. Under the double-loop setting, Kumar et al. (2019) investigated the finite-time local convergence of several AC variants with linear function approximation. Wang et al. (2019) explored the global convergence of AC methods with both the actor and the critic parameterized by neural networks with single hidden layers. Cayci et al. (2022) improved upon the work of Wang et al. (2019) by considering Markovian sampling and reducing sample complexity. Xu et al. (2020a) analyzed natural AC under Markovian sampling, while Chen et al. (2022) studied decentralized AC and decentralized natural AC in the same setting. More recently, Gaur et al. (2024) and Zhang et al. (2025) established global optimality convergence for double-loop AC methods.

Under the two-timescale AC setting, Wu et al. (2020) established the finite-time local convergence to a stationary point at a sample complexity of $\widetilde{\mathcal{O}}(\epsilon^{-2.5})$ under the undiscounted time-average reward setting. Xu et al. (2020c) studied both local convergence and global convergence for two-timescale (natural) AC, with $\widetilde{\mathcal{O}}(\epsilon^{-2.5})$ and $\widetilde{\mathcal{O}}(\epsilon^{-4})$ sample complexity, respectively, under the discounted accumulated reward. The algorithm collects multiple samples to update the critic. Hong et al. (2023) proposed a two-timescale stochastic approximation algorithm for bilevel optimization and the algorithm was subsequently employed in the context of two-timescale AC, serving as a basis

for a wide range of multi-agent reinforcement learning analyses (Wang et al., 2021; Zhang et al., 2023a; 2024). Chen et al. (2023) established the global convergence of two-timescale AC methods for solving linear quadratic regulator (LQR), where only a single sample is used to update the critic in each iteration. However, none of these previous results utilized neural network approximation for the value function (the critic).

Under the most challenging single-timescale setting, Fu et al. (2020) considered the least-squares temporal difference (LSTD) update for the critic and obtained the optimal policy within the energy-based policy class for both linear function approximation and neural network approximation. Zhou & Lu (2023) studied single-timescale AC on LQR. In addition, Chen et al. (2021); Olshevsky & Gharesifard (2023); Chen & Zhao (2024) considered the single-timescale AC in general MDP cases with linear function approximation. Recently, Tian et al. (2024) built upon the results of Olshevsky & Gharesifard (2023) and improved to neural network approximation. A comprehensive review and comparison of all existing results on single-timescale AC in general MDP settings are presented in Table 1.

## B  ADDITIONAL NOTATIONS

We make use of the following auxiliary Markov chain which was introduced in (Zou et al., 2019) to deal with the Markovian noise.

**Auxiliary Markov Chain:**

$$s_{t-\tau} \xrightarrow{\boldsymbol{\theta}_{t-\tau}} a_{t-\tau} \xrightarrow{\mathcal{P}} s_{t-\tau+1} \xrightarrow{\boldsymbol{\theta}_{t-\tau}} \widetilde{a}_{t-\tau+1} \xrightarrow{\mathcal{P}} \widetilde{s}_{t-\tau+2} \xrightarrow{\boldsymbol{\theta}_{t-\tau}} \widetilde{a}_{t-\tau+2} \cdots \xrightarrow{\mathcal{P}} \widetilde{s}_t \xrightarrow{\boldsymbol{\theta}_{t-\tau}} \widetilde{a}_t \xrightarrow{\mathcal{P}} \widetilde{s}_{t+1}.$$
(9)

For reference, we also show the original Markov chain.

**Original Markov Chain:**

$$s_{t-\tau} \xrightarrow{\boldsymbol{\theta}_{t-\tau}} a_{t-\tau} \xrightarrow{\mathcal{P}} s_{t-\tau+1} \xrightarrow{\boldsymbol{\theta}_{t-\tau+1}} \widetilde{a}_{t-\tau+1} \xrightarrow{\mathcal{P}} s_{t-\tau+2} \xrightarrow{\boldsymbol{\theta}_{t-\tau+2}} a_{t-\tau+2} \cdots \xrightarrow{\mathcal{P}} s_t \xrightarrow{\boldsymbol{\theta}_t} a_t \xrightarrow{\mathcal{P}} s_{t+1}.$$
(10)

In the sequel, we denote by $\widetilde{O}_t := (\widetilde{s}_t, \widetilde{a}_t, \widetilde{s}_{t+1})$ the tuple generated from the auxiliary Markov chain in Eq. (9) while $O_t := (s_t, a_t, s_{t+1})$ denotes the tuple generated from the original Markov chain in Eq. (10).

In our work, we use the term Markovian sampling to refer to the setting where all samples are drawn from a Markov chain. Concretely, the samples follow

$$(s_0, a_0) \xrightarrow{(\mathcal{P}, \pi_{\boldsymbol{\theta}_1})} (s_1, a_1) \xrightarrow{(\mathcal{P}, \pi_{\boldsymbol{\theta}_2})} (s_2, a_2) \cdots \xrightarrow{(\mathcal{P}, \pi_{\boldsymbol{\theta}_t})} (s_t, a_t),$$
(11)

forming one trajectory $(s_0, a_0, s_1, a_1, \ldots, s_t, a_t)$.

**Remark.** Note that in Table 1 we label the actor sampling in Tian et al. (2024) as not Markovian. This arises from the fact that Tian et al. (2024) adopts a sampling scheme that is fundamentally different from the Markovian sampling in Eq. (11). For each update at timestep $t$, the state–action pair $(\hat{s}_t, \hat{a}_t)$ used in the actor update in Tian et al. (2024) is obtained by sampling a random horizon

$$T \sim \mathrm{Geom}(1 - \gamma),$$

rolling out a trajectory $(s_0, a_0, s_1, a_1, \ldots, s_T, a_T)$, and using only the terminal pair $(\hat{s}_t, \hat{a}_t) := (s_T, a_T)$. Each $(\hat{s}_t, \hat{a}_t)$ therefore arises from an *independent* rollout, and successive samples do not satisfy any Markovian dependency:

$$(\hat{s}_t, \hat{a}_t) \not\to (\hat{s}_{t+1}, \hat{a}_{t+1}).$$

Because this sampling is based on independent random-horizon rollouts, no Markovian noise arises in this part. In practice, this is also less sample-efficient than single-trajectory Markovian sampling.

For this reason, although Tian et al. (2024) refer to their scheme as "Markovian sampling", we view it as fundamentally different from the standard usage of the term and label it as "not Markovian" in Table 1. Notably, this same sampling mechanism has been used in Zhang et al. (2020a), who

explicitly refer to it as *random-horizon policy gradient*. Following this terminology, we believe "random-horizon sampling" is a more accurate description for this type of sampling.

We define the following functions, which will benefit to decompose the errors and simplify the presentation.

$$
\begin{aligned}
\Delta g(O, \eta, \boldsymbol{\theta}) &:= [J(\boldsymbol{\theta}) - \eta] \nabla \widehat{V}(\boldsymbol{\omega}; s), \\
g(O, \boldsymbol{\omega}, \boldsymbol{\theta}) &:= [r(s, a) - J(\boldsymbol{\theta}) + \widehat{V}(\boldsymbol{\omega}; s') - \widehat{V}(\boldsymbol{\omega}; s)] \nabla \widehat{V}(\boldsymbol{\omega}; s), \\
\bar{g}(\boldsymbol{\omega}, \boldsymbol{\theta}) &:= \mathbb{E}_{(s,a,s') \sim (\mu_{\boldsymbol{\theta}}, \pi_{\boldsymbol{\theta}}, \mathcal{P})} [(r(s, a) - J(\boldsymbol{\theta}) + \widehat{V}(\boldsymbol{\omega}; s') - \widehat{V}(\boldsymbol{\omega}; s)) \nabla \widehat{V}(\boldsymbol{\omega}; s)], \\
\Delta h(O, \eta, \boldsymbol{\omega}, \boldsymbol{\theta}) &:= (J(\boldsymbol{\theta}) - \eta + \widehat{V}(\boldsymbol{\omega}; s') - \widehat{V}(\boldsymbol{\omega}; s) - \widehat{V}(\boldsymbol{\omega}^*(\boldsymbol{\theta}); s') + \widehat{V}(\boldsymbol{\omega}^*(\boldsymbol{\theta}); s)) \nabla \log \pi_{\boldsymbol{\theta}}(a|s), \\
h(O, \boldsymbol{\theta}) &:= (r(s, a) - J(\boldsymbol{\theta}) + \widehat{V}(\boldsymbol{\omega}^*(\boldsymbol{\theta}); s') - \widehat{V}(\boldsymbol{\omega}^*(\boldsymbol{\theta}); s)) \nabla \log \pi_{\boldsymbol{\theta}}(a|s), \\
\Delta h'(O, \boldsymbol{\theta}) &:= ((\widehat{V}(\boldsymbol{\omega}^*(\boldsymbol{\theta}); s') - V_{\boldsymbol{\theta}}(s')) - (\widehat{V}(\boldsymbol{\omega}^*(\boldsymbol{\theta}); s) - V_{\boldsymbol{\theta}}(s))) \nabla \log \pi_{\boldsymbol{\theta}}(a|s).
\end{aligned}
\tag{12}
$$

We also define the following functions, which characterize the Markovian noise.

$$
\begin{aligned}
\Phi(O, \eta, \boldsymbol{\theta}) &:= (\eta - J(\boldsymbol{\theta}))(r(s, a) - J(\boldsymbol{\theta})), \\
\Psi(O, \boldsymbol{\omega}, \boldsymbol{\theta}) &:= \langle \boldsymbol{\omega} - \boldsymbol{\omega}_{\boldsymbol{\theta}}^*, g(O, \boldsymbol{\omega}, \boldsymbol{\theta}) - \bar{g}(\boldsymbol{\omega}, \boldsymbol{\theta}) \rangle, \\
\Xi(O, \boldsymbol{\omega}, \boldsymbol{\theta}) &:= \langle \boldsymbol{\omega} - \boldsymbol{\omega}_{\boldsymbol{\theta}}^*, (\nabla \boldsymbol{\omega}_{\boldsymbol{\theta}}^*)^\top (\mathbb{E}_{O'_{\boldsymbol{\theta}}}[h(O'_{\boldsymbol{\theta}}, \boldsymbol{\theta})] - h(O, \boldsymbol{\theta})) \rangle, \\
\Theta(O, \boldsymbol{\theta}) &:= \langle \nabla J(\boldsymbol{\theta}), \mathbb{E}_{O'_{\boldsymbol{\theta}}}[h(O'_{\boldsymbol{\theta}}, \boldsymbol{\theta})] - h(O, \boldsymbol{\theta}) \rangle,
\end{aligned}
\tag{13}
$$

where $O'_{\boldsymbol{\theta}}$ is a shorthand for an independent sample from stationary distribution $s \sim \mu_{\boldsymbol{\theta}}, a \sim \pi_{\boldsymbol{\theta}}, s' \sim \mathcal{P}$.

To demonstrate the main ideas of the proof of Theorem 1, we use the notations $Y_T, Z_T$ and $G_T$ for the three errors that we seek to bound, namely,

$$
\mathcal{E}^{(y)} := \frac{1}{T - \tau_T} \sum_{t=\tau_T}^{T-1} \mathbb{E} y_t^2, \quad \mathcal{E}^{(z)} := \frac{1}{T - \tau_T} \sum_{t=\tau_T}^{T-1} \mathbb{E} \|\boldsymbol{z}_t\|^2, \quad \mathcal{E}^{(\nabla)} := \frac{1}{T - \tau_T} \sum_{t=\tau_T}^{T-1} \mathbb{E} \|\nabla J(\boldsymbol{\theta}_t)\|^2.
\tag{14}
$$

Here $\mathcal{E}^{(y)}, \mathcal{E}^{(z)}$, and $\mathcal{E}^{(\nabla)}$ represent the reward error, critic error, and actor error (policy gradient norm), respectively. Our proof of Theorem 1 primarily involves analyzing and bounding these three errors relative to one another. The difficulty of this work lies in the continuous state and action spaces and the neural network approximation.

To ease the presentation, we define $u := \max\{u, u_{\boldsymbol{\omega}}, u_v, u_\pi\}$ as a uniform upper bound for $\eta, \boldsymbol{z}, \widehat{V}$ and $\nabla \log \pi_{\boldsymbol{\theta}}(a|s)$, where $u_v$ is defined in Lemma 1. Then we have $\|\delta \nabla \log \pi_{\boldsymbol{\theta}}\| \le 4u^2$. The norm of $\boldsymbol{\omega}$ is defined by $\|\boldsymbol{\omega}\| =: (\sum_{k=1}^K \|\boldsymbol{W}^{(k)}\|_{\mathrm{F}}^2)^{1/2}$, where $\|\cdot\|_{\mathrm{F}}$ is the Frobenius norm of a matrix.

## C  PROOF SKETCH

In this section, we outline the error-term analysis of Theorem 1. After bounding each component, the convergence follows by solving the interconnected iteration inequalities (E.4). The key challenges and new techniques developed are correspondingly *emphasized*.

**Reward error analysis.** using the reward estimator update rule (Line 7 of Algorithm 1), we decompose the reward error into:

$$
\begin{aligned}
y_{t+1}^2 \le (1 - 2\gamma) y_t^2 &+ 2\gamma y_t (r_t - J(\boldsymbol{\theta}_t)) + 2 y_t (J(\boldsymbol{\theta}_t) - J(\boldsymbol{\theta}_{t+1})) \\
&+ 2(J(\boldsymbol{\theta}_t) - J(\boldsymbol{\theta}_{t+1}))^2 + 2\gamma^2 (r_t - \eta_t)^2.
\end{aligned}
\tag{15}
$$

The second term on the right-hand side of Eq. (15) corresponds to a bias induced by *Markovian sampling in MDP with continuous state–action spaces under neural network function approximation*, which we addressed in Lemma 7. The third term captures the variation of the moving actor performance targets $J(\boldsymbol{\theta}_t)$ tracked by the reward error. Leveraging the smoothness of $J(\boldsymbol{\theta})$ (see Lemma 6) and the *boundedness of $\widehat{V}$* (see Lemma 1), we derive an implicit upper bound for this

term expressed as a function of $|y_t|$ and $\nabla J(\boldsymbol{\theta}_t)$. The fourth term represents the difference between the moving actor target, which can be controlled due to its Lipschitz continuity shown in Lemma 5. The last term in Eq. (15) reflects the variance in reward estimation, which is bounded by $\mathcal{O}(\gamma)$.

**Critic error analysis.** using the critic update rule (Line 8 of Algorithm 1), we decompose the squared error by:

$$
\begin{aligned}
\mathbb{E}\|\boldsymbol{z}_{t+1}\|^2 \leq{} & \mathbb{E}\|\boldsymbol{z}_t\|^2 + 2\beta\mathbb{E}\langle\boldsymbol{z}_t, \bar{g}(\boldsymbol{\omega}_t, \boldsymbol{\theta}_t)\rangle + 2\beta\mathbb{E}\Psi(O_t, \boldsymbol{\omega}_t, \boldsymbol{\theta}_t) + 2\beta\mathbb{E}\langle\boldsymbol{z}_t, \Delta g(O_t, \eta_t, \boldsymbol{\theta}_t)\rangle \\
& + 2\mathbb{E}\langle\boldsymbol{z}_t, \boldsymbol{\omega}_t^* - \boldsymbol{\omega}_{t+1}^*\rangle + 2\mathbb{E}\|\boldsymbol{\omega}_t^* - \boldsymbol{\omega}_{t+1}^*\|^2 + 2\mathbb{E}\|\beta(g(O_t, \boldsymbol{\omega}_t, \boldsymbol{\theta}_t) + \Delta g(O_t, \eta_t, \boldsymbol{\theta}_t))\|^2.
\end{aligned}
\tag{16}
$$

The definitions of $g, \bar{g}, \Delta g, O_t,$ and $\Psi$ can be found in Appendix B. First of all, the second term on the right-hand side of Eq. (16) is the inner product between the critic error $\boldsymbol{z}_t$ and the critic's mean-path update $\bar{g}(\boldsymbol{\omega}_t, \boldsymbol{\theta}_t)$, which serves as the key to the convergence. Our analysis for this term is *distinct from all previous results* since considering continuous spaces and deep neural networks substantially complicate the bounding process. we employ the *Bellman equation* to manage error propagation and control the error by leveraging the *approximation capability of the neural network* (Eq. (7)), the *regularity of neural network* (Lemma 1), and *sufficient policy exploration* (see Eq. (27)). It provides an explicit characterization of how sufficient exploration can help the convergence of learning. The third term is a Markovian noise, which we bounded in Lemma 8. The fourth term is caused by inaccurate reward and critic estimations, which can be bounded by the norm of $y_t$ and $\boldsymbol{z}_t$ after applying *the Lipschitzness of $\widehat{V}$* as shown in Lemma 1. The fifth term tracks both the critic error $\boldsymbol{z}_t$ and the difference between the drifting critic targets $\boldsymbol{\omega}_t^*$. We establish an implicit upper bound for this term as a function of $y_t$ and $\boldsymbol{z}_t$. The sixth term represents the difference between the moving critic target, which can be controlled due to its Lipschitz continuity stated in Assumption 3. Finally, the last term reflects the variances of various estimations, which is bounded by $\mathcal{O}(\beta)$.

**Actor error analysis.** using the actor update rule (Line 9 of Algorithm 1) and the smoothness property of $J(\boldsymbol{\theta})$ (see Lemma 6), we derive

$$
\begin{aligned}
\mathbb{E}\|\nabla J(\boldsymbol{\theta}_t)\|^2 \leq{} & \frac{1}{\alpha}(\mathbb{E}[J(\boldsymbol{\theta}_{t+1}) - J(\boldsymbol{\theta}_t)]) - \mathbb{E}\langle\nabla J(\boldsymbol{\theta}_t), \Delta h(O_t, \eta_t, \boldsymbol{\omega}_t, \boldsymbol{\theta}_t)\rangle + \mathbb{E}[\Theta(O_t, \boldsymbol{\theta}_t)] \\
& - \mathbb{E}\langle\nabla J(\boldsymbol{\theta}_t), \mathbb{E}_{O_t'}[\Delta h'(O_t', \boldsymbol{\theta}_t)]\rangle + \frac{h_j}{2}\alpha\mathbb{E}\|\delta_t \nabla \log \pi_{\boldsymbol{\theta}_t}(a_t|s_t)\|^2.
\end{aligned}
\tag{17}
$$

where the definitions of $\Delta h, \Delta h', \Theta$ and $O_t'$ can be found in Appendix B. The first term on the right-hand side of Eq. (17) compares the actor's performances between consecutive updates, which can be bounded after summation. The second term is an error introduced by the inaccurate estimations of both the time-average reward and the critic. After employing the *Lipschitzness of $\widehat{V}$*, we control this term by providing an implicit bound depending on $y_t, \boldsymbol{z}_t,$ and $\nabla J(\boldsymbol{\theta}_t)$. The third term is a noise term introduced by Markovian sampling, which we handled in Lemma 10. The fourth term comes from the linear function approximation error. The final term represents the variance of the stochastic gradient update, which is controlled by $\mathcal{O}(\alpha)$ due to the *boundedness of $\widehat{V}$*, a result we specifically derived in Lemma 1.

## D  PRELIMINARY LEMMAS

**Lemma 1.** *There exists scalars $u_v, l_v,$ and $h_v$ such that for any $s \in \mathcal{S}$ and $\boldsymbol{\omega}_1, \boldsymbol{\omega}_2 \in \mathcal{X}_\Omega$,*

$$
\|\widehat{V}(\boldsymbol{\omega}; s)\| \leq u_v,
$$

$$
\|\widehat{V}(\boldsymbol{\omega}_1; s) - \widehat{V}(\boldsymbol{\omega}_2; s)\| \leq l_v\|\boldsymbol{\omega}_1 - \boldsymbol{\omega}_2\|,
$$

$$
\|\nabla\widehat{V}(\boldsymbol{\omega}_1; s) - \nabla\widehat{V}(\boldsymbol{\omega}_2; s)\| \leq h_v\|\boldsymbol{\omega}_1 - \boldsymbol{\omega}_2\|,
$$

*where $u_v = \mathcal{O}(1), l_v = \mathcal{O}(1)$ and $h_v = \widetilde{O}(\frac{1}{\sqrt{m}})$ with respect to width $m$.*

**Lemma 2.** *There exists a positive constant $l_\pi$ such that for any $\boldsymbol{\theta}_1, \boldsymbol{\theta}_2 \in \mathcal{X}_\Theta$, it holds that*

$$
d_{TV}(\pi_{\boldsymbol{\theta}_1}(\cdot\,|\,s), \pi_{\boldsymbol{\theta}_2}(\cdot\,|\,s)) \leq l_\pi\|\boldsymbol{\theta}_1 - \boldsymbol{\theta}_2\|.
\tag{18}
$$

**Lemma 3** (Distance between stationary distributions). *For any $\boldsymbol{\theta}_1$ and $\boldsymbol{\theta}_2$, it holds that*

$$d_{TV}(\mu_{\boldsymbol{\theta}_1}, \mu_{\boldsymbol{\theta}_2}) \leq l_\pi(\lceil \log_\rho \kappa^{-1} \rceil + \frac{1}{1-\rho}) \|\boldsymbol{\theta}_1 - \boldsymbol{\theta}_2\|,$$

$$d_{TV}(\mu_{\boldsymbol{\theta}_1} \otimes \pi_{\boldsymbol{\theta}_1}, \mu_{\boldsymbol{\theta}_2} \otimes \pi_{\boldsymbol{\theta}_2}) \leq l_\pi(1 + \lceil \log_\rho \kappa^{-1} \rceil + \frac{1}{1-\rho}) \|\boldsymbol{\theta}_1 - \boldsymbol{\theta}_2\|,$$

$$d_{TV}(\mu_{\boldsymbol{\theta}_1} \otimes \pi_{\boldsymbol{\theta}_1} \otimes \mathcal{P}, \mu_{\boldsymbol{\theta}_2} \otimes \pi_{\boldsymbol{\theta}_2} \otimes \mathcal{P}) \leq l_\pi(1 + \lceil \log_\rho \kappa^{-1} \rceil + \frac{1}{1-\rho}) \|\boldsymbol{\theta}_1 - \boldsymbol{\theta}_2\|.$$

**Lemma 4** (Distance between distributions induced by the original and auxiliary chains). *Given time indexes $t$ and $\tau$ such that $t \geq \tau > 0$, consider the auxiliary Markov chain in Eq. (9). Conditioning on $s_{t-\tau+1}$ and $\boldsymbol{\theta}_{t-\tau}$, we have*

$$d_{TV}(\mathbb{P}(s_{t+1} \in \cdot), \mathbb{P}(\widetilde{s}_{t+1} \in \cdot)) \leq d_{TV}(\mathbb{P}(O_t \in \cdot), \mathbb{P}(\widetilde{O}_t \in \cdot)),$$

$$d_{TV}(\mathbb{P}(O_t \in \cdot), \mathbb{P}(\widetilde{O}_t \in \cdot)) = d_{TV}(\mathbb{P}((s_t, a_t) \in \cdot), \mathbb{P}((\widetilde{s}_t, \widetilde{a}_t) \in \cdot)),$$

$$d_{TV}(\mathbb{P}((s_t, a_t) \in \cdot), \mathbb{P}((\widetilde{s}_t, \widetilde{a}_t) \in \cdot)) \leq d_{TV}(\mathbb{P}(s_t \in \cdot), \mathbb{P}(\widetilde{s}_t \in \cdot)) + \frac{1}{2} l_\pi \mathbb{E}[\|\boldsymbol{\theta}_t - \boldsymbol{\theta}_{t-\tau}\|].$$

**Lemma 5** ((Wu et al., 2020)). *For any $\boldsymbol{\theta}_1, \boldsymbol{\theta}_2$, we have*

$$|J(\boldsymbol{\theta}_1) - J(\boldsymbol{\theta}_2)| \leq l_j \|\boldsymbol{\theta}_1 - \boldsymbol{\theta}_2\|,$$

*where*

$$l_j = 2u l_\pi (1 + \lceil \log_\rho \kappa^{-1} \rceil + \frac{1}{1-\rho}). \tag{19}$$

**Lemma 6** ((Zhang et al., 2020a)). *For the performance function $J(\boldsymbol{\theta})$, there exists a constant $h_j > 0$ such that for all $\boldsymbol{\theta}_1, \boldsymbol{\theta}_2 \in \mathbb{R}^d$, it holds that*

$$\|\nabla J(\boldsymbol{\theta}_1) - \nabla J(\boldsymbol{\theta}_2)\| \leq h_j \|\boldsymbol{\theta}_1 - \boldsymbol{\theta}_2\|, \tag{20}$$

*which further implies*

$$J(\boldsymbol{\theta}_2) \geq J(\boldsymbol{\theta}_1) + \langle \nabla J(\boldsymbol{\theta}_1), \boldsymbol{\theta}_2 - \boldsymbol{\theta}_1 \rangle - \frac{h_j}{2} \|\boldsymbol{\theta}_1 - \boldsymbol{\theta}_2\|^2, \tag{21}$$

$$J(\boldsymbol{\theta}_2) \leq J(\boldsymbol{\theta}_1) + \langle \nabla J(\boldsymbol{\theta}_1), \boldsymbol{\theta}_2 - \boldsymbol{\theta}_1 \rangle + \frac{h_j}{2} \|\boldsymbol{\theta}_1 - \boldsymbol{\theta}_2\|^2. \tag{22}$$

# E    PROOF OF MAIN THEOREM

In this section, we aim to show the proof of Theorem 1.

We decompose the whole proof into four steps.

## E.1    STEP 1: REWARD ERROR ANALYSIS

In this subsection, we will establish an implicit bound for estimator.

**Lemma 7** (Markovian noise). *From any $t \geq \tau > 0$, we have*

$$\mathbb{E}[\Phi(O_t, \eta_t, \boldsymbol{\theta}_t)] \leq 16u^2 \tau \alpha l_j + 4u^2 \tau \gamma + 4u^2 \tau (\tau + 1) \alpha l_\pi + 4u^2 \kappa \rho^{\tau-1}.$$

**Theorem 2.** *Choose $\alpha = \frac{c}{\sqrt{T}}, \beta = \gamma = \frac{1}{\sqrt{T}}$, we have*

$$\mathcal{E}^{(y)} \leq \mathcal{O}(\frac{\log^2 T}{\sqrt{T}}) + 4cu\sqrt{\mathcal{E}^{(y)} \mathcal{E}^{(\nabla)}}. \tag{23}$$

*Proof.* From the update rule of reward estimator in Line 7 of Algorithm 1, we have

$$\eta_{t+1} - J(\boldsymbol{\theta}_{t+1}) = \eta_t - J(\boldsymbol{\theta}_t) + J(\boldsymbol{\theta}_t) - J(\boldsymbol{\theta}_{t+1}) + \gamma(r_t - \eta_t),$$

which implies

$$
\begin{aligned}
y_{t+1}^2 &= (y_t + J(\boldsymbol{\theta}_t) - J(\boldsymbol{\theta}_{t+1}) + \gamma(r_t - \eta_t))^2 \\
&\leq y_t^2 + 2y_t(J(\boldsymbol{\theta}_t) - J(\boldsymbol{\theta}_{t+1})) + 2\gamma y_t(r_t - \eta_t) \\
&\quad + 2(J(\boldsymbol{\theta}_t) - J(\boldsymbol{\theta}_{t+1}))^2 + 2\gamma^2(r_t - \eta_t)^2 \\
&= (1 - 2\gamma)y_t^2 + 2\gamma y_t(r_t - J(\boldsymbol{\theta}_t)) + 2y_t(J(\boldsymbol{\theta}_t) - J(\boldsymbol{\theta}_{t+1})) \\
&\quad + 2(J(\boldsymbol{\theta}_t) - J(\boldsymbol{\theta}_{t+1}))^2 + 2\gamma^2(r_t - \eta_t)^2.
\end{aligned}
\tag{24}
$$

Taking expectation up to $s_{t+1}$ (the whole trajectory), rearranging and summing from $\tau_T$ to $T-1$, we have

$$
\sum_{t=\tau_T}^{T-1} \mathbb{E}[y_t^2] \leq \underbrace{\sum_{t=\tau_T}^{T-1} \frac{1}{2\gamma} \mathbb{E}(y_t^2 - y_{t+1}^2)}_{I_1} + \underbrace{\sum_{t=\tau_T}^{T-1} \mathbb{E}[y_t(r_t - J(\boldsymbol{\theta}_t))]}_{I_2} + \underbrace{\sum_{t=\tau_T}^{T-1} \frac{1}{\gamma} \mathbb{E}[y_t(J(\boldsymbol{\theta}_t) - J(\boldsymbol{\theta}_{t+1})]}_{I_3}
$$

$$
+ \underbrace{\sum_{t=\tau_T}^{T-1} \frac{1}{\gamma} \mathbb{E}[(J(\boldsymbol{\theta}_t) - J(\boldsymbol{\theta}_{t+1}))^2]}_{I_4} + \underbrace{\sum_{t=\tau_T}^{T-1} \gamma \mathbb{E}[(r_t - \eta_t)^2]}_{I_5}.
$$

For term $I_1$, by direct computation, we have

$$
\begin{aligned}
I_1 &= \sum_{t=\tau_T}^{T-1} \frac{1}{2\gamma} \mathbb{E}(y_t^2 - y_{t+1}^2) \\
&\leq \frac{2u^2}{\gamma} \\
&= 2u^2\sqrt{T}.
\end{aligned}
$$

For term $I_2$, from Lemma 7, we have

$$
\mathbb{E}[y_t(r_t - J(\boldsymbol{\theta}_t))] \leq 16u^2\tau\alpha l_j + 4u^2\tau\gamma + 4u^2\tau(\tau+1)\alpha l_\pi + 4u^2\kappa\rho^{\tau-1}.
$$

Choose $\tau = \tau_T$, we have

$$
\begin{aligned}
I_2 &= \sum_{t=\tau_T}^{T-1} \mathbb{E}[y_t(r_t - J(\boldsymbol{\theta}_t))] \\
&\leq (16u^2 l_j \tau_T + 4u^2 l_\pi \tau_T(\tau_T + 1)) \sum_{t=\tau_T}^{T-1} \alpha \\
&\quad + 4u^2\tau_T \sum_{t=\tau_T}^{T-1} \gamma + 4u^2 \sum_{t=\tau_T}^{T-1} \frac{1}{\sqrt{T}} \\
&= (16cu^2 l_j \tau_T + 4cu^2 l_\pi \tau_T(\tau_T + 1) + 4u^2\tau_T + 4u^2) \frac{T - \tau_T}{\sqrt{T}}.
\end{aligned}
$$

For $I_3$, if $y_t > 0$, from Eq. (21), we have

$$
\begin{aligned}
y_t(J(\boldsymbol{\theta}_t) - J(\boldsymbol{\theta}_{t+1})) &\leq y_t\left(\frac{h_j}{2}\|\boldsymbol{\theta}_t - \boldsymbol{\theta}_{t+1}\|^2 + \langle \nabla J(\boldsymbol{\theta}_t), \boldsymbol{\theta}_t - \boldsymbol{\theta}_{t+1}\rangle\right) \\
&\leq uh_j\|\boldsymbol{\theta}_t - \boldsymbol{\theta}_{t+1}\|^2 + |y_t|\|\boldsymbol{\theta}_t - \boldsymbol{\theta}_{t+1}\|\|\nabla J(\boldsymbol{\theta}_t)\|.
\end{aligned}
$$

If $y_t \leq 0$, from Eq. (22), we have

$$
\begin{aligned}
y_t(J(\boldsymbol{\theta}_t) - J(\boldsymbol{\theta}_{t+1})) &\leq y_t\left(-\frac{h_j}{2}\|\boldsymbol{\theta}_t - \boldsymbol{\theta}_{t+1}\|^2 + \langle \nabla J(\boldsymbol{\theta}_t), \boldsymbol{\theta}_t - \boldsymbol{\theta}_{t+1}\rangle\right) \\
&\leq uh_j\|\boldsymbol{\theta}_t - \boldsymbol{\theta}_{t+1}\|^2 + |y_t|\|\boldsymbol{\theta}_t - \boldsymbol{\theta}_{t+1}\|\|\nabla J(\boldsymbol{\theta}_t)\|.
\end{aligned}
$$

Overall, we get

$$
\begin{aligned}
I_3 &= \sum_{t=\tau_T}^{T-1} \frac{1}{\gamma} \mathbb{E}[y_t(J(\boldsymbol{\theta}_t) - J(\boldsymbol{\theta}_{t+1}))] \\
&\leq \sum_{t=\tau_T}^{T-1} \frac{1}{\gamma} \mathbb{E}[uh_j\|\boldsymbol{\theta}_t - \boldsymbol{\theta}_{t+1}\|^2 + |y_t|\|\boldsymbol{\theta}_t - \boldsymbol{\theta}_{t+1}\|\|\nabla J(\boldsymbol{\theta}_t)\|] \\
&\leq \sum_{t=\tau_T}^{T-1} \mathbb{E}[16cu^3 h_j \alpha + 16cu^2|y_t|\|\nabla J(\boldsymbol{\theta}_t)\|] \\
&\leq 16c^2 u^3 h_j \frac{T - \tau_T}{\sqrt{T}} + 16cu^2 \left(\sum_{t=\tau_T}^{T-1} \mathbb{E}y_t^2\right)^{\frac{1}{2}} \left(\sum_{t=\tau_T}^{T-1} \mathbb{E}\|\nabla J(\boldsymbol{\theta}_t)\|^2\right)^{\frac{1}{2}}.
\end{aligned}
$$

For term $I_4$, we have

$$
\begin{aligned}
I_4 &= \sum_{t=\tau_T}^{T-1} \frac{1}{\gamma} \mathbb{E}[(J(\boldsymbol{\theta}_t) - J(\boldsymbol{\theta}_{t+1}))^2] \\
&\leq \sum_{t=\tau_T}^{T-1} \frac{1}{\gamma} l_j^2 \mathbb{E}\|\boldsymbol{\theta}_t - \boldsymbol{\theta}_{t+1}\|^2 \\
&\leq \sum_{t=\tau_T}^{T-1} \frac{1}{\gamma} 16 l_j^2 u^2 \alpha^2 \\
&= 16c^2 u^2 l_j^2 \frac{T - \tau_T}{\sqrt{T}}.
\end{aligned}
$$

For term $I_5$, we have

$$
\begin{aligned}
I_5 &= \sum_{t=\tau_T}^{T-1} \gamma \mathbb{E}[(r_t - J(\boldsymbol{\theta}_t))^2] \\
&\leq \sum_{t=\tau_T}^{T-1} 4u^2 \gamma \\
&= 4u^2 \frac{T - \tau_T}{\sqrt{T}}.
\end{aligned}
$$

Therefore, we get

$$
\begin{aligned}
\sum_{t=\tau_T}^{T-1} \mathbb{E}[y_t^2] &\leq I_1 + I_2 + I_3 + I_4 + I_5 \\
&\leq (16cu^2 l_j \tau_T + 4cu^2 l_\pi \tau_T(\tau_T + 1) \\
&\quad + 4u^2(\tau_T + 2) + 16c^2 u^2(uh_j + l_j^2))\frac{T - \tau_T}{\sqrt{T}} \\
&\quad + 2u^2\sqrt{T} + 4cu\left(\sum_{t=\tau_T}^{T-1} \mathbb{E}y_t^2\right)^{\frac{1}{2}}\left(\sum_{t=\tau_T}^{T-1} \mathbb{E}\|\nabla J(\boldsymbol{\theta}_t)\|^2\right)^{\frac{1}{2}}.
\end{aligned}
$$

Since $\tau_T = \mathcal{O}(\log T)$, we have $\frac{\sqrt{T}}{T - \tau_T} \leq \frac{2}{\sqrt{T}}$ for large $T$. Then we get

$$\frac{1}{T - \tau_T} \sum_{t=\tau_T}^{T-1} \mathbb{E}[y_t^2] \leq (16cu^2 l_j \tau_T + 4cu^2 l_\pi \tau_T(\tau_T + 1)$$

$$+ 4u^2(\tau_T + 3) + 16c^2 u^2(uh_j + l_j^2))\frac{1}{\sqrt{T}}$$

$$+ 4cu(\frac{1}{T - \tau_T} \sum_{t=\tau_T}^{T-1} \mathbb{E}y_t^2)^{\frac{1}{2}}(\frac{1}{T - \tau_T} \sum_{t=\tau_T}^{T-1} \mathbb{E}\|\nabla J(\boldsymbol{\theta}_t)\|^2)^{\frac{1}{2}}$$

$$= \mathcal{O}(\frac{\log^2 T}{\sqrt{T}}) + 4cu(\frac{1}{T - \tau_T} \sum_{t=\tau_T}^{T-1} \mathbb{E}y_t^2)^{\frac{1}{2}}(\frac{1}{T - \tau_T} \sum_{t=\tau_T}^{T-1} \mathbb{E}\|\nabla J(\boldsymbol{\theta}_t)\|^2)^{\frac{1}{2}}.$$

Thus we finish the proof. $\qquad\square$

### E.2 STEP 2: CRITIC ERROR ANALYSIS

In this subsection, we will establish an implicit upper bound for critic.

**Lemma 8** (Markovian noise). *For any $t \geq \tau > 0$, we have*

$$\mathbb{E}[\Psi(O_t, \boldsymbol{\omega}_t, \boldsymbol{\theta}_t)] \leq 4c_1 u^2 \tau\alpha + 4c_2 u^2 l_v \tau\beta + 16u^4 l_v l_\pi \tau(\tau + 1)\alpha + 8u^2 l_v \kappa\rho^{\tau-1},$$

*where*

$$c_1 = 4u^2 l_\pi(1 + \lceil \log_\rho \kappa^{-1} \rceil + \frac{1}{1 - \rho}) + 2ul_j l_v + 4ul_\omega l_v,$$

$$c_2 = 2u(8uh_v + 4l_v^2 + 2l_v).$$

**Lemma 9** (Markovian noise). *For any $t \geq \tau > 0$, we have*

$$\mathbb{E}[\Xi(O_t, \boldsymbol{\omega}_t, \boldsymbol{\theta}_t)] \leq 4c_3 u^2 \tau\alpha + 4u^3 l_\omega \tau\beta + 8u^5 l_\omega l_\pi \tau(\tau + 1)\alpha + 8u^3 l_\omega \kappa\rho^{\tau-1}.$$

*where $c_3 := 8u^2 l_\omega^2 + 8u^3 h_\omega + 6ul_\omega(2uh_\pi + ul_j + ul_v l_\omega)$.*

**Theorem 3.** *Choose $\alpha = \frac{c}{\sqrt{T}}, \beta = \gamma = \frac{1}{\sqrt{T}}$, we have*

$$\mathcal{E}^{(z)} \leq \mathcal{O}(\frac{\log^2 T}{\sqrt{T}}) + \frac{2u}{\lambda}\sqrt{\mathcal{E}^{(y)}\mathcal{E}^{(z)}} + \frac{2cul_\omega}{\lambda}\sqrt{\mathcal{E}^{(z)}(2\mathcal{E}^{(y)} + 8l_v^2 \mathcal{E}^{(z)})} + \frac{2cl_\omega}{\lambda}\sqrt{\mathcal{E}^{(z)}\mathcal{E}^{(\nabla)}} + \mathcal{O}(\epsilon_{\text{app}}).$$

$$(25)$$

*Proof.* From the update rule of critic in Line 8 of Algorithm 1, we have

$$\|\boldsymbol{\omega}_{t+1} - \boldsymbol{\omega}_{t+1}^*\| = \|proj_{\mathcal{B}_{\boldsymbol{\omega}_0}}(\boldsymbol{\omega}_t + \beta\delta_t \nabla\widehat{V}(\boldsymbol{\omega}_t; s_t)) - \boldsymbol{\omega}_{t+1}^*\|$$

$$= \|proj_{\mathcal{B}_{\boldsymbol{\omega}_0}}(\boldsymbol{\omega}_t + \beta\delta_t \nabla\widehat{V}(\boldsymbol{\omega}_t; s_t)) - proj_{\mathcal{B}_{\boldsymbol{\omega}_0}}(\boldsymbol{\omega}_{t+1}^*)\|$$

$$\leq \|\boldsymbol{\omega}_t + \beta\delta_t \nabla\widehat{V}(\boldsymbol{\omega}_t; s_t) - \boldsymbol{\omega}_{t+1}^*\|$$

$$= \|\boldsymbol{\omega}_t - \boldsymbol{\omega}_t^* + \boldsymbol{\omega}_t^* - \boldsymbol{\omega}_{t+1}^* + \beta\delta_t \nabla\widehat{V}(\boldsymbol{\omega}_t; s_t)\|$$

Therefore, we have

$$\|z_{t+1}\|^2 = \|z_t + \beta(g(O_t, \boldsymbol{\omega}_t, \boldsymbol{\theta}_t) + \Delta g(O_t, \eta_t, \boldsymbol{\theta}_t)) + \boldsymbol{\omega}_t^* - \boldsymbol{\omega}_{t+1}^*\|^2$$

$$= \|z_t\|^2 + 2\beta\langle z_t, g(O_t, \boldsymbol{\omega}_t, \boldsymbol{\theta}_t)\rangle + 2\beta\langle z_t, \Delta g(O_t, \eta_t, \boldsymbol{\theta}_t)\rangle$$

$$+ 2\langle z_t, \boldsymbol{\omega}_t^* - \boldsymbol{\omega}_{t+1}^*\rangle + \|\beta(g(O_t, \boldsymbol{\omega}_t, \boldsymbol{\theta}_t) + \Delta g(O_t, \eta_t, \boldsymbol{\theta}_t)) + \boldsymbol{\omega}_t^* - \boldsymbol{\omega}_{t+1}^*\|^2$$

$$= \|z_t\|^2 + 2\beta\langle z_t, \bar{g}(\boldsymbol{\omega}_t, \boldsymbol{\theta}_t)\rangle + 2\beta\Psi(O_t, \boldsymbol{\omega}_t, \boldsymbol{\theta}_t) + 2\beta\langle z_t, \Delta g(O_t, \eta_t, \boldsymbol{\theta}_t)\rangle$$

$$+ 2\langle z_t, \boldsymbol{\omega}_t^* - \boldsymbol{\omega}_{t+1}^*\rangle + \|\beta(g(O_t, \boldsymbol{\omega}_t, \boldsymbol{\theta}_t) + \Delta g(O_t, \eta_t, \boldsymbol{\theta}_t)) + \boldsymbol{\omega}_t^* - \boldsymbol{\omega}_{t+1}^*\|^2$$

$$\leq \|z_t\|^2 + 2\beta\langle z_t, \bar{g}(\boldsymbol{\omega}_t, \boldsymbol{\theta}_t)\rangle + 2\beta\Psi(O_t, \boldsymbol{\omega}_t, \boldsymbol{\theta}_t) + 2\beta\langle z_t, \Delta g(O_t, \eta_t, \boldsymbol{\theta}_t)\rangle$$

$$+ 2\langle z_t, \boldsymbol{\omega}_t^* - \boldsymbol{\omega}_{t+1}^*\rangle + 2\|\boldsymbol{\omega}_t^* - \boldsymbol{\omega}_{t+1}^*\|^2 + 2\|\beta(g(O_t, \boldsymbol{\omega}_t, \boldsymbol{\theta}_t) + \Delta g(O_t, \eta_t, \boldsymbol{\theta}_t))\|^2.$$

Taking expectation up to $s_{t+1}$, we have

$$\mathbb{E}\|z_{t+1}\|^2 \le \mathbb{E}\|z_t\|^2 + 2\beta \underbrace{\mathbb{E}\langle z_t, \bar{g}(\omega_t, \theta_t)\rangle}_{I_1} + 2\beta \underbrace{\mathbb{E}\Psi(O_t, \omega_t, \theta_t)}_{I_2} + 2\beta \underbrace{\mathbb{E}\langle z_t, \Delta g(O_t, \eta_t, \theta_t)\rangle}_{I_3}$$
$$+ 2\underbrace{\mathbb{E}\langle z_t, \omega_t^* - \omega_{t+1}^*\rangle}_{I_4} + 2\underbrace{\mathbb{E}\|\omega_t^* - \omega_{t+1}^*\|^2}_{I_5} + 2\underbrace{\mathbb{E}\|\beta(g(O_t, \omega_t, \theta_t) + \Delta g(O_t, \eta_t, \theta_t))\|^2}_{I_6}.$$

$$(26)$$

For term $I_1$, we first analyse the mean-path update $\bar{g}(\omega_t, \theta_t)$. From the definition in Eq. (12), we have

$$\bar{g}(\omega_t, \theta_t) := \mathbb{E}_{s_t, a_t, s_{t+1}}[(r(s_t, a_t) - J(\theta_t) + \widehat{V}(\omega_t; s_{t+1}) - \widehat{V}(\omega_t; s_t))\nabla\widehat{V}(\omega_t; s_t)]$$
$$\overset{(1)}{=} \mathbb{E}_{s_t, a_t, s_{t+1}}[(V(s_t) - V(s_{t+1}) + \widehat{V}(\omega_t; s_{t+1}) - \widehat{V}(\omega_t; s_t))\nabla\widehat{V}(\omega_t; s_t)]$$
$$= \mathbb{E}_{s_t}[(V(s_t) - \widehat{V}(\omega_t, s_t) - \mathbb{E}_{s_{t+1}, a_t}[V(s_{t+1}) - \widehat{V}(\omega_t, s_{t+1})|s_t])\nabla\widehat{V}(\omega_t; s_t)]$$

where (1) comes from the Bellman equation. For $\mathbb{E}_{s_{t+1}, a_t}[V(s_{t+1}) - \widehat{V}(\omega_t, s_{t+1})|s_t]$, it can be shown that

$$\mathbb{E}_{s_{t+1}, a_t}[V(s_{t+1}) - \widehat{V}(\omega_t, s_{t+1})|s_t]$$
$$= \int_{\mathcal{S}} \int_{\mathcal{A}} \pi_{\theta_t}(a_t|s_t)\mathcal{P}(s_{t+1}|s_t, a_t)(V(s_{t+1}) - \widehat{V}(\omega_t; s_{t+1}))da_t ds_{t+1}$$
$$\overset{(1)}{=} P_{\theta}(V(s) - \widehat{V}(\omega, s)),$$

where (1) follows from the definition of $P_{\theta}$ in Eq. (2).

Then for $\bar{g}(\omega_t, \theta_t)$, it follows that

$$\bar{g}(\omega_t, \theta_t) = \mathbb{E}_{s_t}[(I - P_{\theta_t})(V(s_t) - \widehat{V}(\omega_t, s_t))\nabla\widehat{V}(\omega_t; s_t)],$$

where $I$ is the identity operator. Therefore, we have

$$\langle z_t, \bar{g}(\omega_t, \theta_t)\rangle = \mathbb{E}\langle z_t, (I - P_{\theta_t})(V(s_t) - \widehat{V}(\omega_t; s_t))\nabla\widehat{V}(\omega_t; s_t)\rangle$$
$$= \mathbb{E}\langle z_t, (I - P_{\theta_t})(V(s_t) - \widehat{V}(\omega_t^*; s_t) + \widehat{V}(\omega_t^*; s_t) - \widehat{V}(\omega_t; s_t))\nabla\widehat{V}(\omega_t; s_t)\rangle$$
$$= \mathbb{E}\langle z_t, (I - P_{\theta_t})(V(s_t) - \widehat{V}(\omega_t^*; s_t))\nabla\widehat{V}(\omega_t; s_t)\rangle$$
$$\quad + \mathbb{E}\langle z_t, (I - P_{\theta_t})(\widehat{V}(\omega_t^*; s_t) - \widehat{V}(\omega_t; s_t))\nabla\widehat{V}(\omega_t; s_t)\rangle$$
$$= 2ul_v\epsilon_{\text{app}} + \mathbb{E}[(z_t^\top \nabla\widehat{V}(\omega_t; s_t)(I - P_{\theta_t})(\widehat{V}(\omega_t^*; s_t) - \widehat{V}(\omega_t; s_t)))]$$
$$= \underbrace{\mathbb{E}[(z_t^\top \nabla\widehat{V}(\omega_t; s_t) + (\widehat{V}(\omega_t^*; s_t) - \widehat{V}(\omega_t, s_t)))(I - P_{\theta_t})(\widehat{V}(\omega_t^*; s_t) - \widehat{V}(\omega_t; s_t))]}_{J_1}$$
$$\underbrace{- \mathbb{E}[((\widehat{V}(\omega_t^*; s_t) - \widehat{V}(\omega_t, s_t)))(I - P_{\theta_t})(\widehat{V}(\omega_t^*; s_t) - \widehat{V}(\omega_t; s_t))]}_{J_2} + 2ul_v\epsilon_{\text{app}}.$$

$$(27)$$

For term $J_1$, from mean-value theorem, we get

$$J_1 = \mathbb{E}[z_t^\top (\nabla\widehat{V}(\omega_t; s_t) - \nabla\widehat{V}(\omega_{\text{mid}}; s_t))(I - P_{\theta_t})(\widehat{V}(\omega_t^*; s_t) - \widehat{V}(\omega_t; s_t))]$$
$$\le 4uh_v\|z_t\|^2,$$

where $\omega_{\text{mid}} = \mu_1\omega_t + (1 - \mu_1)\omega_t^*$ with $\mu_1 \in [0, 1]$ and the inequality follows from Lemma 1.

For term $J_2$, it can be shown that

$$J_2 = -\langle \widehat{V}(\omega_t^*) - \widehat{V}(\omega_t), D_{\theta}(I - P_{\theta_t})(\widehat{V}(\omega_t^*) - \widehat{V}(\omega_t))\rangle$$
$$\overset{(1)}{\le} -\lambda_2\|(\widehat{V}(\omega_t^*) - \widehat{V}(\omega_t))\|^2$$
$$\overset{(2)}{\le} -\lambda_1^2\lambda_2\|z_t\|^2$$
$$\overset{(3)}{=} -\lambda\|z_t\|^2,$$

where (1) comes from Assumption 4, (2) is due to Assumption 5, (3) holds since we define

$$\lambda := \lambda_1^2 \lambda_2.$$

Overall, we obtain

$$I_1 \leq 4uh_v \mathbb{E}\|\boldsymbol{z}_t\|^2 - \lambda \mathbb{E}\|\boldsymbol{z}_t\|^2 + 2ul_v \epsilon_{\text{app}}. \tag{28}$$

From Lemma 1, we know that $h_v = \tilde{\mathcal{O}}(1/\sqrt{m})$. Therefore, choosing $m$ as a large constant such that

$$4uh_v \leq \frac{\lambda}{2}, \tag{29}$$

it follows that

$$I_1 \leq -\frac{\lambda}{2}\mathbb{E}\|\boldsymbol{z}_t\|^2 + 2ul_v \epsilon_{\text{app}}.$$

For term $I_2$, it can be analyzed by Lemma 8.

For term $I_3$, it follows that

$$\begin{aligned}
I_3 &= \mathbb{E}\langle \boldsymbol{z}_t, \Delta g(O_t, \eta_t, \boldsymbol{\theta}_t)\rangle \\
&\leq u\mathbb{E}|y_t|\|\boldsymbol{z}_t\|.
\end{aligned}$$

For term $I_4$, we have

$$\begin{aligned}
I_4 &= \mathbb{E}\langle \boldsymbol{z}_t, \boldsymbol{\omega}_t^* - \boldsymbol{\omega}_{t+1}^*\rangle \\
&= \underbrace{\mathbb{E}\langle \boldsymbol{z}_t, \boldsymbol{\omega}_t^* - \boldsymbol{\omega}_{t+1}^* + (\nabla\boldsymbol{\omega}_t^*)^\top(\boldsymbol{\theta}_{t+1} - \boldsymbol{\theta}_t)\rangle}_{J_3} \\
&\quad + \underbrace{\mathbb{E}\langle \boldsymbol{z}_t, -(\nabla\boldsymbol{\omega}_t^*)^\top(\boldsymbol{\theta}_{t+1} - \boldsymbol{\theta}_t)\rangle}_{J_4}.
\end{aligned}$$

For $J_3$, from the $h_\omega$-smoothness of $\boldsymbol{\omega}^*$ in Assumption 3, we obtain

$$J_3 \leq h_\omega \|\boldsymbol{z}_t\|\|\boldsymbol{\theta}_{t+1} - \boldsymbol{\theta}_t\|^2.$$

For $J_4$, it follows that

$$\begin{aligned}
\frac{J_4}{\alpha} &= \mathbb{E}\langle \boldsymbol{z}_t, -(\nabla\boldsymbol{\omega}_t^*)^\top \delta_t \nabla\log\pi_{\boldsymbol{\theta}_t}(a_t|s_t)\rangle \\
&= \mathbb{E}\langle \boldsymbol{z}_t, (\nabla\boldsymbol{\omega}_t^*)^\top(-\Delta h(O_t, \eta_t, \boldsymbol{\omega}_t, \boldsymbol{\theta}_t) - h(O_t, \boldsymbol{\theta}_t))\rangle \\
&= -\mathbb{E}\langle \boldsymbol{z}_t, (\nabla\boldsymbol{\omega}_t^*)^\top \Delta h(O_t, \eta_t, \boldsymbol{\omega}_t, \boldsymbol{\theta}_t)\rangle \\
&\quad + \mathbb{E}\langle \boldsymbol{z}_t, (\nabla\boldsymbol{\omega}_t^*)^\top(\mathbb{E}_{O_t'}[h(O_t', \boldsymbol{\theta}_t)] - h(O_t, \boldsymbol{\theta}_t) - \mathbb{E}_{O_t'}[h(O_t', \boldsymbol{\theta}_t)])\rangle \\
&= \mathbb{E}[\Xi(O_t, \boldsymbol{\omega}_t, \boldsymbol{\theta}_t)] - \mathbb{E}\langle \boldsymbol{z}_t, (\nabla\boldsymbol{\omega}_t^*)^\top\mathbb{E}_{O_t'}[h(O_t', \boldsymbol{\theta}_t)]\rangle \\
&\quad - \mathbb{E}\langle \boldsymbol{z}_t, (\nabla\boldsymbol{\omega}_t^*)^\top \Delta h(O_t, \eta_t, \boldsymbol{\omega}_t, \boldsymbol{\theta}_t)\rangle
\end{aligned} \tag{30}$$

Note that from Cauchy-Schwartz inequality and $l_\omega$ is the Lipschitz constant of $\boldsymbol{\omega}^*$ in Assumption 3, we have

$$-\mathbb{E}\langle \boldsymbol{z}_t, (\nabla\boldsymbol{\omega}_t^*)^\top \Delta h(O_t, \eta_t, \boldsymbol{\omega}_t, \boldsymbol{\theta}_t)\rangle \leq ul_\omega\sqrt{\mathbb{E}\|\boldsymbol{z}_t\|^2}\sqrt{2\mathbb{E}y_t^2 + 8l_v^2\mathbb{E}\|\boldsymbol{z}_t\|^2}. \tag{31}$$

From the fact that

$$\begin{aligned}
\mathbb{E}_{O_t'}[h(O_t', \boldsymbol{\theta}_t) - \Delta h'(O_t', \boldsymbol{\theta}_t)] &= \mathbb{E}_{O_t'}[(r(s_t, a_t) - J(\boldsymbol{\theta}_t) + V_{\boldsymbol{\theta}_t}(s_t') - V_{\boldsymbol{\theta}_t}(s_t))\nabla\log\pi_{\boldsymbol{\theta}_t}(a|s)] \\
&= \nabla J(\boldsymbol{\theta}_t),
\end{aligned}$$

we obtain

$$\mathbb{E}\langle \boldsymbol{z}_t, (\nabla\boldsymbol{\omega}_t^*)^\top\mathbb{E}_{O_t'}[h(O_t', \boldsymbol{\theta}_t)]\rangle = \mathbb{E}\langle \boldsymbol{z}_t, (\nabla\boldsymbol{\omega}_t^*)^\top\nabla J(\boldsymbol{\theta}_t)\rangle + \mathbb{E}\langle \boldsymbol{z}_t, (\nabla\boldsymbol{\omega}_t^*)^\top\mathbb{E}_{O_t'}[\Delta h'(O_t', \boldsymbol{\theta}_t)]\rangle.$$

It follows that

$$-\mathbb{E}\langle \boldsymbol{z}_t, (\nabla\boldsymbol{\omega}_t^*)^\top\nabla J(\boldsymbol{\theta}_t)\rangle \leq l_\omega\sqrt{\mathbb{E}\|\boldsymbol{z}_t\|^2}\sqrt{\mathbb{E}\|\nabla J(\boldsymbol{\theta}_t)\|^2}.$$

Furthermore, it holds that

$$
\begin{aligned}
\mathbb{E}_{O'}\|\Delta h'(O, \boldsymbol{\theta})\|^2 &= \mathbb{E}_{O'}\|((\widehat{V}(\boldsymbol{\omega}^*(\boldsymbol{\theta}); s') - V_{\boldsymbol{\theta}}(s')) - (\widehat{V}(\boldsymbol{\omega}^*(\boldsymbol{\theta}); s) - V_{\boldsymbol{\theta}}(s)))\nabla \log \pi_{\boldsymbol{\theta}}(a|s)\|^2 \\
&\leq \mathbb{E}_{O'}[2u^2((\widehat{V}(\boldsymbol{\omega}^*(\boldsymbol{\theta}); s') - V_{\boldsymbol{\theta}}(s'))^2 + (\widehat{V}(\boldsymbol{\omega}^*(\boldsymbol{\theta}); s) - V_{\boldsymbol{\theta}}(s))^2)] \\
&= 4u^2 \mathbb{E}_{O'}[(\widehat{V}(\boldsymbol{\omega}^*(\boldsymbol{\theta}); s) - V_{\boldsymbol{\theta}}(s))^2] \\
&= 4u^2 \epsilon_{\text{app}}^2.
\end{aligned}
\tag{32}
$$

Therefore, we have

$$
\begin{aligned}
-\langle \boldsymbol{z}_t, (\nabla \boldsymbol{\omega}_t^*)^\top \mathbb{E}_{O_t'}[h(O_t', \boldsymbol{\theta}_t)]\rangle &\leq u l_\omega \sqrt{\|\mathbb{E}_{O'}[\Delta h'(O_t, \boldsymbol{\theta}_t)]\|^2} + l_\omega \sqrt{\mathbb{E}\|\boldsymbol{z}_t\|^2}\sqrt{\mathbb{E}\|\nabla J(\theta_t)\|^2} \\
&\leq u l_\omega \sqrt{\mathbb{E}_{O'}\|\Delta h'(O_t, \boldsymbol{\theta}_t)\|^2} + l_\omega \sqrt{\mathbb{E}\|\boldsymbol{z}_t\|^2}\sqrt{\mathbb{E}\|\nabla J(\theta_t)\|^2} \\
&\leq 2u^2 l_\omega \epsilon_{\text{app}} + l_\omega \sqrt{\mathbb{E}\|\boldsymbol{z}_t\|^2}\sqrt{\mathbb{E}\|\nabla J(\theta_t)\|^2}.
\end{aligned}
\tag{33}
$$

Substituting Eq. (31) and Eq. (33) into Eq. (30) yields

$$
\begin{aligned}
J_4 &\leq \alpha \mathbb{E}\Xi(O_t, \boldsymbol{\omega}_t, \boldsymbol{\theta}_t) + 2\alpha B u l_\omega \epsilon_{\text{app}} \\
&\quad + \alpha u l_\omega \sqrt{\mathbb{E}\|z_t\|^2}\sqrt{2\mathbb{E}y_t^2 + 8l_v^2\mathbb{E}\|z_t\|^2} \\
&\quad + \alpha l_\omega \sqrt{\mathbb{E}\|z_t\|^2}\sqrt{\mathbb{E}\|\nabla J(\theta_t)\|^2}.
\end{aligned}
\tag{34}
$$

Overall, we obtain

$$
\begin{aligned}
I_4 = J_3 + J_4 &\leq h_\omega \|\boldsymbol{z}_t\|\|\boldsymbol{\theta}_{t+1} - \boldsymbol{\theta}_t\|^2 + \alpha \mathbb{E}\Xi(O_t, \boldsymbol{\omega}_t, \boldsymbol{\theta}_t) \\
&\quad + \alpha u l_\omega \sqrt{\mathbb{E}\|z_t\|^2}\sqrt{2\mathbb{E}y_t^2 + 8l_v^2\mathbb{E}\|z_t\|^2} \\
&\quad + \alpha l_\omega \sqrt{\mathbb{E}\|z_t\|^2}\sqrt{\mathbb{E}\|\nabla J(\theta_t)\|^2} + 2\alpha u^2 l_\omega \epsilon_{\text{app}}.
\end{aligned}
$$

For term $I_5$, it holds that

$$
\begin{aligned}
I_5 &= \mathbb{E}\|\boldsymbol{\omega}_t^* - \boldsymbol{\omega}_{t+1}^*\|^2 \\
&\leq l_\omega^2 \mathbb{E}\|\boldsymbol{\theta}_t - \boldsymbol{\theta}_{t+1}\|^2 \\
&\leq 16u^2 l_\omega^2 \alpha^2.
\end{aligned}
$$

For term $I_6$, it follows that

$$
\begin{aligned}
I_6 &= \mathbb{E}\|\beta(g(O_t, \boldsymbol{\omega}_t, \boldsymbol{\theta}_t) + \Delta g(O_t, \eta_t, \boldsymbol{\theta}_t))\|^2 \\
&\leq u^2 l_v^2 \beta^2.
\end{aligned}
$$

Plugging $I_1 - I_6$ into Eq. (26), we obtain

$$
\begin{aligned}
\mathbb{E}\|z_{t+1}\|^2 &\leq \mathbb{E}\|z_t\|^2 - \lambda\beta\mathbb{E}\|z_t\|^2 + 2\beta\mathbb{E}\Psi(O_t, \boldsymbol{\omega}_t, \boldsymbol{\theta}_t) + 2\beta u\mathbb{E}|y_t|\|z_t\| + 2h_\omega\|z_t\|\|\boldsymbol{\theta}_{t+1} - \boldsymbol{\theta}_t\|^2 \\
&\quad + 2\alpha\mathbb{E}\Xi(O_t, \boldsymbol{\omega}_t, \boldsymbol{\theta}_t) + 2\alpha u l_\omega \sqrt{\mathbb{E}\|z_t\|^2}\sqrt{2\mathbb{E}y_t^2 + 8l_v^2\mathbb{E}\|z_t\|^2} \\
&\quad + 2\alpha l_\omega \sqrt{\mathbb{E}\|z_t\|^2}\sqrt{\mathbb{E}\|\nabla J(\theta_t)\|^2} + 4\alpha u^2 l_\omega \epsilon_{\text{app}} + 4u\beta l_v \epsilon_{\text{app}} + 32l_\omega^2 u^2 \alpha^2 + 2u^2 l_v^2 \beta^2.
\end{aligned}
$$

Rearranging and summing from $\tau_T$ to $T-1$ gives

$$\lambda \sum_{\tau_T}^{T-1} \mathbb{E}\|\boldsymbol{z}_t\|^2 \leq \underbrace{\sum_{t=\tau_T}^{T-1} \frac{1}{\beta}(\mathbb{E}\|\boldsymbol{z}_t\|^2 - \mathbb{E}\|\boldsymbol{z}_{t+1}\|^2)}_{K_1} + \underbrace{2\sum_{t=\tau_T}^{T-1} \mathbb{E}\Psi(O_t, \boldsymbol{\omega}_t, \boldsymbol{\theta}_t)}_{K_2} + \underbrace{2c\sum_{t=\tau_T}^{T-1} \mathbb{E}\Xi(O_t, \boldsymbol{\omega}_t, \boldsymbol{\theta}_t)}_{K_3}$$

$$+ \underbrace{2u\sum_{t=\tau_T}^{T-1} \sqrt{\mathbb{E}y_t^2}\sqrt{\mathbb{E}\|\boldsymbol{z}_t\|^2}}_{K_4} + \underbrace{2cul_\omega \sum_{t=\tau_T}^{T-1} \sqrt{\mathbb{E}\|\boldsymbol{z}_t\|^2}\sqrt{2\mathbb{E}y_t^2 + 8l_v^2\mathbb{E}\|\boldsymbol{z}_t\|^2}}_{K_5}$$

$$+ \underbrace{2cl_\omega \sum_{t=\tau_T}^{T-1} \sqrt{\mathbb{E}\|\boldsymbol{z}_t\|^2}\sqrt{\mathbb{E}\|\nabla J(\theta_t)\|^2}}_{K_6}$$

$$+ \sum_{t=\tau_T}^{T-1} (2u^2 l_v^2\beta + 32cu^2 l_\omega^2\alpha + (4cu^2 l_\omega + 4ul_v)\epsilon_{\text{app}}).$$

In the sequel, we will tackle $K_1, K_2, K_3, K_4, K_5, K_6$ respectively.

For term $K_1$, we have

$$I_1 = \sum_{t=\tau_T}^{T-1} \frac{1}{\beta}(\mathbb{E}\|\boldsymbol{z}_t\|^2 - \mathbb{E}\|\boldsymbol{z}_{t+1}\|^2) \leq u^2\sqrt{T}.$$

For term $K_2$, from Lemma 8, choose $\tau = \tau_T$, we have

$$\mathbb{E}\Psi(O_t, \boldsymbol{\omega}_t, \boldsymbol{\theta}_t) \leq 4c_1 u^2\tau_T\alpha + 4c_2 u^2 l_v\tau_T\beta + 16u^4 l_v l_\pi\tau_T(\tau_T + 1)\alpha + \frac{8u^2 l_v}{\sqrt{T}}.$$

Then we get

$$K_2 = 2\sum_{T=\tau_T}^{T-1} \mathbb{E}\Psi(O_t, \boldsymbol{\omega}_t, \boldsymbol{\theta}_t) \leq 2\sum_{T=\tau_T}^{T-1} (4c_1 u^2\tau_T\alpha + 4c_2 u^2 l_v\tau_T\beta + 16u^4 l_v l_\pi\tau_T(\tau_T + 1)\alpha + \frac{8u^2 l_v}{\sqrt{T}}).$$

For term $K_3$, from Lemma 9, choose $\tau = \tau_T$, we have

$$\mathbb{E}[\Xi(O_t, \boldsymbol{\omega}_t, \boldsymbol{\theta}_t)] \leq 4c_3 u^2\tau_T\alpha + 8u^5 l_\omega l_\pi\tau_T(\tau_T + 1)\alpha + 4u^3 l_\omega\tau_T\beta + \frac{8u^3 l_\omega}{\sqrt{T}}.$$

Therefore, we have

$$K_3 = 2c\sum_{t=\tau_T}^{T-1} \mathbb{E}\Xi(O_t, \boldsymbol{\omega}_t, \boldsymbol{\theta}_t)$$

$$\leq 2c\sum_{t=\tau_T}^{T-1} (4c_3 u^2\tau_T\alpha + 8u^5 l_\omega l_\pi\tau_T(\tau_T + 1)\alpha + 4u^3 l_\omega\tau_T\beta + \frac{8u^3 l_\omega}{\sqrt{T}}).$$

For term $K_4$, $K_5$, and $K_6$, from Cauchy-Schwartz inequality, we have

$$K_4 \leq 2u(\sum_{t=\tau_T}^{T-1} \mathbb{E}y_t^2)^{\frac{1}{2}}(\sum_{t=\tau_T}^{T-1} \mathbb{E}\|\boldsymbol{z}_t\|^2)^{\frac{1}{2}},$$

$$K_5 \leq 2cul_\omega(\sum_{t=\tau_T}^{T-1} \mathbb{E}\|\boldsymbol{z}_t\|^2)^{\frac{1}{2}}(2\sum_{t=\tau_T}^{T-1} \mathbb{E}y_t^2 + 8l_v^2\sum_{t=\tau_T}^{T-1} \mathbb{E}\|\boldsymbol{z}_t\|^2)^{\frac{1}{2}},$$

$$K_6 \leq 2cl_\omega(\sum_{t=\tau_T}^{T-1} \mathbb{E}\|\boldsymbol{z}_t\|^2)^{\frac{1}{2}}(\sum_{t=\tau_T}^{T-1} \mathbb{E}\|\nabla J(\boldsymbol{\theta}_t)\|)^{\frac{1}{2}}.$$

Overall, we get

$$
\begin{aligned}
\lambda \sum_{t=\tau_T}^{T-1} \mathbb{E}\|\boldsymbol{z}_t\|^2 \leq{} & 2u\Big(\sum_{t=\tau_T}^{T-1} \mathbb{E}y_t^2\Big)^{\frac{1}{2}}\Big(\sum_{t=\tau_T}^{T-1} \mathbb{E}\|\boldsymbol{z}_t\|^2\Big)^{\frac{1}{2}} \\
& + 2cul_\omega\Big(\sum_{t=\tau_t}^{T-1} \mathbb{E}\|\boldsymbol{z}_t\|^2\Big)^{\frac{1}{2}}\Big(2\sum_{t=\tau_T}^{T-1} \mathbb{E}y_t^2 + 8l_v^2\sum_{t=\tau_T}^{T-1} \mathbb{E}\|\boldsymbol{z}_t\|^2\Big)^{\frac{1}{2}} \\
& + 2cl_\omega\Big(\sum_{t=\tau_T}^{T-1} \mathbb{E}\|\boldsymbol{z}_t\|^2\Big)^{\frac{1}{2}}\Big(\sum_{t=\tau_T}^{T-1} \mathbb{E}\|\nabla J(\boldsymbol{\theta}_t)\|\Big)^{\frac{1}{2}} \\
& + u^2\sqrt{T} + 2\sum_{T=\tau_T}^{T-1}\Big(4c_1u^2\tau_T\alpha + 4c_2u^2l_v\tau_T\beta + 16u^4l_vl_\pi\tau_T(\tau_T+1)\alpha + \frac{8u^2l_v}{\sqrt{T}}\Big) \\
& + 2c\sum_{t=\tau_T}^{T-1}\Big(4c_3u^2\tau_T\alpha + 8u^5l_\omega l_\pi\tau_T(\tau_T+1)\alpha + 4u^3l_\omega\tau_T\beta + \frac{8u^3l_\omega}{\sqrt{T}}\Big) \\
& + \sum_{t=\tau_T}^{T-1}\Big(2u^2l_v^2\beta + 32cu^2l_\omega^2\alpha + (4cu^2l_\omega + 4ul_v)\epsilon_{\mathrm{app}}\Big).
\end{aligned}
$$

Therefore, we have

$$
\begin{aligned}
\mathcal{E}^{(z)} \overset{(1)}{\leq}{} & \mathcal{O}\Big(\frac{\log^2 T}{\sqrt{T}}\Big) + \mathcal{O}(\epsilon_{\mathrm{app}}) + \frac{2u}{\lambda}\Big(\frac{1}{T-\tau_T}\sum_{t=\tau_T}^{T-1} \mathbb{E}y_t^2\Big)^{\frac{1}{2}}\Big(\frac{1}{T-\tau_T}\sum_{t=\tau_T}^{T-1} \mathbb{E}\|\boldsymbol{z}_t\|^2\Big)^{\frac{1}{2}} \\
& + \frac{2cul_\omega}{\lambda}\Big(\frac{1}{T-\tau_T}\sum_{t=\tau_t}^{T-1} \mathbb{E}\|\boldsymbol{z}_t\|^2\Big)^{\frac{1}{2}}\Big(2\frac{1}{T-\tau_T}\sum_{t=\tau_T}^{T-1} \mathbb{E}y_t^2 + 8l_v^2\frac{1}{T-\tau_T}\sum_{t=\tau_T}^{T-1} \mathbb{E}\|\boldsymbol{z}_t\|^2\Big)^{\frac{1}{2}} \\
& + \frac{2cl_\omega}{\lambda}\Big(\frac{1}{T-\tau_T}\sum_{t=\tau_T}^{T-1} \mathbb{E}\|\boldsymbol{z}_t\|^2\Big)^{\frac{1}{2}}\Big(\frac{1}{T-\tau_T}\sum_{t=\tau_T}^{T-1} \mathbb{E}\|\nabla J(\boldsymbol{\theta}_t)\|\Big)^{\frac{1}{2}},
\end{aligned}
$$

where (1) follows from $\tau_T = \mathcal{O}(\log T)$ so that $T - \tau_T \geq \frac{1}{2}T$ for large $T$. Therefore, we have

$$
\mathcal{E}^{(z)} \leq \mathcal{O}\Big(\frac{\log^2 T}{\sqrt{T}}\Big) + \frac{2u}{\lambda}\sqrt{\mathcal{E}^{(y)}\mathcal{E}^{(z)}} + \frac{2cul_\omega}{\lambda}\sqrt{\mathcal{E}^{(z)}\big(2\mathcal{E}^{(y)} + 8l_v^2\mathcal{E}^{(z)}\big)} + \frac{2cl_\omega}{\lambda}\sqrt{\mathcal{E}^{(z)}\mathcal{E}^{(\nabla)}} + \mathcal{O}(\epsilon_{\mathrm{app}}),
$$

which completes the proof. $\qquad\square$

### E.3 STEP 3: ACTOR ERROR ANALYSIS

In this subsection, we will establish an implicit upper bound for actor error (policy gradient norm).

**Lemma 10** (Markovian noise). *For any $t \geq \tau > 0$, it holds that*

$$
\mathbb{E}[\Theta(O_t, \boldsymbol{\theta}_t)] \leq 4u^2(8u^2h_j + 3l_jl_h)\tau\alpha + 8u^4l_jl_\pi\tau(\tau+1)\alpha + 4u^2l_j\kappa\rho^{\tau-1}.
$$

**Theorem 4.** *We have*

$$
\mathcal{E}^{(\nabla)} \leq \mathcal{O}\Big(\frac{\log^2 T}{\sqrt{T}}\Big) + \mathcal{O}(\epsilon_{\mathrm{app}}) + u\sqrt{\mathcal{E}^{(\nabla)}\big(2\mathcal{E}^{(y)} + 8l_v^2\mathcal{E}^{(z)}\big)}. \tag{35}
$$

*Proof.* From the update rule of actor in Line 9 of Algorithm 1 and Eq. (21), we have

$$
\begin{aligned}
J(\boldsymbol{\theta}_{t+1}) &\geq J(\boldsymbol{\theta}_t) + \langle \nabla J(\boldsymbol{\theta}_t), \boldsymbol{\theta}_{t+1} - \boldsymbol{\theta}_t \rangle - \frac{h_j}{2}\|\boldsymbol{\theta}_t - \boldsymbol{\theta}_{t+1}\|^2 \\
&= J(\boldsymbol{\theta}_t) + \alpha\langle \nabla J(\boldsymbol{\theta}_t), \delta_t \nabla \log \pi_{\boldsymbol{\theta}_t}(a_t|s_t)\rangle - \frac{h_j}{2}\alpha^2\|\delta_t \nabla \log \pi_{\boldsymbol{\theta}_t}(a_t|s_t)\|^2 \\
&= J(\boldsymbol{\theta}_t) + \alpha\langle \nabla J(\boldsymbol{\theta}_t), \Delta h(O_t, \eta_t, \boldsymbol{\omega}_t, \boldsymbol{\theta}_t)\rangle \\
&\quad + \alpha\langle \nabla J(\boldsymbol{\theta}_t), h(O_t, \boldsymbol{\theta}_t)\rangle - \frac{h_j}{2}\alpha^2\|\delta_t \nabla \log \pi_{\boldsymbol{\theta}_t}(a_t|s_t)\|^2 \\
&= J(\boldsymbol{\theta}_t) + \alpha\langle \nabla J(\boldsymbol{\theta}_t), \Delta h(O_t, \eta_t, \boldsymbol{\omega}_t, \boldsymbol{\theta}_t)\rangle - \alpha\Theta(O_t, \boldsymbol{\theta}_t) \\
&\quad + \alpha\langle \nabla J(\boldsymbol{\theta}_t), \mathbb{E}_{O'_t}[h(O'_t, \boldsymbol{\theta}_t)]\rangle - \frac{h_j}{2}\alpha^2\|\delta_t \nabla \log \pi_{\boldsymbol{\theta}_t}(a_t|s_t)\|^2 \\
&= J(\boldsymbol{\theta}_t) + \alpha\langle \nabla J(\boldsymbol{\theta}_t), \Delta h(O_t, \eta_t, \boldsymbol{\omega}_t, \boldsymbol{\theta}_t)\rangle - \alpha\Theta(O_t, \boldsymbol{\theta}_t) + \alpha\|\nabla J(\boldsymbol{\theta}_t)\|^2 \\
&\quad + \alpha\langle \nabla J(\boldsymbol{\theta}_t), \mathbb{E}_{O'_t}[\Delta h'(O'_t, \boldsymbol{\theta}_t)]\rangle - \frac{h_j}{2}\alpha^2\|\delta_t \nabla \log \pi_{\boldsymbol{\theta}_t}(a_t|s_t)\|^2,
\end{aligned}
$$

where the last equality is due to the fact

$$
\mathbb{E}_{O'}[h(O', \boldsymbol{\theta}) - \Delta h'(O', \boldsymbol{\theta})] = \mathbb{E}_{O'}[(r(s,a) - J(\boldsymbol{\theta}) + V_{\boldsymbol{\theta}}(s') - V_{\boldsymbol{\theta}}(s))\nabla \log \pi_{\boldsymbol{\theta}}(a|s)] = \nabla J(\boldsymbol{\theta}).
$$

Rearranging the above inequality and taking expectation, we have

$$
\begin{aligned}
\mathbb{E}\|\nabla J(\boldsymbol{\theta}_t)\|^2 \leq &\frac{1}{\alpha}(\mathbb{E}[J(\boldsymbol{\theta}_{t+1}) - J(\boldsymbol{\theta}_t)]) - \mathbb{E}\langle \nabla J(\boldsymbol{\theta}_t), \Delta h(O_t, \eta_t, \boldsymbol{\omega}_t, \boldsymbol{\theta}_t)\rangle + \mathbb{E}[\Theta(O_t, \boldsymbol{\theta}_t)] \\
&- \mathbb{E}\langle \nabla J(\boldsymbol{\theta}_t), \mathbb{E}_{O'_t}[\Delta h'(O'_t, \boldsymbol{\theta}_t)]\rangle + \frac{h_j}{2}\alpha\mathbb{E}\|\delta_t \nabla \log \pi_{\boldsymbol{\theta}_t}(a_t|s_t)\|^2.
\end{aligned}
\tag{36}
$$

Note that from Cauchy-Schwartz inequality, we have

$$
-\mathbb{E}\langle \nabla J(\boldsymbol{\theta}_t), \Delta h(O_t, \eta_t, \boldsymbol{\omega}_t, \boldsymbol{\theta}_t)\rangle \leq u\sqrt{\mathbb{E}\|\nabla J(\boldsymbol{\theta}_t)\|^2}\sqrt{2\mathbb{E}y_t^2 + 8l_v^2\mathbb{E}\|\boldsymbol{z}_t\|^2}.
$$

From Lemma 10 and choosing $\tau = \tau_T$, we have

$$
\mathbb{E}[\Theta(O_t, \boldsymbol{\theta}_t)] \leq 4u^2(8u^2h_j + 3l_jl_h)\tau_T\alpha + 8u^4l_jl_\pi\tau_T(\tau_T + 1)\alpha + \frac{4u^2l_j}{\sqrt{T}}.
$$

From Eq. (32), it has been shown that

$$
\mathbb{E}_{O'}\|\Delta h'(O, \boldsymbol{\theta})\|^2 \leq 4u^2\epsilon_{\text{app}}^2.
$$

Therefore, we have

$$
\begin{aligned}
-\langle \nabla J(\boldsymbol{\theta}_t), \mathbb{E}_{O'_t}[\Delta h'(O'_t, \boldsymbol{\theta}_t)]\rangle &\leq l_j\sqrt{\|\mathbb{E}_{O'}[\Delta h'(O'_t, \boldsymbol{\theta}_t)]\|^2} \\
&\leq l_j\sqrt{\mathbb{E}_{O'}\|\Delta h'(O'_t, \boldsymbol{\theta}_t)\|^2} \\
&\leq 2ul_j\epsilon_{\text{app}},
\end{aligned}
$$

where we use $\|\nabla J(\boldsymbol{\theta})\| \leq l_j$ which comes from Lemma 5. Plugging the three terms yields

$$
\begin{aligned}
\mathbb{E}\|\nabla J(\boldsymbol{\theta}_t)\|^2 \leq &\frac{1}{\alpha}(\mathbb{E}[J(\boldsymbol{\theta}_{t+1})] - \mathbb{E}[J(\boldsymbol{\theta}_t)]) + u\sqrt{\mathbb{E}\|\nabla J(\boldsymbol{\theta}_t)\|^2}\sqrt{2\mathbb{E}y_t^2 + 8l_v^2\mathbb{E}\|\boldsymbol{z}_t\|^2} \\
&+ 4u^2(8u^2h_j + 3l_jl_h)\tau_T\alpha + 8u^4l_jl_\pi\tau_T(\tau_T + 1)\alpha + \frac{4u^2l_j}{\sqrt{T}} + 8u^4h_j\alpha + 2ul_j\epsilon_{\text{app}}.
\end{aligned}
$$

Summing over $t$ from $\tau_T$ to $T - 1$ gives

$$
\begin{aligned}
\sum_{t=\tau_T}^{T-1} \mathbb{E}\|\nabla J(\boldsymbol{\theta}_t)\|^2 \leq &\sum_{t=\tau_T}^{T-1} \frac{1}{\alpha}(\mathbb{E}[J(\boldsymbol{\theta}_{t+1})] - \mathbb{E}[J(\boldsymbol{\theta}_t)]) + u\sum_{t=\tau_T}^{T-1}\sqrt{\mathbb{E}\|\nabla J(\boldsymbol{\theta}_t)\|^2}\sqrt{2\mathbb{E}y_t^2 + 8l_v^2\mathbb{E}\|\boldsymbol{z}_t\|^2} \\
&+ \mathcal{O}(\log^2 T)\frac{T - \tau_T}{\sqrt{T}} + 2ul_j\epsilon_{\text{app}}(T - \tau_T) \\
\leq &\frac{2u}{c}\sqrt{T} + u(\sum_{t=\tau_T}^{T-1} \mathbb{E}\|\nabla J(\boldsymbol{\theta}_t)\|^2)^{\frac{1}{2}}(2\sum_{t=\tau_T}^{T-1} \mathbb{E}y_t^2 + 8l_v^2\sum_{t=\tau_T}^{T-1} \mathbb{E}\|\boldsymbol{z}_t\|^2)^{\frac{1}{2}} \\
&+ \mathcal{O}(\log^2 T)\frac{T - \tau_T}{\sqrt{T}} + 2ul_j\epsilon_{\text{app}}(T - \tau_T).
\end{aligned}
$$

Therefore, we get

$$\mathcal{E}^{(\nabla)} \leq \mathcal{O}(\frac{\log^2 T}{\sqrt{T}}) + 2ul_j\epsilon_{\mathrm{app}} + u\sqrt{\mathcal{E}^{(\nabla)}(2\mathcal{E}^{(y)} + 8l_v^2\mathcal{E}^{(z)})}$$

$$= \mathcal{O}(\frac{\log^2 T}{\sqrt{T}}) + \mathcal{O}(\epsilon_{\mathrm{app}}) + u\sqrt{\mathcal{E}^{(\nabla)}(2\mathcal{E}^{(y)} + 8l_v^2\mathcal{E}^{(z)})},$$

which concludes the proof. $\square$

### E.4 STEP 4: INTERCONNECTED ITERATION SYSTEM ANALYSIS

In this subsection, we perform an interconnected iteration system analysis to prove Theorem 1.

**Proof of Theorem 1.**

*Proof.* Combining Eq. (23), Eq. (25), and Eq. (35), we have

$$\mathcal{E}^{(y)} \leq \mathcal{O}(\frac{\log^2 T}{\sqrt{T}}) + 4cu\sqrt{\mathcal{E}^{(y)}\mathcal{E}^{(\nabla)}},$$

$$\mathcal{E}^{(z)} \leq \mathcal{O}(\frac{\log^2 T}{\sqrt{T}}) + \mathcal{O}(\epsilon_{\mathrm{app}}) + \frac{2u}{\lambda}\sqrt{\mathcal{E}^{(y)}\mathcal{E}^{(z)}} + \frac{2cul_\omega}{\lambda}\sqrt{\mathcal{E}^{(z)}(2\mathcal{E}^{(y)} + 8l_v^2\mathcal{E}^{(z)})} + \frac{2cl_\omega}{\lambda}\sqrt{\mathcal{E}^{(z)}\mathcal{E}^{(\nabla)}},$$

$$\mathcal{E}^{(\nabla)} \leq \mathcal{O}(\frac{\log^2 T}{\sqrt{T}}) + \mathcal{O}(\epsilon_{\mathrm{app}}) + u\sqrt{\mathcal{E}^{(\nabla)}(2\mathcal{E}^{(y)} + 8l_v^2\mathcal{E}^{(z)})}.$$

Denote

$$l_1 := 4cu, l_2 := \frac{2u}{\lambda}, l_3 := \frac{2cul_\omega}{\lambda}, l_4 := 8l_v^2, l_5 := \frac{2cl_\omega}{\lambda}, l_6 := u. \tag{37}$$

Then we have

$$\mathcal{E}^{(y)} \leq \mathcal{O}(\frac{\log^2 T}{\sqrt{T}}) + l_1\sqrt{\mathcal{E}^{(y)}\mathcal{E}^{(\nabla)}},$$

$$\mathcal{E}^{(z)} \leq \mathcal{O}(\frac{\log^2 T}{\sqrt{T}}) + \mathcal{O}(\epsilon_{\mathrm{app}}) + l_2\sqrt{\mathcal{E}^{(y)}\mathcal{E}^{(z)}} + l_3\sqrt{\mathcal{E}^{(z)}(2\mathcal{E}^{(y)} + l_4\mathcal{E}^{(z)})} + l_5\sqrt{\mathcal{E}^{(z)}\mathcal{E}^{(\nabla)}},$$

$$\mathcal{E}^{(\nabla)} \leq \mathcal{O}(\frac{\log^2 T}{\sqrt{T}}) + \mathcal{O}(\epsilon_{\mathrm{app}}) + l_6\sqrt{\mathcal{E}^{(\nabla)}(2\mathcal{E}^{(y)} + l_4\mathcal{E}^{(z)})}.$$

For $\mathcal{E}^{(\nabla)}$, we get

$$\mathcal{E}^{(\nabla)} \leq \mathcal{O}(\frac{\log^2 T}{\sqrt{T}}) + \mathcal{O}(\epsilon_{\mathrm{app}}) + \frac{1}{2}\mathcal{E}^{(\nabla)} + l_6^2(\mathcal{E}^{(y)} + \frac{1}{2}l_4\mathcal{E}^{(z)})$$

It follows that

$$\mathcal{E}^{(\nabla)} \leq \mathcal{O}(\frac{\log^2 T}{\sqrt{T}}) + \mathcal{O}(\epsilon_{\mathrm{app}}) + l_6^2(2\mathcal{E}^{(y)} + l_4\mathcal{E}^{(z)}). \tag{38}$$

For $\mathcal{E}^{(z)}$, we have

$$\mathcal{E}^{(z)} \leq \mathcal{O}(\frac{\log^2 T}{\sqrt{T}}) + \mathcal{O}(\epsilon_{\mathrm{app}}) + \frac{1}{4}\mathcal{E}^{(z)} + l_2^2\mathcal{E}^{(y)} + (\frac{1}{2} + \frac{1}{2}l_4)l_3\mathcal{E}^{(z)} + l_3\mathcal{E}^{(y)} + \frac{1}{4}\mathcal{E}^{(z)} + l_5^2\mathcal{E}^{(\nabla)}.$$

If it satisfies $(\frac{1}{2} + \frac{1}{2}l_4)l_3 \leq \frac{1}{4}$, we further have

$$\mathcal{E}^{(z)} \leq \mathcal{O}(\frac{\log^2 T}{\sqrt{T}}) + \mathcal{O}(\epsilon_{\mathrm{app}}) + (2l_2^2 + 2l_3)\mathcal{E}^{(y)} + 2l_5^2\mathcal{E}^{(\nabla)}. \tag{39}$$

Plugging Eq. (38) into Eq. (39), it holds that

$$\mathcal{E}^{(z)} \leq \mathcal{O}(\frac{\log^2 T}{\sqrt{T}}) + \mathcal{O}(\epsilon_{\mathrm{app}}) + (2l_2^2 + 2l_3 + 4l_5^2l_6^2)\mathcal{E}^{(y)} + 2l_4l_5^2l_6^2\mathcal{E}^{(z)}.$$

If it satisfies $2l_4 l_5^2 l_6^2 \leq \frac{1}{2}$, we have

$$\mathcal{E}^{(z)} \leq \mathcal{O}(\frac{\log^2 T}{\sqrt{T}}) + \mathcal{O}(\epsilon_{\mathrm{app}}) + 4(l_2^2 + l_3 + 2l_5^2 l_6^2)\mathcal{E}^{(y)}. \tag{40}$$

For $\mathcal{E}^{(y)}$, we get

$$\mathcal{E}^{(y)} \leq \mathcal{O}(\frac{\log^2 T}{\sqrt{T}}) + \frac{l_1}{2}(\mathcal{E}^{(y)} + \mathcal{E}^{(\nabla)}). \tag{41}$$

Plugging Eq. (38) and Eq. (40) into Eq. (41) gives

$$\mathcal{E}^{(y)} \leq \mathcal{O}(\frac{\log^2 T}{\sqrt{T}}) + \mathcal{O}(\epsilon_{\mathrm{app}}) + \frac{l_1}{2}(\mathcal{E}^{(y)} + 2l_6^2 \mathcal{E}^{(y)} + l_4 l_6^2 \mathcal{E}^{(z)})$$

$$\leq \mathcal{O}(\frac{\log^2 T}{\sqrt{T}}) + \mathcal{O}(\epsilon_{\mathrm{app}}) + \frac{l_1}{2}(\mathcal{E}^{(y)} + 2l_6^2 \mathcal{E}^{(y)} + 4l_4 l_6^2(l_2^2 + l_3 + 2l_5^2 l_6^2)\mathcal{E}^{(y)})$$

$$= \mathcal{O}(\frac{\log^2 T}{\sqrt{T}}) + \mathcal{O}(\epsilon_{\mathrm{app}}) + \frac{l_1}{2}(1 + 2l_6^2 + 4l_4 l_6^2(l_2^2 + l_3 + 2l_5^2 l_6^2))\mathcal{E}^{(y)}.$$

Therefore, if $l_1(1 + 2l_6^2 + 4l_4 l_6^2(l_2^2 + l_3 + 2l_5^2 l_6^2)) \leq \frac{1}{2}$, we have

$$\mathcal{E}^{(y)} \leq \mathcal{O}(\frac{\log^2 T}{\sqrt{T}}) + \mathcal{O}(\epsilon_{\mathrm{app}}).$$

Overall, we require

$$(\frac{1}{2} + \frac{1}{2}l_4)l_3 \leq \frac{1}{4}, \ 2l_4 l_5^2 l_6^2 \leq \frac{1}{2}, \ l_1(1 + 2l_6^2 + 4l_4 l_6^2(l_2^2 + l_3 + 2l_5^2 l_6^2)) \leq \frac{1}{2}.$$

According to the definition of $l_1, l_2, l_3, l_4, l_5, l_6$, it can be shown that

$$\begin{aligned} (\frac{1}{2} + 4l_v^2)\frac{2cul_\omega}{\lambda} &\leq \frac{1}{4} \implies c \leq \frac{\lambda}{4ul_\omega(1 + 8l_v^2)}, \\ \frac{64c^2 u^2 l_v^2 l_\omega^2}{\lambda^2} &\leq \frac{1}{2} \implies c \leq \frac{\lambda}{8\sqrt{2}ul_v l_\omega}, \\ 4cu(1 + 2u^2 + 32l_v^2 u^2(\frac{4u^2}{\lambda^2} + \frac{2cul_\omega}{\lambda} + \frac{8c^2 u^2 l_\omega^2}{\lambda^2})) &\leq \frac{1}{2} \implies c \leq \frac{1}{8u(1 + 6u^2 + \frac{128l_v^2 u^4}{\lambda^2})}. \end{aligned} \tag{42}$$

From the fact that for positive constants $t_i$, we have

$$\frac{1}{\sum_i 1/t_i} \leq \min_i t_i.$$

Thus we choose

$$c \leq \left[ 8u\left(1 + 6u^2 + \frac{128\, l_v^2 u^4}{\lambda^2}\right) + \frac{l_\omega}{\lambda}\left(\frac{1}{2} + 4l_v^2 + \sqrt{2}\, l_v\right) \right]^{-1} \tag{43}$$

which satisfies the three inequalities of $c$ shown in Eq. (42). Therefore, we have

$$\mathcal{E}^{(y)} = \mathcal{O}(\frac{\log^2 T}{\sqrt{T}}) + \mathcal{O}(\epsilon_{\mathrm{app}}),$$

and consequently,

$$\mathcal{E}^{(z)} = \mathcal{O}(\frac{\log^2 T}{\sqrt{T}}) + \mathcal{O}(\epsilon_{\mathrm{app}}),$$

$$\mathcal{E}^{(\nabla)} = \mathcal{O}(\frac{\log^2 T}{\sqrt{T}}) + \mathcal{O}(\epsilon_{\mathrm{app}}).$$

Thus we conclude our proof. $\qquad\qquad\square$

## F   Proof of Preliminary Lemmas

**Proof of Lemma 1.**

*Proof.* We will divide the proof of this lemma into four steps.

**Step 1:** show that for all $k \in \{1, 2, \cdots, K\}$, we have

$$\|\boldsymbol{W}^{(k)}\| \leq \mathcal{O}(\sqrt{m}). \tag{44}$$

It can be shown that

$$\begin{aligned}
\|\boldsymbol{W}^{(k)}\| &\leq \|\boldsymbol{W}^{(k)} - \boldsymbol{W}_0^{(k)}\| + \|\boldsymbol{W}_0^{(K)}\| \\
&\leq u_{\boldsymbol{\omega}} + \|\boldsymbol{W}_0^{(k)}\| \\
&\leq \mathcal{O}(\sqrt{m}),
\end{aligned}$$

where the last inequality id due to Assumption 2 and the fact that $u_{\boldsymbol{\omega}}$ is constant to $m$.

**Step 2:** show that for all $k \in \{1, 2, \cdots, K\}$, we have

$$\|s^{(k)}\| \leq \mathcal{O}(\sqrt{m}). \tag{45}$$

From Assumption 1, we have $\|s^{(0)}\| \leq 1$. From Eq. (44), it holds that

$$\begin{aligned}
\|s^{(1)}\| &= \|\frac{1}{\sqrt{m}}\sigma(\boldsymbol{W}^{(1)}s^{(0)})\| \\
&\leq \frac{1}{m}L_a^2\|\boldsymbol{W}^{(1)}\|^2\|s^{(0)}\|^2 + \|\sigma(0)\|^2 \\
&\leq \mathcal{O}(m).
\end{aligned}$$

By induction, suppose $\|s^{(k)}\|^2 \leq \mathcal{O}(m)$. We have

$$\begin{aligned}
\|s^{(k+1)}\|^2 &= \|\frac{1}{\sqrt{m}}\sigma(\boldsymbol{W}^{(k+1)}s^{(k)})\|^2 \\
&\leq \frac{1}{m}L_a^2\|\boldsymbol{W}^{(k+1)}\|^2\|s^{(k)}\|^2 + \|\sigma(0)\|^2 \\
&\leq \mathcal{O}(m),
\end{aligned}$$

which concludes the proof. Therefore, from Eq. (45), it can be shown that

$$\|\widehat{V}(\boldsymbol{\omega}; s)\| = \|\frac{1}{\sqrt{m}}\boldsymbol{b}^{\top}s^{(K)}\| \leq \mathcal{O}(1).$$

**Step 3:** show that for all $k \in \{1, 2, \cdots, K\}$, we have

$$\|\nabla_{s^{(k-1)}}s^{(k)}\| \leq \mathcal{O}(1). \tag{46}$$

From the chain rule, we have

$$\nabla_{s^{(k-1)}}s^{(k)}(i, j) = \frac{1}{\sqrt{m}}\sigma'(\sum_j \boldsymbol{W}^{(k)}(i, j)s^{(k-1)}(j))\boldsymbol{W}^{(k)}(i, j).$$

Therefore, we get

$$\begin{aligned}
\|\nabla_{s^{(k-1)}}s^{(k)}\|^2 &= \sup_{\|v\|=1}\sum_{i=1}^{m}(\sum_j \nabla_{s^{(k-1)}}s^{(k)}(i, j)v_j)^2 \\
&= \sup_{\|v\|=1}\frac{1}{m}\|\Sigma'\boldsymbol{W}^{(k)}v\|^2 \\
&\leq \frac{1}{m}\|\Sigma'\|^2 \cdot \|\boldsymbol{W}^{(k)}\|^2 \\
&\leq \mathcal{O}(1),
\end{aligned}$$

where $\Sigma'$ is a diagonal matrix with $\Sigma'(i,i) = \sigma'(\Sigma_j \boldsymbol{W}^{(k)}(i,j)s^{(k-1)}(j)) := \xi(i)$.

**Step 4:** show that for all $k \in \{1, 2, \cdots, K\}$, we have

$$\|\nabla_{\boldsymbol{W}^{(k)}} s^{(k)}\| \leq \mathcal{O}(1), \tag{47}$$

where $\nabla_{\boldsymbol{W}^{(k)}} s^{(k)}$ is defined to be a matrix whose $(I, (j-i)m+h)$'th entry $\nabla_{\boldsymbol{W}^{(k)}} s^{(k)}(i,j,h)$ is given by

$$\nabla_{\boldsymbol{W}^{(k)}} s^{(k)}(i,j,h) = \frac{\partial s^{(k)}(i)}{\partial \boldsymbol{W}^{(k)}(j,h)}.$$

It holds that

$$\nabla_{\boldsymbol{W}^{(k)}} s^{(k)}(i,j,j') = \frac{1}{\sqrt{m}} \mathbf{1}\{i-j\}\sigma'(\sum_h \boldsymbol{W}^{(k)}(i,h)s^{(k-1)}(h))s^{(k-1)}(j'),$$

which can be written as

$$\nabla_{\boldsymbol{W}^{(k)}} s^{(k)}(i,j,j') = \frac{1}{\sqrt{m}} \mathbf{1}\{i=j\}\xi(i)s^{(k-1)}(j').$$

Therefore, we get

$$
\begin{aligned}
\|\nabla_{\boldsymbol{W}^{(k)} s^{(k)}}\|^2 &= \sup_{\|V\|_{\mathrm{F}}=1} \sum_{i=1}^{m} (\sum_{j,j'} \nabla_{\boldsymbol{W}^{(k)}} s^{(k)}(i,j,j')V_{j,j'})^2 \\
&= \frac{1}{m} \sup_{\|V\|_{\mathrm{F}}=1} \sum_{i=1}^{m} (\sum_{j,j'} \mathbf{1}\{i=j\}\xi(i)s^{(k-1)}(j')V_{j,j'})^2 \\
&= \frac{1}{m} \sup_{\|V\|_{\mathrm{F}}=1} \sum_{i=1}^{m} (\sum_{j,j'} \mathbf{1}\{i=j\}\xi(i)[Vs^{(k-1)}]_j)^2 \\
&= \frac{1}{m} \sup_{\|V\|_{\mathrm{F}}=1} \sum_{i=1}^{m} \xi(i)^2 [Vs^{(k-1)}]_i^2 \\
&= \sup_{\|V\|_{\mathrm{F}}=1} \frac{1}{m} \|\Sigma' V s^{(k-1)}\|^2 \\
&\leq \frac{1}{m} \|\Sigma'\|^2 \cdot \|s^{(k-1)}\|^2 \\
&\leq \mathcal{O}(1),
\end{aligned}
$$

where the last inequality follows Eq. (45).

We then show the Lipschitzness of the neural network. Since each entry of $b$ satisfies $|b_i| \leq 1$, it is easy to see that

$$\|\nabla_{s^{(K)}} \widehat{V}(\boldsymbol{\omega}; s)\| = \frac{1}{\sqrt{m}} \|\boldsymbol{b}\| \leq 1.$$

By Eq. (46), Eq. (47), and the chain rule, we have

$$\|\nabla_{\boldsymbol{W}^{(k)}} V(\boldsymbol{\omega}; s) = \|\nabla_{\boldsymbol{W}^{(K)}} V(\boldsymbol{\omega}; s)\nabla_{\boldsymbol{W}^{(K-1)}} s^{(K)} \cdots \nabla_{s^{(k)}} s^{(k+1)} \nabla_{\boldsymbol{W}^{(k)}} s^{(k)}\| \leq \mathcal{O}(1).$$

It can be shown that

$$\|\nabla \widehat{V}(\boldsymbol{\omega}; s)\|^2 = \sup_{\|V\|_{\mathrm{F}}=1} \sum_{k=1}^{K} (\nabla_{\boldsymbol{W}^{(k)}} \widehat{V}(\boldsymbol{\omega}; s)V_k)^2 \leq \mathcal{O}(1),$$

which concludes the proof of Lipschitzness.

The proof of smoothness property has been shown in (Liu et al., 2020). □

**Proof of Lemma 2.**

*Proof.* From the definition of the total variation distance, we have

$$
\begin{aligned}
d_{\mathrm{TV}}(\pi_{\boldsymbol{\theta}_1}(\cdot \mid s) - \pi_{\boldsymbol{\theta}_2}(\cdot \mid s)) &= \frac{1}{2} \int_{\mathcal{A}} |\pi_{\boldsymbol{\theta}_1}(a \mid s) - \pi_{\boldsymbol{\theta}_2}(a \mid s)| \, da \\
&= \frac{1}{2} \int_{\bar{\mathcal{A}}} |\pi_{\boldsymbol{\theta}_1}(a \mid s) - \pi_{\boldsymbol{\theta}_2}(a \mid s)| \, da \\
&\leq \frac{1}{2} \int_{\bar{\mathcal{A}}} l_p \|\boldsymbol{\theta}_1 - \boldsymbol{\theta}_2\| \, da \\
&\leq \frac{1}{2} \bar{A} l_p \|\boldsymbol{\theta}_1 - \boldsymbol{\theta}_2\|,
\end{aligned}
$$

where $\bar{\mathcal{A}}$ is the bounded support of $\pi_{\boldsymbol{\theta}}(a \mid s)$ which satisfies $\int_{\bar{\mathcal{A}}} da = \bar{A}$. Define $l_\pi := 1/2 \bar{A} l_p$, which completes the proof. $\qquad\square$

**Proof of Lemma 3.**

*Proof.* For any $\theta_1$ and $\theta_2$, define the transition kernels respectively as follows:

$$
P_i(s, ds') = \int_{\mathcal{A}} \mathcal{P}(ds'|s, a) \pi_{\theta_i}(a|s), \quad i = 1, 2
$$

Following from Theorem 3.1 in (Mitrophanov, 2005), we obtain

$$
d_{TV}(\mu_{\theta_1}, \mu_{\theta_2}) \leq (\lceil \log_\rho \kappa^{-1} \rceil + \frac{1}{1 - \rho}) \|P_1 - P_2\|_{\mathrm{op}},
$$

where $\| \cdot \|_{\mathrm{op}}$ is the operator norm defined in (Mitrophanov, 2005): $\|A\| := \sup_{\|q\|_{\mathrm{TV}}=1} \|qA\|_{\mathrm{TV}}$, and $\| \cdot \|_{\mathrm{TV}}$ denotes the total-variation norm. Then we have

$$
\begin{aligned}
\|P_1 - P_2\|_{\mathrm{op}} &= \sup_{\|q\|_{\mathrm{TV}}=1} \| \int_{\mathcal{S}} q(ds)(P_1 - P_2)(s, \cdot) \|_{\mathrm{TV}} \\
&= \sup_{\|q\|_{\mathrm{TV}}=1} \int_{\mathcal{S}} | \int_{\mathcal{S}} q(ds)(P_1 - P_2)(s, ds')| \\
&\leq \sup_{\|q\|_{\mathrm{TV}}=1} \int_{\mathcal{S}} \int_{\mathcal{S}} q(ds)|(P_1 - P_2)(s, ds')| \\
&= \sup_{\|q\|_{\mathrm{TV}}=1} \int_{\mathcal{S}} \int_{\mathcal{S}} q(ds)| \int_{\mathcal{A}} \mathcal{P}(ds'|s, a)(\pi_{\theta_1}(da|s) - \pi_{\theta_2}(da|s))| \\
&= \sup_{\|q\|_{\mathrm{TV}}=1} \int_{\mathcal{S}} \int_{\mathcal{S}} q(ds) \int_{\mathcal{A}} \mathcal{P}(ds'|s, a)|(\pi_{\theta_1}(da|s) - \pi_{\theta_2}(da|s))| \\
&= \sup_{\|q\|_{\mathrm{TV}}=1} \int_{\mathcal{S}} q(ds) \int_{\mathcal{A}} |(\pi_{\theta_1}(da|s) - \pi_{\theta_2}(da|s))| \\
&\leq l_\pi \|\theta_1 - \theta_2\|.
\end{aligned}
$$

The first equation results from the definition of the operation norm, the second equation results from the definition of total variation. Therefore, we have

$$
d_{TV}(\mu_{\boldsymbol{\theta}_1}, \mu_{\boldsymbol{\theta}_2}) \leq l_\pi (\lceil \log_\rho \kappa^{-1} \rceil + \frac{1}{1 - \rho}) \|\boldsymbol{\theta}_1 - \boldsymbol{\theta}_2\|.
$$

For the second inequality, we have

$$d_{TV}(\mu_{\boldsymbol{\theta}_1} \otimes \pi_{\boldsymbol{\theta}_1}, \mu_{\boldsymbol{\theta}_2} \otimes \pi_{\boldsymbol{\theta}_2}) = \int_{\mathcal{S}} \int_{\mathcal{A}} |\mu_{\theta_1}(ds)\pi_{\theta_1}(a|s) - \mu_{\theta_2}(ds)\pi_{\theta_2}(a|s)|$$

$$\leq \int_{\mathcal{S}} \int_{\mathcal{A}} |\mu_{\theta_1}(ds)(\pi_{\theta_1}(a|s) - \pi_{\theta_2}(a|s))|$$

$$+ \int_{\mathcal{S}} \int_{\mathcal{A}} |(\mu_{\theta_1}(ds) - \mu_{\theta_2}(ds))\pi_{\theta_2}(a|s))|$$

$$= d_{TV}(\pi_{\theta_1}, \pi_{\theta_2}) + d_{TV}(\mu_{\theta_1}, \mu_{\theta_2})$$

$$\leq l_\pi \|\theta_1 - \theta_2\| + C(\lceil \log_\rho \kappa^{-1} \rceil + \frac{1}{1-\rho})\|\boldsymbol{\theta}_1 - \boldsymbol{\theta}_2\|$$

$$= l_\pi(1 + \lceil \log_\rho \kappa^{-1} \rceil + \frac{1}{1-\rho})\|\boldsymbol{\theta}_1 - \boldsymbol{\theta}_2\|.$$

For the third inequality, we have

$$d_{TV}(\mu_{\boldsymbol{\theta}_1} \otimes \pi_{\boldsymbol{\theta}_1} \otimes \mathcal{P}, \mu_{\boldsymbol{\theta}_2} \otimes \pi_{\boldsymbol{\theta}_2} \otimes \mathcal{P})$$

$$= \frac{1}{2} \int_S \int_{\mathcal{A}} \int_S |\mu_{\theta_1}(ds)\pi_{\theta_1}(a|s)\mathcal{P}(ds'|s,a) - \mu_{\theta_2}(ds)\pi_{\theta_2}(a|s)\mathcal{P}(ds'|s,a)|$$

$$= \frac{1}{2} \int_S \int_{\mathcal{A}} |\mu_{\theta_1}(ds)\pi_{\theta_1}(a|s) - \mu_{\theta_2}(ds)\pi_{\theta_2}(a|s)|$$

$$= d_{TV}(\mu_{\boldsymbol{\theta}_1} \otimes \pi_{\boldsymbol{\theta}_1}, \mu_{\boldsymbol{\theta}_2} \otimes \pi_{\boldsymbol{\theta}_2}),$$

which concludes the proof. $\qquad\square$

**Proof of Lemma 4.**

*Proof.* From the fact that

$$\mathbb{P}(s_{t+1} \in \cdot) = \int_{\mathcal{S}} \int_{\mathcal{A}} \mathbb{P}(s_t = ds, a_t = da, s_{t+1} \in \cdot),$$

we have

$$2d_{TV}(\mathbb{P}(s_{t+1} \in \cdot), \mathbb{P}(\tilde{s}_{t+1} \in \cdot))$$

$$= \int_{\mathcal{S}} |\int_{\mathcal{S}} \int_{\mathcal{A}} \mathbb{P}(s_t = ds, a_t = da, s_{t+1} = ds') - \int_{\mathcal{S}} \int_{\mathcal{A}} \mathbb{P}(\tilde{s}_t = ds, \tilde{a}_t = da, \tilde{s}_{t+1} = ds')|$$

$$\leq \int_{\mathcal{S}} \int_{\mathcal{S}} \int_{\mathcal{A}} |\mathbb{P}(s_t = ds, a_t = da, s_{t+1} = ds') - \mathbb{P}(\tilde{s}_t = ds, \tilde{a}_t = da, \tilde{s}_{t+1} = ds')|$$

$$= \int_{\mathcal{S}} \int_{\mathcal{S}} \int_{\mathcal{A}} |\mathbb{P}(O_t = (ds, da, ds')) - \mathbb{P}(\tilde{O}_t = (ds, da, ds'))|$$

$$= 2d_{TV}(\mathbb{P}(O_t \in \cdot), \mathbb{P}(\tilde{O} \in \cdot)),$$

where the last equality requires the exchange of integral which is guaranteed by Fubini's theorem since $\mathbb{P}$ is an absolute integrable function.

For the second equality, we have

$$2d_{TV}(\mathbb{P}(O_t \in \cdot), \mathbb{P}(\tilde{O}_t \in \cdot))$$

$$= \int_{\mathcal{S}} \int_{\mathcal{A}} \int_{\mathcal{S}} |\mathbb{P}(O_t = (ds, da, ds')) - \mathbb{P}(\tilde{O}_t = (ds, da, ds'))|$$

$$= \int_{\mathcal{S}} \int_{\mathcal{A}} \int_{\mathcal{S}} |\mathcal{P}(ds'|s,a)\mathbb{P}((s_t, a_t) = (ds, da)) - \mathcal{P}(ds'|s,a)\mathbb{P}((\tilde{s}_t, \tilde{a}_t) = (ds, da))|$$

$$= \int_{\mathcal{S}} \int_{\mathcal{A}} \int_{\mathcal{S}} \mathcal{P}(ds'|s,a)|\mathbb{P}((s_t, a_t) = (ds, da)) - \mathbb{P}((\tilde{s}_t, \tilde{a}_t) = (ds, da))|$$

$$= \int_{\mathcal{S}} \int_{\mathcal{A}} |\mathbb{P}((s_t, a_t) = (ds, da)) - \mathbb{P}((\tilde{s}_t, \tilde{a}_t) = (ds, da))|$$

$$= 2d_{TV}(\mathbb{P}((s_t, a_t) \in \cdot), \mathbb{P}((\tilde{s}_t, \tilde{a}_t) \in \cdot)).$$

For the third inequality, since $\boldsymbol{\theta}_t$ is dependent on $s_t$ as shown in Eq. (10), it holds that

$$
\begin{aligned}
& 2d_{TV}\left(\mathbb{P}((s_t, a_t) \in \cdot), \mathbb{P}((\tilde{s}_t, \tilde{a}_t) \in \cdot)\right) \\
&= \int_{\mathcal{S}} \int_{\mathcal{A}} |\mathbb{P}(s_t = ds, a_t = da) - \mathbb{P}(\tilde{s}_t = ds, \tilde{a}_t = da)| \\
&= \int_{\mathcal{S}} \int_{\mathcal{A}} | \int_{\boldsymbol{\theta}} \mathbb{P}(s_t = ds)\mathbb{P}(\boldsymbol{\theta}_t = d\boldsymbol{\theta}|s_t = s)\mathbb{P}(a_t = da|s_t = s, \boldsymbol{\theta}_t = \boldsymbol{\theta}) - \mathbb{P}(\tilde{s}_t = ds, \tilde{a}_t = da)| \\
&= \int_{\mathcal{S}} \int_{\mathcal{A}} |\mathbb{P}(s_t = ds) \int_{\boldsymbol{\theta}} \mathbb{P}(\boldsymbol{\theta}_t = d\boldsymbol{\theta}|s_t = s)\pi_{\boldsymbol{\theta}_t}(da|s) - \mathbb{P}(\tilde{s}_t = ds)\pi_{\boldsymbol{\theta}_{t-\tau}}(da|s)| \\
&= \int_{\mathcal{S}} \int_{\mathcal{A}} |\mathbb{P}(s_t = ds)\mathbb{E}[\pi_{\boldsymbol{\theta}_t}(da|s)|s_t = s] - \mathbb{P}(\tilde{s}_t = ds)\pi_{\boldsymbol{\theta}_{t-\tau}}(da|s)| \\
&= \int_{\mathcal{S}} \int_{\mathcal{A}} |\mathbb{P}(s_t = ds)\mathbb{E}[\pi_{\boldsymbol{\theta}_t}(da|s)|s_t = s] - \mathbb{P}(s_t = ds)\pi_{\boldsymbol{\theta}_{t-\tau}}(da|s)| \\
&\quad + \int_{\mathcal{S}} \int_{\mathcal{A}} |\mathbb{P}(s_t = ds)\pi_{\boldsymbol{\theta}_{t-\tau}}(da|s) - \mathbb{P}(\tilde{s}_t = ds)\pi_{\boldsymbol{\theta}_{t-\tau}}(da|s)| \\
&= \int_{\mathcal{S}} \mathbb{P}(s_t = ds) \int_{\mathcal{A}} |\mathbb{E}[\pi_{\boldsymbol{\theta}_t}(da|s)|s_t = s] - \pi_{\boldsymbol{\theta}_{t-\tau}}(da|s)| \\
&\quad + 2d_{TV}(\mathbb{P}(s_t \in \cdot), \mathbb{P}(\tilde{s}_t \in \cdot)) \\
&\leq l_\pi \mathbb{E}\|\boldsymbol{\theta}_t - \boldsymbol{\theta}_{t-\tau}\| + 2d_{TV}(\mathbb{P}(s_t \in \cdot), \mathbb{P}(\tilde{s}_t \in \cdot)),
\end{aligned}
$$

where the last inequality holds due to the Lipschitz continuity of policy made in Assumption 7. $\quad\square$

**Proof of Lemma 5.**

*Proof.* By definition, we have

$$
J(\theta_1) - J(\theta_2) = \mathbb{E}[r(s^1, a^1) - r(s^2, a^2)],
$$

where $s^i \sim \mu_{\boldsymbol{\theta}_i}, a^i \sim \pi_{\boldsymbol{\theta}_i}$. Therefore, it holds that

$$
\begin{aligned}
J(\boldsymbol{\theta}_1) - J(\boldsymbol{\theta}_2) &= \mathbb{E}[r(s^1, a^1) - r(s^1, a^1)] \\
&\leq 2u d_{TV}(\mu_{\boldsymbol{\theta}_1} \otimes \pi_{\boldsymbol{\theta}_1}, \mu_{\boldsymbol{\theta}_2} \otimes \pi_{\boldsymbol{\theta}_2}) \\
&\leq 2u l_\pi (1 + \lceil \log_\rho \kappa^{-1} \rceil + \frac{1}{1-\rho})\|\boldsymbol{\theta}_1 - \boldsymbol{\theta}_2\| \\
&= l_j \|\boldsymbol{\theta}_1 - \boldsymbol{\theta}_2\|.
\end{aligned}
$$

$\square$

**Proof of Lemma 6.**

*Proof.* The proof of this lemma can be found in Lemma 3.2 of (Zhang et al., 2020a). $\quad\square$

## G   PROOF OF MARKOVIAN NOISES

The following four lemmas deal with the Markovian noise.

**Proof of Lemma 7.**

*Proof.* We will divide the proof of this lemma into four steps.

**Step 1:** show that for any $\boldsymbol{\theta}_1, \boldsymbol{\theta}_2, \eta, O = (s, a, s')$, we have

$$
|\Phi(O, \eta, \boldsymbol{\theta}_1) - \Phi(O, \eta, \boldsymbol{\theta}_2)| \leq 4u l_j \|\boldsymbol{\theta}_1 - \boldsymbol{\theta}_2\|. \tag{48}
$$

By the definition of $\Phi(O, \eta, \boldsymbol{\theta})$ in Eq. (13), we have

$$
\begin{aligned}
|\Phi(O, \eta, \boldsymbol{\theta}_1) - \Phi(O, \boldsymbol{\theta}, \boldsymbol{\theta}_2)| &= |(\eta - J(\boldsymbol{\theta}_1))(r - J(\boldsymbol{\theta}_1)) - (\eta - J(\boldsymbol{\theta}_2))(r - J(\boldsymbol{\theta}_2))| \\
&\leq |(\eta - J(\boldsymbol{\theta}_1))(r - J(\boldsymbol{\theta}_1)) - (\eta - J(\boldsymbol{\theta}_1))(r - J(\boldsymbol{\theta}_2))| \\
&\quad + |(\eta - J(\boldsymbol{\theta}_1))(r - J(\boldsymbol{\theta}_2)) - (\eta - J(\boldsymbol{\theta}_2))(r - J(\boldsymbol{\theta}_2))| \\
&\leq 4u|J(\boldsymbol{\theta}_1) - J(\boldsymbol{\theta}_2)| \\
&\leq 4u l_j \|\boldsymbol{\theta}_1 - \boldsymbol{\theta}_2\|.
\end{aligned}
$$

**Step 2:** show that for any $\boldsymbol{\theta}, \eta_1, \eta_2, O$, we have

$$
|\Phi(O, \eta_1, \boldsymbol{\theta}) - \Phi(O, \eta_2, \boldsymbol{\theta}) \leq 2u|\eta_1 - \eta_2|. \tag{49}
$$

By definition, we have

$$
\begin{aligned}
|\Phi(O, \eta_1, \boldsymbol{\theta}) - \Phi(O, \eta_2, \boldsymbol{\theta})| &= |(\eta_1 - J(\boldsymbol{\theta}))(r - J(\boldsymbol{\theta})) - (\eta_2 - J(\boldsymbol{\theta}))(r - J(\boldsymbol{\theta}))| \\
&\leq 2u|\eta_1 - \eta_2|.
\end{aligned}
$$

**Step 3:** show that for original tuple $O_t$ and the auxiliary tuple $\widetilde{O}_t$, conditioned on $s_{t-\tau+1}$ and $\boldsymbol{\theta}_{t-\tau}$, we have

$$
|\mathbb{E}[\Phi(O_t, \eta_{t-\tau}, \boldsymbol{\theta}_{t-\tau}) - \mathbb{E}[\Phi(\widetilde{O}_t, \eta_{t-\tau}, \boldsymbol{\theta}_{t-\tau})]| \leq u l_\pi \sum_{k=t-\tau}^{t} \mathbb{E}\|\boldsymbol{\theta}_k - \boldsymbol{\theta}_{t-\tau}\|. \tag{50}
$$

By definition, we have

$$
\mathbb{E}[\Phi(O_t, \eta_{t-\tau}, \boldsymbol{\theta}_{t-\tau}) - \mathbb{E}[\Phi(\widetilde{O}_t, \eta_{t-\tau}, \boldsymbol{\theta}_{t-\tau})] = (\eta_{t-\tau} - J(\boldsymbol{\theta}_{t-\tau}))\mathbb{E}[r(s_t, a_t) - r(\widetilde{s}_t, \widetilde{a}_t)].
$$

By definition of total variation norm, we have

$$
\mathbb{E}[r(s_t, a_t) - r(\widetilde{s}_t, \widetilde{a}_t)] \leq 2u d_{TV}(\mathbb{P}(O_t \in \cdot | s_{t-\tau+1}, \boldsymbol{\theta}_{t-\tau}), \mathbb{P}(\widetilde{O}_t \in \cdot | s_{t-\tau+1}, \boldsymbol{\theta}_{t-\tau})). \tag{51}
$$

By Lemma 4, we get

$$
\begin{aligned}
&d_{TV}(\mathbb{P}(O_t \in \cdot | s_{t-\tau+1}, \boldsymbol{\theta}_{t-\tau}), \mathbb{P}(\widetilde{O}_t \in \cdot | s_{t-\tau+1}, \boldsymbol{\theta}_{t-\tau})) \\
&= d_{TV}(\mathbb{P}((s_t, a_t) \in \cdot | s_{t-\tau+1}, \boldsymbol{\theta}_{t-\tau}), \mathbb{P}((\widetilde{s}_t, \widetilde{a}_t) \in \cdot | s_{t-\tau+1}, \boldsymbol{\theta}_{t-\tau})) \\
&\leq d_{TV}(\mathbb{P}(s_t \in \cdot | s_{t-\tau+1}, \boldsymbol{\theta}_{t-\tau}), \mathbb{P}(\widetilde{s}_t \in \cdot | s_{t-\tau+1}, \boldsymbol{\theta}_{t-\tau})) + \frac{1}{2} l_\pi \mathbb{E}\|\boldsymbol{\theta}_t - \boldsymbol{\theta}_{t-\tau}\| \\
&\leq d_{TV}(\mathbb{P}(O_{t-1} \in \cdot | s_{t-\tau+1}, \boldsymbol{\theta}_{t-\tau}), \mathbb{P}(\widetilde{O}_{t-1} \in \cdot | s_{t-\tau+1}, \boldsymbol{\theta}_{t-\tau})) + \frac{1}{2} l_\pi \mathbb{E}\|\boldsymbol{\theta}_t - \boldsymbol{\theta}_{t-\tau}\|.
\end{aligned}
$$

Repeat the above argument from $t$ to $t - \tau$, we have

$$
d_{TV}(\mathbb{P}(O_t \in \cdot | s_{t-\tau+1}, \boldsymbol{\theta}_{t-\tau}), \mathbb{P}(\widetilde{O}_t \in \cdot | s_{t-\tau+1}, \boldsymbol{\theta}_{t-\tau})) \leq \frac{1}{2} l_\pi \sum_{k=t-\tau}^{t} \mathbb{E}\|\boldsymbol{\theta}_k - \boldsymbol{\theta}_{t-\tau}\|. \tag{52}
$$

Plugging Eq. (52) into Eq. (51), we have

$$
|\mathbb{E}[\Phi(O_t, \eta_{t-\tau}, \boldsymbol{\theta}_{t-\tau}) - \mathbb{E}[\Phi(\widetilde{O}_t, \eta_{t-\tau}, \boldsymbol{\theta}_{t-\tau})]| \leq u l_\pi \sum_{k=t-\tau}^{t} \mathbb{E}\|\boldsymbol{\theta}_k - \boldsymbol{\theta}_{t-\tau}\|.
$$

**Step 4:** show that conditioned on $s_{t-\tau+1}$ and $\boldsymbol{\theta}_{t-\tau}$, we have

$$
\mathbb{E}[\Phi(\widetilde{O}_t, \eta_{t-\tau}, \boldsymbol{\theta}_{t-\tau})] \leq 4u^2 \kappa \rho^{\tau-1}. \tag{53}
$$

Note that according to definition, we have

$$
\mathbb{E}[\Phi(O'_{t-\tau}, \eta_{t-\tau}, \boldsymbol{\theta}_{t-\tau}) | \boldsymbol{\theta}_{t-\tau}] = 0,
$$

where $O'_{t-\tau} = (s'_{t-\tau}, a'_{t-\tau}, s'_{t-\tau+1})$ is the tuple generated by $s'_{t-\tau} \sim \mu_{\boldsymbol{\theta}_{t-\tau}}, a'_{t-\tau} \sim \pi_{\boldsymbol{\theta}_{t-\tau}}, s'_{t-\tau+1} \sim \mathcal{P}$. From the uniform ergodicity in Assumption 6, it shows that

$$
d_{TV}(\mathbb{P}(\widetilde{s}_t = \cdot | s_{t-\tau+1}, \boldsymbol{\theta}_{t-\tau}), \mu_{\boldsymbol{\theta}_{t-\tau}}) \leq \kappa \rho^{\tau-1}.
$$

Then we have

$$
\begin{aligned}
\mathbb{E}[\Phi(\widetilde{O}_t, \eta_{t-\tau}, \boldsymbol{\theta}_{t-\tau})] &= \mathbb{E}[\Phi(\widetilde{O}_t, \eta_{t-\tau}, \boldsymbol{\theta}_{t-\tau}) - \Phi(O'_{t-\tau}, \eta_{t-\tau}, \boldsymbol{\theta}_{t-\tau})] \\
&= \mathbb{E}[(\eta_{t-\tau} - J(\boldsymbol{\theta}_{t-\tau}))(r(\tilde{s}_t, \tilde{a}_t) - r(s'_{t-\tau}, a'_{t-\tau}))] \\
&\leq 4u^2 d_{TV}(\mathbb{P}(\widetilde{O}_{t-\tau} = \cdot | s_{t-\tau+1}, \boldsymbol{\theta}_{t-\tau}), \mu_{\boldsymbol{\theta}_{t-\tau}} \otimes \pi_{\boldsymbol{\theta}_{t-\tau}} \otimes \mathcal{P}) \\
&\leq 4u^2 \kappa \rho^{\tau-1}.
\end{aligned}
$$

Combining Eq. (48), Eq. (49), Eq. (50), and Eq. (53), we have

$$
\begin{aligned}
\mathbb{E}[\Phi(O_t, \eta_t, \boldsymbol{\theta}_t)] &= \mathbb{E}[\Phi(O_t, \eta_t, \boldsymbol{\theta}_t) - \Phi(O_t, \eta_t, \boldsymbol{\theta}_{t-\tau})] + \mathbb{E}[\Phi(O_t, \eta_t, \boldsymbol{\theta}_{t-\tau}) - \Phi(O_t, \eta_{t-\tau}, \boldsymbol{\theta}_{t-\tau})] \\
&\quad + \mathbb{E}[\Phi(O_t, \eta_{t-\tau}, \boldsymbol{\theta}_{t-\tau}) - \Phi(\widetilde{O}_t, \eta_{t-\tau}, \boldsymbol{\theta}_{t-\tau})] + \mathbb{E}[\Phi(\widetilde{O}_t, \eta_{t-\tau}, \boldsymbol{\theta}_{t-\tau})] \\
&\leq 4u l_j \mathbb{E}\|\boldsymbol{\theta}_t - \boldsymbol{\theta}_{t-\tau}\| + 2u\mathbb{E}|\eta_t - \eta_{t-\tau}| + u l_\pi \sum_{i=t-\tau}^{t} \mathbb{E}\|\boldsymbol{\theta}_i - \boldsymbol{\theta}_{t-\tau}\| + 4u^2 \kappa \rho^{\tau-1} \\
&\leq 16u^2 \tau \alpha l_j + 4u^2 \tau \gamma + 4u^2 \tau (\tau+1) \alpha l_\pi + 4u^2 \kappa \rho^{\tau-1}.
\end{aligned}
$$

which concludes the proof. $\qquad\square$

**Proof of Lemma 8.**

*Proof.* We will divide the proof of this lemma into four steps.

**Step 1:** show that for any $\boldsymbol{\theta}_1, \boldsymbol{\theta}_2, \boldsymbol{\omega}$ and tuple $O = (s, a, s')$, we have

$$
|\Psi(O, \boldsymbol{\omega}, \boldsymbol{\theta}_1) - \Psi(O, \boldsymbol{\omega}, \boldsymbol{\theta}_2) \leq c_1 \|\boldsymbol{\theta}_1 - \boldsymbol{\theta}_2\|, \tag{54}
$$

where $c_1 = 4u^2 l_\pi (1 + \lceil \log_\rho \kappa^{-1} \rceil + \frac{1}{1-\rho}) + 2u l_j l_v + 4u l_\omega l_v$.

By definition of $\Psi(O, \boldsymbol{\omega}, \boldsymbol{\theta})$ in Eq. (13), we have

$$
\begin{aligned}
&|\Psi(O, \boldsymbol{\omega}, \boldsymbol{\theta}_1) - \Psi(O, \boldsymbol{\omega}, \boldsymbol{\theta}_2)| \\
&= |\langle \boldsymbol{\omega} - \boldsymbol{\omega}_1^*, g(O, \boldsymbol{\omega}, \boldsymbol{\theta}_1) - \bar{g}(\boldsymbol{\omega}, \boldsymbol{\theta}_1) \rangle - \langle \boldsymbol{\omega} - \boldsymbol{\omega}_2^*, g(O, \boldsymbol{\omega}, \boldsymbol{\theta}_2) - \bar{g}(\boldsymbol{\omega}, \boldsymbol{\theta}_2) \rangle| \\
&\leq \underbrace{|\langle \boldsymbol{\omega} - \boldsymbol{\omega}_1^*, g(O, \boldsymbol{\omega}, \boldsymbol{\theta}_1) - \bar{g}(\boldsymbol{\omega}, \boldsymbol{\theta}_1) \rangle - \langle \boldsymbol{\omega} - \boldsymbol{\omega}_1^*, g(O, \boldsymbol{\omega}, \boldsymbol{\theta}_2) - \bar{g}(\boldsymbol{\omega}, \boldsymbol{\theta}_2) \rangle|}_{I_1} \\
&\quad + \underbrace{|\langle \boldsymbol{\omega} - \boldsymbol{\omega}_1^*, g(O, \boldsymbol{\omega}, \boldsymbol{\theta}_2) - \bar{g}(\boldsymbol{\omega}, \boldsymbol{\theta}_2) \rangle - \langle \boldsymbol{\omega} - \boldsymbol{\omega}_2^*, g(O, \boldsymbol{\omega}, \boldsymbol{\theta}_2) - \bar{g}(\boldsymbol{\omega}, \boldsymbol{\theta}_2) \rangle|}_{I_2}.
\end{aligned}
$$

For term $I_1$, we have

$$
\begin{aligned}
I_1 &= |\langle \boldsymbol{\omega} - \boldsymbol{\omega}_1^*, g(O, \boldsymbol{\omega}, \boldsymbol{\theta}_1) - \bar{g}(\boldsymbol{\omega}, \boldsymbol{\theta}_1) \rangle - \langle \boldsymbol{\omega} - \boldsymbol{\omega}_1^*, g(O, \boldsymbol{\omega}, \boldsymbol{\theta}_2) - \bar{g}(\boldsymbol{\omega}, \boldsymbol{\theta}_2) \rangle| \\
&= |\langle \boldsymbol{\omega} - \boldsymbol{\omega}_1^*, g(O, \boldsymbol{\omega}, \boldsymbol{\theta}_1) - g(O, \boldsymbol{\omega}, \boldsymbol{\theta}_2) \rangle| + |\langle \boldsymbol{\omega} - \boldsymbol{\omega}_1^*, \bar{g}(\boldsymbol{\omega}, \boldsymbol{\theta}_1) - \bar{g}(\boldsymbol{\omega}, \boldsymbol{\theta}_2) \rangle| \\
&= |\langle \boldsymbol{\omega} - \boldsymbol{\omega}_1^*, (J(\boldsymbol{\theta}_1) - J(\boldsymbol{\theta}_2))\nabla\widehat{V}(\boldsymbol{\omega}; s) \rangle| + |\langle \boldsymbol{\omega} - \boldsymbol{\omega}_1^*, \bar{g}(\boldsymbol{\omega}, \boldsymbol{\theta}_1) - \bar{g}(\boldsymbol{\omega}, \boldsymbol{\theta}_2) \rangle| \\
&\leq 2u l_j l_v \|\boldsymbol{\theta}_1 - \boldsymbol{\theta}_2\| + 2u\|\bar{g}(\boldsymbol{\omega}, \boldsymbol{\theta}_1) - \bar{g}(\boldsymbol{\omega}, \boldsymbol{\theta}_2)\| \\
&\leq 2u l_j l_v \|\boldsymbol{\theta}_1 - \boldsymbol{\theta}_2\| + 4u^2 d_{TV}(\mu_{\boldsymbol{\theta}_1} \otimes \pi_{\boldsymbol{\theta}_1} \otimes \mathcal{P}, \mu_{\boldsymbol{\theta}_2} \otimes \pi_{\boldsymbol{\theta}_2} \otimes \mathcal{P}) \\
&\leq (2u l_j l_v + 4u^2 l_\pi (1 + \lceil \log_\rho \kappa^{-1} \rceil + \frac{1}{1-\rho}))\|\boldsymbol{\theta}_1 - \boldsymbol{\theta}_2\|.
\end{aligned}
$$

For term $I_2$, from Cauchy-Schwartz inequality, we have

$$
\begin{aligned}
I_2 &= |\langle \boldsymbol{\omega} - \boldsymbol{\omega}_1^*, g(O, \boldsymbol{\omega}, \boldsymbol{\theta}_2) - \bar{g}(\boldsymbol{\omega}, \boldsymbol{\theta}_2) \rangle - \langle \boldsymbol{\omega} - \boldsymbol{\omega}_2^*, g(O, \boldsymbol{\omega}, \boldsymbol{\theta}_2) - \bar{g}(\boldsymbol{\omega}, \boldsymbol{\theta}_2) \rangle| \\
&= |\langle \boldsymbol{\omega}_1^* - \boldsymbol{\omega}_2^*, g(O, \boldsymbol{\omega}, \boldsymbol{\theta}_2) - \bar{g}(\boldsymbol{\omega}, \boldsymbol{\theta}_2) \rangle| \\
&\leq 4u l_v \|\boldsymbol{\omega}_1^* - \boldsymbol{\omega}_2^*\| \\
&\leq 4u l_v l_\omega \|\boldsymbol{\theta}_1 - \boldsymbol{\theta}_2\|.
\end{aligned}
$$

Combining the results from $I_1$ and $I_2$, we get

$$
|\Psi(O, \boldsymbol{\omega}, \boldsymbol{\theta}_1) - \Psi(O, \boldsymbol{\omega}, \boldsymbol{\theta}_2) \leq c_1 \|\boldsymbol{\theta}_1 - \boldsymbol{\theta}_2\|,
$$

where $c_1 = 4u^2 l_\pi (1 + \lceil \log_\rho \kappa^{-1} \rceil + \frac{1}{1-\rho}) + 2u l_j l_v + 4u l_\omega l_v$.

**Step 2:** show that for any $\boldsymbol{\theta}, \boldsymbol{\omega}_1, \boldsymbol{\omega}_2$ and tuple $O(s, a, s')$, we have

$$|\Psi(O, \boldsymbol{\omega}_1, \boldsymbol{\theta}) - \Psi(O, \boldsymbol{\omega}_2, \boldsymbol{\theta})| \le c_2 \|\boldsymbol{\omega}_1 - \boldsymbol{\omega}_2\|, \tag{55}$$

where $c_2 = 2u(8uh_v + 4l_v^2 + 2l_v)$.

By definition, we have

$$
\begin{aligned}
&|\Psi(O, \boldsymbol{\omega}_1, \boldsymbol{\theta}) - \Psi(O, \boldsymbol{\omega}_2, \boldsymbol{\theta})| \\
={} & |\langle \boldsymbol{\omega}_1 - \boldsymbol{\omega}^*, g(O, \boldsymbol{\omega}_1, \boldsymbol{\theta}) - \bar{g}(\boldsymbol{\omega}_1, \boldsymbol{\theta}) \rangle - \langle \boldsymbol{\omega}_2 - \boldsymbol{\omega}^*, g(O, \boldsymbol{\omega}_2, \boldsymbol{\theta}) - \bar{g}(\boldsymbol{\omega}_2, \boldsymbol{\theta}) \rangle| \\
\le{} & |\langle \boldsymbol{\omega}_1 - \boldsymbol{\omega}^*, g(O, \boldsymbol{\omega}_1, \boldsymbol{\theta}) - \bar{g}(\boldsymbol{\omega}_1, \boldsymbol{\theta}) \rangle - \langle \boldsymbol{\omega}_1 - \boldsymbol{\omega}^*, g(O, \boldsymbol{\omega}_2, \boldsymbol{\theta}) - \bar{g}(\boldsymbol{\omega}_2, \boldsymbol{\theta}) \rangle| \\
& + |\langle \boldsymbol{\omega}_1 - \boldsymbol{\omega}^*, g(O, \boldsymbol{\omega}_2, \boldsymbol{\theta}) - \bar{g}(\boldsymbol{\omega}_2, \boldsymbol{\theta}) \rangle - \langle \boldsymbol{\omega}_2 - \boldsymbol{\omega}^*, g(O, \boldsymbol{\omega}_2, \boldsymbol{\theta}) - \bar{g}(\boldsymbol{\omega}_2, \boldsymbol{\theta}) \rangle| \\
\le{} & 2u \|(g(O, \boldsymbol{\omega}_1, \boldsymbol{\theta}) - g(O, \boldsymbol{\omega}_2, \boldsymbol{\theta})) - (\bar{g}(\boldsymbol{\omega}_1, \boldsymbol{\theta}) - \bar{g}(\boldsymbol{\omega}_2, \boldsymbol{\theta}))\| + 4u l_v \|\boldsymbol{\omega}_1 - \boldsymbol{\omega}_2\|.
\end{aligned}
$$

It holds that

$$
\begin{aligned}
\|(g(O, \boldsymbol{\omega}_1, \boldsymbol{\theta}) - g(O, \boldsymbol{\omega}_2, \boldsymbol{\theta}))\| ={} & \|(r(s,a) - J(\boldsymbol{\theta}))(\nabla \widehat{V}(\boldsymbol{\omega}_1; s) - \nabla \widehat{V}(\boldsymbol{\omega}_2; s)) \\
& + \widehat{V}(\boldsymbol{\omega}_1; s') \nabla \widehat{V}(\boldsymbol{\omega}_1; s) - \widehat{V}(\boldsymbol{\omega}_2; s') \nabla \widehat{V}(\boldsymbol{\omega}_2; s) \\
& + \widehat{V}(\boldsymbol{\omega}_2; s) \nabla \widehat{V}(\boldsymbol{\omega}_2; s) - \widehat{V}(\boldsymbol{\omega}_1; s) \nabla \widehat{V}(\boldsymbol{\omega}_1; s)\| \\
\le{} & \|\widehat{V}(\boldsymbol{\omega}_1; s') \nabla \widehat{V}(\boldsymbol{\omega}_1; s) - \widehat{V}(\boldsymbol{\omega}_1; s') \nabla \widehat{V}(\boldsymbol{\omega}_2; s) \\
& + \widehat{V}(\boldsymbol{\omega}_1; s') \nabla \widehat{V}(\boldsymbol{\omega}_2; s) - \widehat{V}(\boldsymbol{\omega}_2; s') \nabla \widehat{V}(\boldsymbol{\omega}_2; s)\| \\
& + \|\widehat{V}(\boldsymbol{\omega}_2; s) \nabla \widehat{V}(\boldsymbol{\omega}_2; s) - \widehat{V}(\boldsymbol{\omega}_1; s) \nabla \widehat{V}(\boldsymbol{\omega}_2; s) \\
& + \widehat{V}(\boldsymbol{\omega}_1; s) \nabla \widehat{V}(\boldsymbol{\omega}_2; s) - \widehat{V}(\boldsymbol{\omega}_1; s) \nabla \widehat{V}(\boldsymbol{\omega}_1; s)\| \\
& + 2u h_v \|\boldsymbol{\omega}_1 - \boldsymbol{\omega}_2\| \\
\le{} & 2u h_v \|\boldsymbol{\omega}_1 - \boldsymbol{\omega}_2\| + 2l_v^2 \|\boldsymbol{\omega}_1 - \boldsymbol{\omega}_2\| + 2u h_v \|\boldsymbol{\omega}_1 - \boldsymbol{\omega}_2\| \\
={} & (4u h_v + 2l_v^2) \|\boldsymbol{\omega}_1 - \boldsymbol{\omega}_2\|.
\end{aligned}
$$

It follows that

$$\mathbb{E}\|(g(O, \boldsymbol{\omega}_1, \boldsymbol{\theta}) - g(O, \boldsymbol{\omega}_2, \boldsymbol{\theta})) - (\bar{g}(\boldsymbol{\omega}_1, \boldsymbol{\theta}) - \bar{g}(\boldsymbol{\omega}_2, \boldsymbol{\theta}))\| \le (8u h_v + 4l_v^2)\mathbb{E}\|\boldsymbol{\omega}_1 - \boldsymbol{\omega}_2\|.$$

Therefore, we obtain

$$\mathbb{E}|\Psi(O, \boldsymbol{\omega}_1, \boldsymbol{\theta}) - \Psi(O, \boldsymbol{\omega}_2, \boldsymbol{\theta})| \le c_2 \|\boldsymbol{\omega}_1 - \boldsymbol{\omega}_2\|,$$

where $c_2 = 2u(8u h_v + 4l_v^2 + 2l_v)$.

**Step 3:** show that for tuples $O_t = (s_t, a_t, s_{t+1})$ and $\widetilde{O}_t = (\widetilde{s}_t, \widetilde{a}_t, \widetilde{s}_{t+1})$. Conditioning on $s_{t-\tau+1}$ and $\boldsymbol{\theta}_{t-\tau}$, we have

$$\mathbb{E}[\Psi(O_t, \boldsymbol{\omega}_{t-\tau}, \boldsymbol{\theta}_{t-\tau}) - \Psi(\widetilde{O}_t, \boldsymbol{\omega}_{t-\tau}, \boldsymbol{\theta}_{t-\tau})] \le 16 u^4 l_v l_\pi \tau(\tau+1)\alpha. \tag{56}$$

By the definition of total variation norm, we have

$$
\begin{aligned}
& \mathbb{E}[\Psi(O_t, \boldsymbol{\omega}_{t-\tau}, \boldsymbol{\theta}_{t-\tau}) - \Psi(\widetilde{O}_t, \boldsymbol{\omega}_{t-\tau}, \boldsymbol{\theta}_{t-\tau})] \\
={} & \mathbb{E}[\langle \boldsymbol{\omega}_{t-\tau} - \boldsymbol{\omega}_{t-\tau}^*, g(O_t, \boldsymbol{\omega}_{t-\tau}, \boldsymbol{\theta}_{t-\tau}) - g(\widetilde{O}_t, \boldsymbol{\omega}_{t-\tau}, \boldsymbol{\theta}_{t-\tau})) \rangle] \\
\le{} & 8u^2 l_v d_{TV}(\mathbb{P}(O_t \in \cdot | s_{t-\tau+1}, \boldsymbol{\theta}_{-\tau}), \mathbb{P}(\widetilde{O}_t \in \cdot | s_{t-\tau+1}, \boldsymbol{\theta}_{t-\tau})) \\
\overset{(1)}{\le}{} & 4u^2 l_v l_\pi \sum_{k=t-\tau}^{t} \mathbb{E}\|\boldsymbol{\theta}_k - \boldsymbol{\theta}_{t-\tau}\| \\
\le{} & 16 u^4 l_v l_\pi \tau(\tau+1)\alpha,
\end{aligned}
$$

where (1) follows from Eq. (52).

**Step 4:** show that conditioning on $s_{t-\tau+1}$ and $\boldsymbol{\theta}_{t-\tau}$,

$$\mathbb{E}[\Psi(\widetilde{O}_t, \boldsymbol{\omega}_{t-\tau}, \boldsymbol{\theta}_{t-\tau})] \le 8u^2 l_v \kappa \rho^{\tau-1} \tag{57}$$

From the definition of $\Psi(O, \boldsymbol{\omega}, \boldsymbol{\theta})$, we have

$$\mathbb{E}[\Psi(O'_{t-\tau}, \boldsymbol{\omega}_{t-\tau}, \boldsymbol{\theta}_{t-\tau})|s_{t-\tau+1}, \boldsymbol{\theta}_{t-\tau}] = 0,$$

where $O'_{t-\tau}$ is the tuple generated by $s'_{t-\tau} \sim \mu_{\boldsymbol{\theta}_{t-\tau}}, a'_{t-\tau} \sim \pi_{\boldsymbol{\theta}_{t-\tau}}, s'_{t-\tau+1} \sim \mathcal{P}$. From Assumption 6, we have

$$d_{TV}(\mathbb{P}(\widetilde{s}_t = \cdot|s_{t-\tau+1}, \boldsymbol{\theta}_{t-\tau}), \mu_{\boldsymbol{\theta}_{t-\tau}}) \leq \kappa\rho^{\tau-1}.$$

Then, it holds that

$$
\begin{aligned}
\mathbb{E}[\Psi(\widetilde{O}_t, \boldsymbol{\omega}_{t-\tau}, \boldsymbol{\theta}_{t-\tau})] &= \mathbb{E}[\Psi(\widetilde{O}_t, \boldsymbol{\omega}_{t-\tau}, \boldsymbol{\theta}_{t-\tau}) - \Psi(O'_{t-\tau}, \boldsymbol{\omega}_{t-\tau}, \boldsymbol{\theta}_{t-\tau})] \\
&= \mathbb{E}\langle\boldsymbol{\omega}_{t-\tau} - \boldsymbol{\omega}^*_{t-\tau}, g(\widetilde{O}_t, \boldsymbol{\omega}_{t-\tau}, \boldsymbol{\theta}_{t-\tau}) - g(O'_{t-\tau}, \boldsymbol{\omega}_{t-\tau}, \boldsymbol{\theta}_{t-\tau})\rangle \\
&\leq 8u^2 l_v d_{TV}(\mathbb{P}(\widetilde{O}_t = \cdot|s_{t-\tau+1}, \boldsymbol{\theta}_{t-\tau}), \mu_{\boldsymbol{\theta}_{t-\tau}} \otimes \pi_{\boldsymbol{\theta}_{t-\tau}} \otimes \mathcal{P}) \\
&= 8u^2 l_v d_{TV}(\mathbb{P}((\widetilde{s}_t, \widetilde{a}_t) \in \cdot|s_{t-\tau+1}, \boldsymbol{\theta}_{t-\tau}), \mu_{\boldsymbol{\theta}_{t-\tau}} \otimes \pi_{\boldsymbol{\theta}_{t-\tau}}) \\
&= 8u^2 l_v d_{TV}(\mathbb{P}(\widetilde{s}_t = \cdot|s_{t-\tau+1}, \boldsymbol{\theta}_{t-\tau}), \mu_{\boldsymbol{\theta}_{t-\tau}}) \\
&\leq 8u^2 l_v \kappa\rho^{\tau-1}.
\end{aligned}
$$

Combining Eq. (54), Eq. (55), Eq. (56), and Eq. (57), we have

$$
\begin{aligned}
\mathbb{E}[\Psi(O_t, \boldsymbol{\omega}_t, \boldsymbol{\theta}_t)] &= \mathbb{E}[\Psi(O_t, \boldsymbol{\omega}_t, \boldsymbol{\theta}_t) - \Psi(O_t, \boldsymbol{\omega}_t, \boldsymbol{\theta}_{t-\tau})] + \mathbb{E}[\Psi(O_t, \boldsymbol{\omega}_t, \boldsymbol{\theta}_{t-\tau}) - \Psi(O_t, \boldsymbol{\omega}_{t-\tau}, \boldsymbol{\theta}_{t-\tau})] \\
&\quad + \mathbb{E}[\Psi(O_t, \boldsymbol{\omega}_{t-\tau}, \boldsymbol{\theta}_{t-\tau}) - \Psi(\widetilde{O}_t, \boldsymbol{\omega}_{t-\tau}, \boldsymbol{\theta}_{t-\tau})] + \mathbb{E}[\Psi(\widetilde{O}_t, \boldsymbol{\omega}_{t-\tau}, \boldsymbol{\theta}_{t-\tau})] \\
&\leq c_1\mathbb{E}\|\boldsymbol{\theta}_t - \boldsymbol{\theta}_{t-\tau}\| + c_2\mathbb{E}\|\boldsymbol{\omega}_t - \boldsymbol{\omega}_{t-\tau}\| + 16u^4 l_v l_\pi \tau(\tau+1)\alpha + 8u^2 l_v \kappa\rho^{\tau-1} \\
&\leq 4c_1 u^2 \tau\alpha + 4c_2 u^2 l_v \tau\beta + 16u^4 l_v l_\pi \tau(\tau+1)\alpha + 8u^2 l_v \kappa\rho^{\tau-1}
\end{aligned}
$$

where $c_1 = 4u^2 l_\pi(1 + \lceil\log_\rho \kappa^{-1}\rceil + \frac{1}{1-\rho}) + 2u l_j l_v + 4u l_\omega l_v$ and $c_2 = 2u(8u h_v + 4l_v^2 + 2l_v)$. $\quad\square$

**Proof of Lemma 9.**

*Proof.* We will divide the proof of this lemma into four steps.

**Step 1:** show that for any $O, \boldsymbol{\omega}, \boldsymbol{\theta}_1, \boldsymbol{\theta}_2$, we have

$$\|\Xi(O, \boldsymbol{\omega}, \boldsymbol{\theta}_1) - \Xi(O, \boldsymbol{\omega}, \boldsymbol{\theta}_2)\| \leq c_3\|\boldsymbol{\theta}_1 - \boldsymbol{\theta}_2\|, \tag{58}$$

where $c_3 := 8u^2 l_\omega^2 + 8u^3 h_\omega + 6u l_\omega(2u h_\pi + u l_j + u l_v l_\omega)$.

Since $\Xi(O, \boldsymbol{\omega}, \boldsymbol{\theta}) = \langle\boldsymbol{\omega} - \boldsymbol{\omega}^*, (\nabla\boldsymbol{\omega}^*_{\boldsymbol{\theta}})^\top(\mathbb{E}_{O'}[h(O', \boldsymbol{\theta})] - h(O, \boldsymbol{\theta}))\rangle$, we define $\mathbb{E}_{\boldsymbol{\theta}}[h(O', \boldsymbol{\theta})] := \mathbb{E}_{O'}[h(O', \boldsymbol{\theta})]$, where $\mathbb{E}_{\boldsymbol{\theta}}$ is the shorthand of $\mathbb{E}_{O' \sim (\mu_{\boldsymbol{\theta}}, \pi_{\boldsymbol{\theta}}, \mathcal{P})}$. In the following, we will show that each term in $\Xi(O, \boldsymbol{\omega}, \boldsymbol{\theta})$ is Lipschitz with respect to $\boldsymbol{\theta}$.

Term $\boldsymbol{\omega}$ is not related to $\boldsymbol{\theta}$, term $\boldsymbol{\omega}^* := \boldsymbol{\omega}^*(\boldsymbol{\theta})$ is $l_\omega$-Lipschitz, and term $\nabla\boldsymbol{\omega}^*_{\boldsymbol{\theta}}$ is $h_\omega$-Lipschitz.

For term $h(O, \boldsymbol{\theta})$, denote $\delta(O, \boldsymbol{\theta}) := r(s, a) - J(\boldsymbol{\theta}) + \widehat{V}(\boldsymbol{\omega}^*(\boldsymbol{\theta}); s') - \widehat{V}(\boldsymbol{\omega}^*(\boldsymbol{\theta}); s)$, we have

$$
\begin{aligned}
&\|h(O, \boldsymbol{\theta}_1) - h(O, \boldsymbol{\theta}_2)\| \\
&= \|\delta(O, \boldsymbol{\theta}_1)\nabla\log\pi_{\boldsymbol{\theta}_1}(a|s) - \delta(O, \boldsymbol{\theta}_2)\nabla\log\pi_{\boldsymbol{\theta}_2}(a|s)\| \\
&\leq \|\delta(O, \boldsymbol{\theta}_1)\nabla\log\pi_{\boldsymbol{\theta}_1}(a|s) - \delta(O, \boldsymbol{\theta}_1)\nabla\log\pi_{\boldsymbol{\theta}_2}(a|s)\| \\
&\quad + \|\delta(O, \boldsymbol{\theta}_1)\nabla\log\pi_{\boldsymbol{\theta}_2}(a|s) - \delta(O, \boldsymbol{\theta}_2)\nabla\log\pi_{\boldsymbol{\theta}_2}(a|s)\| \\
&\leq 4u h_\pi\|\boldsymbol{\theta}_1 - \boldsymbol{\theta}_2\| + u|\delta(O, \boldsymbol{\theta}_1) - \delta(O, \boldsymbol{\theta}_2)| \\
&\leq 4u h_\pi\|\boldsymbol{\theta}_1 - \boldsymbol{\theta}_2\| + u(|J(\boldsymbol{\theta}_1) - J(\boldsymbol{\theta}_2)| + \|\widehat{V}(\boldsymbol{\omega}^*(\boldsymbol{\theta}_1); s') - \widehat{V}(\boldsymbol{\omega}^*(\boldsymbol{\theta}_2); s')\| \\
&\quad + \|\widehat{V}(\boldsymbol{\omega}^*(\boldsymbol{\theta}_1); s) - \widehat{V}(\boldsymbol{\omega}^*(\boldsymbol{\theta}_2); s)\|) \\
&\leq (4u h_\pi + 2u l_j)\|\boldsymbol{\theta}_1 - \boldsymbol{\theta}_2\| + 2u l_v\|\boldsymbol{\omega}^*(\boldsymbol{\theta}_1) - \boldsymbol{\omega}^*(\boldsymbol{\theta}_2)\| \\
&\leq l_h\|\boldsymbol{\theta}_1 - \boldsymbol{\theta}_2\|.
\end{aligned}
$$

Hence we have $h(O, \boldsymbol{\theta})$ is $l_h$-Lipschitz, where

$$l_h = 4u h_\pi + 2u l_j + 2u l_v l_\omega. \tag{59}$$

For term $\mathbb{E}_{\boldsymbol{\theta}}[h(O', \boldsymbol{\theta})]$, we have

$$\|\mathbb{E}_{\boldsymbol{\theta}_1}[h(O', \boldsymbol{\theta}_1)] - \mathbb{E}_{\boldsymbol{\theta}_2}[h(O', \boldsymbol{\theta}_2)]\|\|$$

$$\leq \|\mathbb{E}_{\boldsymbol{\theta}_1}[h(O', \boldsymbol{\theta}_1)] - \mathbb{E}_{\boldsymbol{\theta}_1}[h(O', \boldsymbol{\theta}_2)]\|\| + \|\mathbb{E}_{\boldsymbol{\theta}_1}[h(O', \boldsymbol{\theta}_2)] - \mathbb{E}_{\boldsymbol{\theta}_2}[h(O', \boldsymbol{\theta}_2)]\|\|$$

$$\leq \mathbb{E}_{\boldsymbol{\theta}_1}[\|h(O', \boldsymbol{\theta}_1) - h(O', \boldsymbol{\theta}_2)\|\|] + \|\mathbb{E}_{\boldsymbol{\theta}_1}[h(O', \boldsymbol{\theta}_2)] - \mathbb{E}_{\boldsymbol{\theta}_2}[h(O', \boldsymbol{\theta}_2)]\|\|$$

$$\leq l_h \|\boldsymbol{\theta}_1 - \boldsymbol{\theta}_2\| + \|\mathbb{E}_{\boldsymbol{\theta}_1}[h(O', \boldsymbol{\theta}_2)] - \mathbb{E}_{\boldsymbol{\theta}_2}[h(O', \boldsymbol{\theta}_2)]\|\|$$

$$\leq l_h \|\boldsymbol{\theta}_1 - \boldsymbol{\theta}_2\| + 4u^2 d_{TV}(\mu_{\boldsymbol{\theta}_1} \otimes \pi_{\boldsymbol{\theta}_1}, \mu_{\boldsymbol{\theta}_2} \otimes \pi_{\boldsymbol{\theta}_2})$$

$$\leq (l_h + 4u^2 l_\pi (1 + \lceil \log_\rho \kappa^{-1} \rceil + \frac{1}{1-\rho}))\|\boldsymbol{\theta}_1 - \boldsymbol{\theta}_2\|$$

$$\overset{(1)}{\leq} (l_h + 2ul_j)\|\boldsymbol{\theta}_1 - \boldsymbol{\theta}_2\|$$

$$\overset{(2)}{\leq} 2l_h \|\boldsymbol{\theta}_1 - \boldsymbol{\theta}_2\|,$$

where (1) follows from Eq. (19) and (2) comes from the definition of $l_h$ in Eq. (59).

Then we have $\boldsymbol{\omega} - \boldsymbol{\omega}_{\boldsymbol{\theta}}^*$ is $u$-bounded and $l_\omega$-Lipschitz; $\nabla \boldsymbol{\omega}_{\boldsymbol{\theta}}^*$ is $l_\omega$-bounded and $h_\omega$-Lipschitz; $\mathbb{E}_{\boldsymbol{\theta}}[h(O', \boldsymbol{\theta})] - h(O, \boldsymbol{\theta})$ is $8u^2$-bounded and $3l_h$-Lipschitz. By the triangle inequality, we have

$$\|\Xi(O, \boldsymbol{\omega}, \boldsymbol{\theta}_1) - \Xi(O, \boldsymbol{\omega}, \boldsymbol{\theta}_2)\| \leq (8u^2 l_\omega^2 + 8u^3 h_\omega + 3ul_\omega l_h)\|\boldsymbol{\theta}_1 - \boldsymbol{\theta}_2\| \leq c_3 \|\boldsymbol{\theta}_1 - \boldsymbol{\theta}_2\|,$$

where $c_3 := 8u^2 l_\omega^2 + 8u^3 h_\omega + 6ul_\omega(2uh_\pi + ul_j + ul_v l_\omega)$.

**Step 2:** show that

$$\|\Xi(O, \boldsymbol{\omega}_1, \boldsymbol{\theta}) - \Xi(O, \boldsymbol{\omega}_2, \boldsymbol{\theta})\| \leq 4u^2 l_\omega \|\boldsymbol{\omega}_1 - \boldsymbol{\omega}_2\|. \tag{60}$$

Actually, we have

$$\|\Xi(O, \boldsymbol{\omega}_1, \boldsymbol{\theta}) - \Xi(O, \boldsymbol{\omega}_2, \boldsymbol{\theta})\| = \|\langle \boldsymbol{\omega}_1 - \boldsymbol{\omega}_2, (\nabla \boldsymbol{\omega}_{\boldsymbol{\theta}}^*)^\top \mathbb{E}_{O'}[h(O', \boldsymbol{\theta})] - h(O, \boldsymbol{\theta})\rangle\|$$

$$\leq 4u^2 l_\omega \|\boldsymbol{\omega}_1 - \boldsymbol{\omega}_2\|.$$

**Step 3:** show that for tuples $O_t = (s_t, a_t, s_{t+1})$ and $\widetilde{O}_t = (\widetilde{s}_t, \widetilde{a}_t, \widetilde{s}_{t+1})$. Conditioning on $s_{t-\tau+1}$ and $\boldsymbol{\theta}_{t-\tau}$, we have

$$\mathbb{E}[\Xi(O_t, \boldsymbol{\omega}_{t-\tau}, \boldsymbol{\theta}_{t-\tau}) - \Xi(\widetilde{O}_t, \boldsymbol{\omega}_{t-\tau}, \boldsymbol{\theta}_{t-\tau})] \leq 8u^5 l_\omega l_\pi \tau(\tau+1)\alpha. \tag{61}$$

By definition of $\Xi(O, \boldsymbol{\omega}, \boldsymbol{\theta})$, we have

$$\|\mathbb{E}[\Xi(O_t, \boldsymbol{\omega}_{t-\tau}, \boldsymbol{\theta}_{t-\tau}) - \Xi(\widetilde{O}_t, \boldsymbol{\omega}_{t-\tau}, \boldsymbol{\theta}_{t-\tau})]\|\|$$

$$= \|\mathbb{E}[\langle \boldsymbol{\omega}_{t-\tau} - \boldsymbol{\omega}_{t-\tau}^*, (\nabla \boldsymbol{\omega}_{t-\tau}^*)^\top (h(\widetilde{O}_t, \boldsymbol{\theta}_{t-\tau}) - h(O_t, \boldsymbol{\theta}_{t-\tau}))]\|\|$$

$$\leq 4u^3 l_\omega d_{TV}(\mathbb{P}(O_t \in \cdot|s_{t-\tau+1}, \boldsymbol{\theta}_{t-\tau}), \mathbb{P}(\widetilde{O}_t \in \cdot|s_{t-\tau+1}, \boldsymbol{\theta}_{t-\tau})), \tag{62}$$

where the inequality comes from the definition of total variation distance. The total variation norm between $O_t$ and $\widetilde{O}_t$ has been computed in Eq. (52). Plugging Eq. (52) into Eq. (62), we get

$$\|\mathbb{E}[\Xi(O_t, \boldsymbol{\omega}_{t-\tau}, \boldsymbol{\theta}_{t-\tau}) - \Xi(\widetilde{O}_t, \boldsymbol{\omega}_{t-\tau}, \boldsymbol{\theta}_{t-\tau})]\|\| \leq 2u^3 l_\omega l_\pi \sum_{k=t-\tau}^{t} \mathbb{E}\|\boldsymbol{\theta}_k - \boldsymbol{\theta}_{t-\tau}\|$$

$$\leq 8u^5 l_\omega l_\pi \tau(\tau+1)\alpha.$$

**Step 4:** Show that conditioning on $s_{t-\tau+1}$ and $\boldsymbol{\theta}_{t-\tau}$, we have

$$\|\mathbb{E}[\Xi(\widetilde{O}_t, \boldsymbol{\omega}_{t-\tau}, \boldsymbol{\theta}_{t-\tau})]\|\| \leq 8u^3 l_\omega \kappa \rho^{\tau-1}. \tag{63}$$

It can be shown that

$$\|\mathbb{E}[\Xi(\widetilde{O}_t, \boldsymbol{\omega}_{t-\tau}, \boldsymbol{\theta}_{t-\tau})]\|\| \overset{(1)}{=} \|\mathbb{E}[\Xi(\widetilde{O}_t, \boldsymbol{\omega}_{t-\tau}, \boldsymbol{\theta}_{t-\tau}) - \Xi(O'_{t-\tau}, \boldsymbol{\omega}_{t-\tau}, \boldsymbol{\theta}_{t-\tau})]\|\|$$

$$\overset{(2)}{\leq} 8u^3 l_\omega d_{TV}(\mathbb{P}(\widetilde{O}_t \in \cdot|s_{t-\tau+1}, \boldsymbol{\theta}_{t-\tau}), \mu_{\boldsymbol{\theta}_{t-\tau}} \otimes \pi_{\boldsymbol{\theta}_{t-\tau}} \otimes \mathcal{P}),$$

where (1) is due to the fact that $O'_t$ is from the stationary distribution which satisfies $\mathbb{E}[\Xi(O'_{t-\tau}, \omega_{t-\tau}, \theta_{t-\tau})|\theta_{t-\tau}, s_{t-\tau+1}] = 0$ and (2) follows from the definition of total variation distance. From Assumption 6, we know that

$$d_{TV}(\mathbb{P}(\widetilde{s}_t \in \cdot), \mu_{\theta_{t-\tau}}) \leq \kappa \rho^{\tau-1}.$$

Therefore, we have

$$
\begin{aligned}
\|\mathbb{E}[\Xi(\widetilde{O}_t, \omega_{t-\tau}, \theta_{t-\tau})]\| &\leq 8u^3 l_\omega d_{TV}(\mathbb{P}(\widetilde{O}_t = \cdot|s_{t-\tau+1}, \theta_{t-\tau}), \mu_{\theta_{t-\tau}} \otimes \pi_{\theta_{t-\tau}} \otimes \mathcal{P}) \\
&= 8u^3 l_\omega d_{TV}(\mathbb{P}((\widetilde{s}_t, \widetilde{a}_t) \in \cdot|s_{t-\tau+1}, \theta_{t-\tau}), \mu_{\theta_{t-\tau}} \otimes \pi_{\theta_{t-\tau}}) \\
&= 8u^3 l_\omega d_{TV}(\mathbb{P}(\widetilde{s}_t = \cdot|s_{t-\tau+1}, \theta_{t-\tau}), \mu_{\theta_{t-\tau}}) \\
&\leq 8u^3 l_\omega \kappa \rho^{\tau-1}.
\end{aligned}
$$

Combining Eq. (58)-Eq. (63), we can decompose the Markovian bias as

$$
\begin{aligned}
\mathbb{E}[\Xi(O_t, \omega_t, \theta_t)] &= \mathbb{E}[\Xi(O_t, \omega_t, \theta_t) - \Xi(O_t, \omega_t, \theta_{t-\tau})] + \mathbb{E}[\Xi(O_t, \omega_t, \theta_{t-\tau}) - \Xi(O_t, \omega_{t-\tau}, \theta_{t-\tau})] \\
&\quad + \mathbb{E}[\Xi(O_t, \omega_{t-\tau}, \theta_{t-\tau}) - \Xi(\widetilde{O}_t, \omega_{t-\tau}, \theta_{t-\tau})] + \mathbb{E}[\Xi(\widetilde{O}_t, \omega_{t-\tau}, \theta_{t-\tau})] \\
&\leq c_3 \mathbb{E}\|\theta_t - \theta_{t-\tau}\| + 4u^2 l_\omega \mathbb{E}\|\omega_t - \omega_{t-\tau}\| + 8u^5 l_\omega l_\pi \tau(\tau+1)\alpha + 8u^3 l_\omega \kappa \rho^{\tau-1} \\
&\leq 4c_3 u^2 \tau\alpha + 4u^3 l_\omega \tau\beta + 8u^5 l_\omega l_\pi \tau(\tau+1)\alpha + 8u^3 l_\omega \kappa \rho^{\tau-1}.
\end{aligned}
$$

Thus we conclude our proof. $\qquad\square$

**Proof of Lemma 10.**

*Proof.* We will divide the proof of this lemma into three steps.

**Step 1:** show that

$$|\Theta(O, \theta_1) - \Theta(O, \theta_2)| \leq (2uBh_j + 3l_j l_h)\|\theta_1 - \theta_2\|, \tag{64}$$

where $l_h = 4uh_\pi + 2ul_j + 2ul_v l_\omega$ is defined in the proof of Lemma 9.

Since $\Theta(O, \theta) = \langle \nabla J(\theta), \mathbb{E}_{O'_\theta}[h(O'_\theta, \theta)] - h(O, \theta)\rangle$, we will show that each term in $\Theta(O, \theta)$ is Lipschitz.

For the term $\nabla J(\theta)$, we know it's $l_j$-bounded and $h_j$-Lipschitz. For term $\mathbb{E}_\theta[h(O', \theta)] - h(O, \theta)$, we have shown in the proof of Lemma 9 that it's $8u^2$-bounded and $3l_h$-Lipschitz. By the triangle inequality, we have

$$|\Theta(O, \theta_1) - \Theta(O, \theta_2)| \leq (8u^2 h_j + 3l_j l_h)\|\theta_1 - \theta_2\|$$

**Step 2:** show that conditioning on $s_{t-\tau+1}$ and $\theta_{t-\tau}$, we have

$$|\mathbb{E}[\Theta(O_t, \theta_{t-\tau}) - \Theta(\widetilde{O}_t, \theta_{t-\tau})]| \leq 2u^2 l_j l_\pi \sum_{k=t-\tau}^{t} \|\theta_k - \theta_{t-\tau}\| \tag{65}$$

By definition of $\Theta(O, \theta)$, we have

$$
\begin{aligned}
&|\mathbb{E}[\Theta(O_t, \theta_{t-\tau}) - \Theta(\widetilde{O}_t, \theta_{t-\tau})]| \\
&= |\mathbb{E}[\langle \nabla J(\theta_{t-\tau}), h(\widetilde{O}_t, \theta_{t-\tau}) - h(O_t, \theta_{t-\tau})\rangle]| \\
&\leq 4u^2 l_j d_{TV}(\mathbb{P}(O_t \in \cdot|s_{t-\tau+1}, \theta_{t-\tau}), \mathbb{P}(\widetilde{O}_t \in \cdot|s_{t-\tau+1}, \theta_{t-\tau})), \tag{66}
\end{aligned}
$$

where the inequality comes from the definition of total variation distance. The total variation distance between $O_t$ and $\widetilde{O}_t$ has been computed in Eq. (52). Plugging Eq. (52) into Eq. (66), we get

$$|\mathbb{E}[\Theta(O_t, \theta_{t-\tau}) - \Theta(\widetilde{O}_t, \theta_{t-\tau})]| \leq 2u^2 l_j l_\pi \sum_{k=t-\tau}^{t} \|\theta_k - \theta_{t-\tau}\|.$$

**Step 3:** show that conditioning on $s_{t-\tau+1}$ and $\theta_{t-\tau}$, we have

$$|\mathbb{E}[\Theta(\widetilde{O}_t, \theta_{t-\tau}) - \Theta(O'_{t-\tau}, \theta_{t-\tau})]| \leq 4u^2 l_j \kappa \rho^{\tau-1}. \tag{67}$$

From the definition of $\Theta(O, \boldsymbol{\theta})$, we have

$$|\mathbb{E}[\Theta(\widetilde{O}_t, \boldsymbol{\theta}_{t-\tau}) - \Theta(O'_{t-\tau}, \boldsymbol{\theta}_{t-\tau})]| = |\mathbb{E}[\langle \nabla J(\boldsymbol{\theta}_{t-\tau}), h(O'_t, \boldsymbol{\theta}_{t-\tau}) \rangle - \langle \nabla J(\boldsymbol{\theta}_{t-\tau}), h(\widetilde{O}_t, \boldsymbol{\theta}_{t-\tau}) \rangle]|$$

$$\leq 4u^2 l_j d_{TV}(\mathbb{P}(\widetilde{O}_t \in \cdot | s_{t-\tau+1}, \boldsymbol{\theta}_{t-\tau}), \mu_{\boldsymbol{\theta}_{t-\tau}} \otimes \pi_{\boldsymbol{\theta}_{t-\tau}} \otimes \mathcal{P})$$

$$= 4u^2 l_j d_{TV}(\mathbb{P}((\widetilde{s}_t, \widetilde{a}_t) \in \cdot | s_{t-\tau+1}, \boldsymbol{\theta}_{t-\tau}), \mu_{\boldsymbol{\theta}_{t-\tau}} \otimes \pi_{\boldsymbol{\theta}_{t-\tau}})$$

$$= 4u^2 l_j d_{TV}(\mathbb{P}(\widetilde{s}_t = \cdot | s_{t-\tau+1}, \boldsymbol{\theta}_{t-\tau}), \mu_{\boldsymbol{\theta}_{t-\tau}})$$

$$\leq 4u^2 l_j \kappa \rho^{\tau-1},$$

where the last inequality follows from Assumption 6. Therefore, we have

$$|\mathbb{E}[\Theta(\widetilde{O}_t, \boldsymbol{\theta}_{t-\tau}) - \Theta(O'_{t-\tau}, \boldsymbol{\theta}_{t-\tau})]| \leq 4u^2 l_j \kappa \rho^{\tau-1}.$$

Combining Eq. (64), Eq. (65), and Eq. (67), we can decompose the Markovian bias as

$$\mathbb{E}[\Theta(O_t, \boldsymbol{\theta}_t)] = \mathbb{E}[\Theta(O_t, \boldsymbol{\theta}_t) - \Theta(O_t, \boldsymbol{\theta}_{t-\tau})]$$

$$+ \mathbb{E}[\Theta(O_t, \boldsymbol{\theta}_{t-\tau}) - \Theta(\widetilde{O}_t, \boldsymbol{\theta}_{t-\tau})]$$

$$+ \mathbb{E}[\Theta(\widetilde{O}_t, \boldsymbol{\theta}_{t-\tau}) - \Theta(O'_{t-\tau}, \boldsymbol{\theta}_{t-\tau})]$$

$$+ \mathbb{E}[\Theta(O'_{t-\tau}, \boldsymbol{\theta}_{t-\tau})],$$

where $\widetilde{O}_t$ is from the auxiliary Markovian chain defined in Eq. (9) and $O'_{t-\tau}$ is from the stationary distribution which satisfies $\mathbb{E}[\Theta(O'_{t-\tau}, \boldsymbol{\theta}_{t-\tau}) | \boldsymbol{\theta}_{t-\tau}] = 0$.

Then we have

$$\mathbb{E}[\Theta(O_t, \boldsymbol{\theta}_t)] \leq (8u^2 h_j + 3l_j l_h)\mathbb{E}\|\boldsymbol{\theta}_t - \boldsymbol{\theta}_{t-\tau}\| + 2u^2 l_j l_\pi \sum_{k=t-\tau}^{t} \mathbb{E}\|\boldsymbol{\theta}_k - \boldsymbol{\theta}_{t-\tau}\| + 4u^2 l_j \kappa \rho^{\tau-1}$$

$$\leq 4u^2 (8u^2 h_j + 3l_j l_h)\tau\alpha + 8u^4 l_j l_\pi \tau(\tau+1)\alpha + 4u^2 l_j \kappa \rho^{\tau-1}.$$

Therefore, we conclude the proof. □

## H  DECLARATION

I declare that Large Language Models (LLMs) were used solely for language polishing in this paper. No other usage of LLMs was involved.

