# OpenReview forum: "Finite-Time Analysis of Actor-Critic Methods with Deep Neural Network Approximation"
_ICLR.cc/2026/Conference — ICLR 2026 Poster_

### Official Review · Reviewer_qM9u · 2025-10-14

**Soundness:** 3
**Presentation:** 4
**Contribution:** 3
**Rating:** 6
**Confidence:** 5

**Summary:**

This paper established a finite-time convergence result for a proposed DNN-based actor-critic reinforcement learning algorithm.

Specifically, the authors considered a challenging RL setting with 1) continuous state and action spaces, 2) Markovian samplings, and 3) average reward model. To address this problem, they developed a *single-timescale DNN-based* actor-critic algorithm, and, under some assumptions, proved a convergence rate of $\tilde{\mathcal{O}}(T^{-0.5})$. Finally, this paper presented some numerical results to corroborate their theoretical findings.

**Strengths:**

**The general RL setting is impressive.** As most works on RL theory focus on tackling the discount reward model and discrete (even if can be infinite) state and action spaces, this paper extends the results to a more general setup.

**The theoretical analysis is concrete.** Even though some of their techniques and analytical methods are similar to those found in existing research, the authors demonstrate commendable effort in handling a more practical and challenging scenario. I appreciate this theory-driven work, particularly in an era where empirical performance usually overshadows theoretical significance.

In addition, the paper is well-organized and the writing is pretty good.

**Weaknesses:**

My main concern is: **The contributions and insights are not highlighted.** For example, the authors claim that the "single-timescale" design is a primary contribution, yet their analysis lacks a discussion of the associated challenges and the techniques employed to resolve them. Likewise, for continuous spaces, the difficulties introduced by this setup and the authors' solutions remain unexplained.

While these points could represent the main contributions, the authors merely enumerate their findings without offering in-depth discussion or critical analysis.

Besides, I find some claims could be incorrect or inaccurate. (See Questions 2,4.)

**Questions:**

**Please try to answer the following questions:**

1. While the single-timescale method offers practical advantages, its convergence rate is inferior to some two-timescale algorithms [1,2]. This raises a question of potential tradeoff: could the noisier observations inherent in strongly coupled actor-critic methods contribute to this performance disparity?

2. I disagree with the claim regarding $m$-dependence in Lines 70~78. Numerous studies, including [3-4], present $m$-dependent (and depth-dependent) convergence results to emphasize the influence of DNNs. Actually, this paper's findings also connect to the width $m$, though it is implicitly contained within $\epsilon_{app}$. (To some extent, the authors are encouraged to better characterize the $\epsilon_{app}$ with the parameters of the DNNs.) Thus, I think the $m$-dependence is supposed to be a strength rather than a drawback.

3. Can the authors provide the key differences in the analysis among MLP, CNN and ResNet? It could have been interesting, but it is regrettable that the authors do not elaborate on it in sufficient detail.

4. The authors state that a "stationary policy" is optimal due to non-convexity. However, given that numerous studies demonstrate global convergence results [1,2,5,6], should this also be acknowledged by the authors?

5. The numerical experiments are kind of limited in two ways: 1) No other baselines are included; 2) The impact of depth and width remains unclear.

[1] Closing the gap: Achieving global convergence (last iterate) of actor-critic under markovian sampling with neural network parametrization. ICML’24.

[2] Finite-Time Global Optimality Convergence in Deep Neural Actor-Critic Methods for Decentralized Multi-Agent Reinforcement Learning. ICML’25.

[3] Neural Temporal-Difference Learning Converges to Global Optima. NeurIPS’19.

[4] Convergence of Actor-Critic Methods with Multi-Layer Neural Networks. NeurIPS’23.

[5] Sample and Communication-Efficient Decentralized Actor-Critic Algorithms. ICML’22.

[6] Improving sample complexity bounds for (natural) actor-critic algorithms. NeurIPS’20.

---

> ### Author Response · Authors · 2025-11-15
> **Author Response -- Part I**
>
> **(W1 \& Q1: on single-timescale)**
>
> Thanks for your question. In contrast to double-loop and two-timescale AC methods—where the critic is (explicitly or asymptotically) near-converged at each actor update— the single-timescale setting is substantially harder to analyze because the actor and critic are updated with proportional step sizes and thus remain strongly coupled at every iteration. As a result, the standard decoupled error analyses used in prior works no longer apply: the actor error, critic error, DNN approximation error, and Markovian sampling noise evolve jointly and must be controlled simultaneously. This tight coupling forces a fundamentally different error-decomposition strategy from that used in decoupled settings, which is why our analysis introduces several new techniques tailored specifically to the single-timescale regime (see line 86-line 99).
>
> The convergence rate of single-timescale actor–critic is faster than that of two-timescale methods: the former typically achieves a rate of $\tilde{O}(1/\sqrt{T})$, whereas the latter (e.g., Wu et al., 2020) attains only $\tilde{O}(T^{-2/5})$ (see the detailed discussion in the Related Work section of the Appendix). This difference is expected, since two-timescale schemes deliberately slow down the actor updates, which inherently reduces the overall learning speed.
>
> The references [1] and [2] both study double-loop actor–critic algorithms. In [1], the critic is updated in Line 9 (inner loop) while the actor is updated in Line 15 (outer loop). Similarly, [2] updates the critic in Algorithm 2 (inner loop) and the actor in Algorithm 1 (outer loop). Both works therefore incur a sample complexity of $\mathcal{O}(\epsilon^{-3})$, which is worse than our $\tilde{\mathcal{O}}(\epsilon^{-2})$ guarantee. Their seemingly faster convergence in terms of the iteration index $T$ arises from the fact that $T$ counts only the outer-loop iterations; the total sample complexity must additionally multiply by the number of inner-loop critic updates.
>
> In addition, as pointed out by the reviewer, we also note that these double-loop methods establish global optimality. However, they typically requires substantially more samples. Overall, these settings are not directly comparable. In general, the single-timescale actor–critic algorithm is a concise and effective approach with relatively more favorable sample efficiency.
>
> **(Q2: on $m$-dependence)**
>
> Thank you for providing this interesting insight. We agree that characterizing the approximation error $\epsilon_{\rm app}$ in terms of the DNN parameters is a fascinating and important question. However, in a general MDP setting this remains open, as the value function can take diverse forms. In fact, Tian et al. (2024) adopt the same treatment of the approximation error as we do: they use $\epsilon$ to denote the approximation error (Eq. (9) in their paper and directly assume the existence of an $\epsilon$-accurate critic (Assumption 2.5), rather than expressing $\epsilon$ as a function of $m$.
>
> Moreover, a key fact is that the $m$-dependence in their final result arises solely from the critic-error analysis, where a loose bound leaves the smoothness-induced term $\tilde{O}(1/\sqrt{m})$ as a standalone residual, requiring $m\to\infty$ for convergence. In contrast, our refined analysis controls this term by a vanishing critic error multiplier, which removes the need for $m\to\infty$. This is a tighter result and a substantial theoretical improvement.
>
> This refinement also yields an important conceptual advantage. According to NTK theory (Jacot et al., 2018), letting the width $m\to\infty$ pushes the network toward the linearized NTK regime. In Tian et al. (2024), the presence of the $m$-dependent error term forces $m$ to be extremely large in order to suppress $\tilde{O}(1/\sqrt{m})$, which inevitably drives the network closer to linearity and weakens its approximation capacity—thereby increasing $\epsilon_{\rm app}$. By avoiding this $m\to\infty$ requirement, our analysis provides both theoretical and practical benefits.

---

> > ### Author Response · Authors · 2025-11-15
> > **Author Response -- Part II**
> >
> > **(Q3: on analysis among MLP, CNN, and ResNet)**
> >
> > Thank you for the thoughtful question. Our analysis focuses on the fundamental learning dynamics of single-timescale actor–critic methods under a general nonlinear neural network approximator. In the theoretical development, the critic is modeled as a generic neural network (Eq. (5)), and the proofs rely only on high-level properties such as smoothness, bounded gradients, and approximation capability. These assumptions are broad enough to cover MLPs, CNNs, and ResNets without requiring architecture-specific arguments.
> >
> > A detailed comparison among MLPs, CNNs, and ResNets would require architectural assumptions that go beyond the scope of our finite-time analysis. For example:
> >
> > - MLPs rely on universal approximation but do not impose structural constraints (e.g., locality).
> >
> > - CNNs introduce weight sharing and locality, which can improve sample efficiency in structured input domains, but the core convergence arguments in our analysis do not hinge on such structure.
> >
> > - ResNets incorporate skip connections that help optimization stability, yet their theoretical treatment typically requires separate assumptions (e.g., near-identity initialization).
> >
> > In our framework, these architectural differences affect the approximation error term and the Lipschitz constants, but do not alter the main convergence rate or proof structure.
> >
> >
> > **(Q4: on global convergence results)**
> >
> > Thank you for pointing out these references. We truly appreciate and acknowledge these contributions, and we have reviewed and incorporated a discussion of these results in the Related Work section of our updated manuscript.
> >
> > **(Q5: on experiments)**
> >
> > Thank you for your comment. We would like to note that the linear critic baseline was included in the original submission to highlight the limitations of linear approximation and motivate the study of neural network critics. The effects of network width and depth were also reported in Table 2 of the initial submission.

---

> ### Comment · Reviewer_qM9u · 2025-11-23
>
> Thanks for your responses.
>
> I realize that the distinction between two-timescale and single-timescale methods may be greater than I initially thought.
>
> For the numerical experiments, my primary concerns are: 1) the authors did not include any DNN-based baselines, and 2) while the width and depth of the DNNs are reported in the experiments, their theoretical bounds remain unclear.
>
> I've decided to keep my current score.

---

> > ### Author Response · Authors · 2025-11-24
> >
> > Thank you for your response and for maintaining your positive score. We sincerely appreciate your constructive feedback, which has been instrumental in guiding our revisions.
> >
> > Regarding the numerical experiments, if you have any specific DNN baseline you would like to see a comparison, please feel free to let us know.
> >
> > For the theoretical guarantees, we would like to clarify that our bound is a general upper bound that accommodates a wide range of network widths and depths, provided that the approximation error $\epsilon_{\rm app}$ does not dominate the learning error.
> >
> > Thank you again for your time and thoughtful comments. We would be very happy to further discuss any additional concerns you may have.

---

### Official Review · Reviewer_8pXP · 2025-10-29

**Soundness:** 3
**Presentation:** 2
**Contribution:** 1
**Rating:** 4
**Confidence:** 4

**Summary:**

This work proposes an actor–critic algorithm with finite-time analysis under a neural network function approximation setting. Compared with previous studies, the paper establishes a sample convergence rate for environments with continuous action spaces. In addition, the authors present simulation results to demonstrate the effectiveness of the proposed approach.

**Strengths:**

The study derives theoretical guarantees by establishing the sample convergence rate of the Actor-Critic algorithm for MDP with continuous action spaces.

In addition, the authors include simulation experiments that demonstrate the empirical validity of the theoretical results.

**Weaknesses:**

1. For Table 1, I am confused whether the comparison is fair, especially for the (Tian et.al. 2024).
(a) Firstly, the sampling process is Markovian, both in Actor and Critic part. The restart setting is just to achieve different distribution for policy gradient under the discount finite horizon setting.  Please carefully check this part and correct the table.
(b) Besides, the width of the Neural Network is for the $\epsilon$-order approximation error of the value function. Your work choose to avoid this width but will lead to approximation error.   Finally, previous works with neural approximation will converge to $ \mathcal{O}(\epsilon)$ accurate set but your work will converge to $ \mathcal{O}(\epsilon+\epsilon_{approx})$.

2. Compared with (Tian et.al. 2024) and other previous works, could the author detailed  explain where are the techique novelty or improvement. I went through the proof sketch but analysis looks like standard.

**Questions:**

See weaknesses, please.

---

> ### Author Response · Authors · 2025-11-15
> **Author Response -- Part I**
>
> **(W1(a): on sampling scheme)**
>
> Thank you for seeking clarification. In our work, we use the term Markovian sampling to refer to the setting where all samples are drawn from a Markov chain. Concretely, the samples follow
>
> \begin{align*}
> (s_0,a_0) \xrightarrow{(\mathcal{P},\pi_{\theta_1})} (s_1,a_1)
> \xrightarrow{(\mathcal{P},\pi_{\theta_2})} (s_2,a_2)
> \cdots
> \xrightarrow{(\mathcal{P},\pi_{\theta_t})} (s_t,a_t),
> \end{align*}
>
> forming one trajectory $(s_0,a_0,s_1,a_1,\dots,s_t,a_t)$.
> This is the standard and is also referred to as Markovian sampling in prior work, such as Wu et al. (2020) and
> Chen et al. (2023). Using such a sampling scheme induces Markovian noise that must be handled.
>
> In contrast, Tian et al. (2024) employ a fundamentally different sampling
> scheme. For each update at timestep $t$, the state-action pair
> $(\hat{s}_t,\hat{a}_t)$ used in the actor update is obtained by sampling a
> random horizon
> \begin{align*}
> T \sim {\rm Geom}(1-\gamma),
> \end{align*}
> rolling out a trajectory $(s_0,a_0,s_1,a_1,\dots,s_T,a_T)$, and using only
> the terminal pair $(\hat{s}_t,\hat{a}_t) := (s_T,a_T)$. Each $(\hat{s}_t,\hat{a}_t)$ therefore arises from an independent rollout, and successive samples do not satisfy any Markovian dependency:
> \begin{align*}
> (\hat{s}_t,\hat{a}_t) \not\rightarrow (\hat{s}\_{t+1},\hat{a}\_{t+1})
> \end{align*}
>
> Because the samples used for the actor update are drawn from independent random-horizon rollouts rather than from a Markov chain, no Markovian noise arises in this part. In practice, this is also less sample-efficient
> than single-trajectory Markovian sampling.
>
> For this reason, although Tian et al. (2024) refer to their scheme as
> “Markovian sampling”, we view it as fundamentally different from the standard
> usage of the term and label it as “not Markovian” in Table 1. Notably, this same sampling mechanism has been used in [1], who explicitly refer to it as random-horizon policy gradient. Following this terminology, we believe “random-horizon sampling” is a more accurate description for this type of sampling.
>
> Thank you for allowing us to clarify this point. We have added the above definition of Markovian sampling in the Appendix (additional notations) to illustrate why the actor sampling in Tian et al. (2024) is not Markovian.
>
> [1] Kaiqing Zhang, Alec Koppel, Hao Zhu, and Tamer Basar. Global convergence of policy gradient
> methods to (almost) locally optimal policies. SIAM Journal on Control and Optimization,
> 58(6):3586–3612, 2020.

---

> ### Author Response · Authors · 2025-11-15
> **Author Response -- Part II**
>
> **(W1(b): on approximation error)**
>
> Thank you for seeking clarification. In Tian et al. (2024), the approximation
> error of the neural-network value function is denoted by $\epsilon$ in Eq. (9), and the existence
> of an $\epsilon$-accurate critic is explicitly assumed in Assumption 2.5.
> Accordingly, their $\mathcal{O}(\epsilon)$ represents the same approximation error
> as our $\mathcal{O}(\epsilon_{\rm app})$, with the only difference being that we use a
> subscript for clarity. Hence, our analysis does not introduce any additional approximation
> error beyond what is already assumed in Tian et al. (2024). This approximation error is inherent to using a function approximator for the critic and is present in all prior works summarized in Table 1.
>
> For clarity, we summarize the role of each error term below.
>
> \begin{aligned}
> \text{Tian et al.\ (2024):}\quad
> \Delta_Q
> &\le
> \underbrace{O\\left(\frac{\log^2 T}{\sqrt{T}}\right)}\_{\text{finite-sample statistical error}}+\underbrace{O(\epsilon)\vphantom{\frac{\log^2 T}{\sqrt{T}}}}\_{\text{approximation error}}+
> \underbrace{\tilde{O}\\!\left(\tfrac{1}{\sqrt{m}}\right)\vphantom{\frac{\log^2 T}{\sqrt{T}}}}\_{\textcolor{red}{\text{additional error}}}
> \\\\
> \text{Ours:}\quad
> \mathcal{E}^{(\nabla)}
> &\le
> \underbrace{O\\!\left(\frac{\log^2 T}{\sqrt{T}}\right)}\_{\text{finite-sample statistical error}}
> +
> \underbrace{O(\epsilon_{\mathrm{app}})\vphantom{\frac{\log^2 T}{\sqrt{T}}}}\_{\text{approximation error}}
> \end{aligned}
>
> Moreover, the statement that ``the width of the neural network is for the
> $\epsilon$-order approximation error of the value function'' is not precise.
> The additional term $\tilde{O}(1/\sqrt{m})$ in Tian et al. (2024) does not arise
> from the $\epsilon$-approximation assumption. As discussed earlier, their analysis
> already assumes an $\epsilon$-accurate critic. The $\tilde{O}(1/\sqrt{m})$
> term instead comes from the technical treatment of the critic error in their proof.
> This term is not an inherent error and can be removed by a tighter analysis,
> which is precisely what our work achieves.
>
> This refinement also leads to an important conceptual advantage. According to NTK
> theory (Jacot et al., 2018), taking the width $m \to \infty$ pushes the network
> toward the linearized NTK regime. In Tian et al. (2024), the presence of the
> additional error term forces $m$ to be extremely large ($m\to\infty$) in order to suppress
> $\tilde{O}(1/\sqrt{m})$. However, making $m$ extremely large inevitably pushes the network toward linear behavior and degrades its approximation capacity, thereby leading to a larger $\epsilon_{\rm app}$ value.
>
> In contrast, our analysis does not require $m \to \infty$, allowing the width to
> remain finite and flexible. This preserves the nonlinear expressive power of the
> critic and potentially leads to better approximation capability.
>
> **(W2: on technique novelty)**
>
> Thank you for the question. Below we highlight the three main technical novelties that distinguish our analysis from Tian et al. (2024) and prior works.
>
>
> (1) A refined critic-error analysis that removes the requirement $m \to \infty$. The additional error term in Tian et al. (2024) arises solely from their
> critic-error analysis, where a loose bound causes the smoothness-induced term
> $\tilde{O}(1/\sqrt{m})$ to remain as a standalone residual, thereby forcing
> $m \to \infty$ for convergence. In contrast, our refined analysis controls this term by a vanishing critic error multiplier, allowing us to avoid the infinite-width requirement. This is a tighter result and a substantial theoretical improvement.
>
> (2) An operator-based analytical framework for continuous state-action spaces. To handle uncountable domains, we introduce an operator formulation (Eq. (1)) that enables precise control of error propagation. This framework is absent from prior works. Moreover, we establish several additional intermediate lemmas—such as Lemmas 2-4—to support the analysis in continuous state–action spaces.
>
> (3) A new treatment of the interaction between DNN approximation error and Markovian sampling noise under continuous spaces. Unlike linear settings or i.i.d. sampling analyses, our work develops refined bounds (Lemmas 7-10) to control the coupled dynamics between approximation error and Markovian noise—an interaction that is analytically more challenging.

---

> > ### Author Response · Authors · 2025-11-26
> >
> > Dear Reviewer 8pXp,
> >
> > Thank you again for your thoughtful feedback, which has been very helpful in improving our work. As the discussion period is coming to a close, we would greatly appreciate it if you could take a moment to look over our rebuttal at your convenience. Please feel free to let us know if any points require further clarification—we are more than happy to elaborate.
> > Thank you once again for your time and effort!
> >
> > Best,
> > Authors

---

> > ### Comment · Reviewer_8pXP · 2025-11-28
> >
> > Thanks for the authors’ response. My major concern has been addressed, and I will increase my rating. However, I recommend that the authors add some notes or subscript annotations to clearly distinguish this difference, since the work by Tian et al. (2024) claims that they follow Markov sampling.

---

> > > ### Author Response · Authors · 2025-11-28
> > >
> > > Thank you for your response and for increasing the score. We are pleased that our clarification has addressed your concern. For clarity, we have added a remark in the Appendix that explains Markovian sampling and clarifies how our setting differs from that of Tian et al. (2024) in the updated manuscript.
> > >
> > > Thank you again for your time and thoughtful feedback. We would be very happy to further discuss any additional concerns you may have.

---

### Official Review · Reviewer_Duep · 2025-10-30

**Soundness:** 3
**Presentation:** 3
**Contribution:** 2
**Rating:** 6
**Confidence:** 4

**Summary:**

This paper provides the first finite-time convergence analysis for single-timescale actor-critic (AC) algorithms utilizing deep neural network approximation in continuous state-action spaces under the time-average reward setting. The authors prove convergence to a stationary point at a rate of  $\tilde{O}({T}^{-1/2})$ for the coupled reward, critic, and actor errors. The theoretical claims are substantiated with experiments on the Pendulum task and MuJoCo benchmarks, demonstrating the superior approximation capability of neural critics over linear ones and empirically validating the predicted convergence rate.

**Strengths:**

1. This paper  provides a finite-time analysis for the challenging single-timescale neural AC setting, with continuous spaces and Markovian sampling, is a substantial theoretical advance.
2. The paper goes beyond pure theory by including comprehensive experiments. The empirical confirmation of the $\tilde{O}({T}^{-1/2})$ convergence rate on Pendulum and the demonstration of strong performance on MuJoCo benchmarks provide crucial support for the theoretical results and highlight the practical relevance of the analyzed algorithm.
3. The paper is well-structured and easy to follow.

**Weaknesses:**

The analysis operates in the neural tangent kernel (NTK) or overparameterized regime, where the network is wide enough to be well-approximated by its linearization around initialization. This regime, while theoretically fruitful, does not fully capture the feature learning dynamics that are believed to be crucial for the success of deep learning in practice.

**Questions:**

1. How is $m$ avoided in the convergence result? Does it depend on the assumption that the network is wide enough?
2. What is the convergence rate guarantee for the discounted reward setting, as it is more common in RL formulation?

---

> ### Author Response · Authors · 2025-11-15
> **Author Response**
>
> **(W1 \& Q1: on neural network width)**
>
> Thanks for the insightful question. The $m$ is avoided in the convergence result via a more accurate and less conservative analysis proposed in our paper. Specifically, we note that the
> term involving $m$ arises from the smoothness-induced error
> $h_v = \tilde{O}(1/\sqrt{m})$ in Lemma 1, which appears in the critic-error analysis. In our mean-path analysis (the term $I_1$ in Eq. (25)), a key observation is that the
> smoothness-induced error $h_v$ does not stand alone; instead, it is multiplied by the critic error $\mathbb{E}\||\mathbf{z}_t\||$ (see the first term in Eq. (27) of the
> updated manuscript). Since $\mathbb{E}\||\mathbf{z}_t\||$ vanishes as the iteration
> progresses, controlling this interaction only requires $h_v$ to be a constant
> satisfying Eq. (28), and the decay of the product is then governed by the decay of
> $\mathbb{E}\||\mathbf{z}_t\||$.
>
> In contrast, the prior analysis in Tian et al. (2024) applies a loose bound that detaches the smoothness-induced error $h_v=\tilde{O}(1/\sqrt{m})$ from the critic error. As a result of this relaxation, $h_v$ appears as
> a standalone residual term rather than being coupled with a vanishing quantity.
> Consequently, the $\tilde{O}(1/\sqrt{m})$ term persists in their final convergence
> result, and suppressing it requires taking $m\to\infty$.
>
>
> In short, our analysis leverages the fact that $\mathbb{E}\||\mathbf{z}_t\||$ vanishes,
> rather than bounding it loosely by a constant, which allows us to eliminate the
> dependence on $m$ in the convergence guarantee.
>
> Therefore, the only requirement on $m$ in our analysis is to satisfy Eq. (28),
> which merely imposes that $m$ be larger than a problem-dependent constant. Since this threshold depends on the underlying MDP, we cannot characterize the network as ‘wide enough’ or ‘not too wide,’ as the required width is a problem-dependent constant. In our experiments, we demonstrate that choosing a moderate width  (e.g., $m \approx 100$) already performs well.
>
> **(Q2: on discounted reward setting)**
>
> Thank you for your inspiring question. We expect that the convergence rate in the discounted-reward setting would also be $\tilde{O}(1/\sqrt{T})$, as many components of the learning dynamics transfer directly and do not introduce convergence-rate differences. That said, establishing a formal guarantee for the discounted case requires additional technical developments to handle the specific distinctions of that setting. We view this as an interesting direction for future work.

---

### Official Review · Reviewer_VBtF · 2025-11-01

**Soundness:** 3
**Presentation:** 3
**Contribution:** 3
**Rating:** 6
**Confidence:** 3

**Summary:**

The paper studies single-timescale actor--critic (AC) with deep neural network function approximation in continuous state--action spaces under the average-reward objective. It analyzes a practical AC loop that jointly updates a reward estimator, a TD(0)-style critic, and a policy-gradient actor with Markovian on-policy samples. The main theoretical result is a finite-time convergence guarantee to a stationary point at a $\tilde{O}(T^{-1/2})$ rate (up to logarithmic factors), simultaneously controlling reward-estimation, critic, and actor errors. Experiments (Pendulum, MuJoCo-style tasks) illustrate empirical trends, including a measured slope near $-1/2$ on log--log plots and improvements from neural critics over linear baselines.

**Strengths:**

1. Realistic setting: single-timescale updates with Markovian sampling in continuous spaces---closer to practice than idealized double-loop or two-timescale analyses.

2.  Finite-time $\tilde{O}(T^{-1/2})$ rate: matches the best known dependence on $T$ (up to logs) for this setting; jointly tracks three coupled sources of error.

3. Empirical checks: (i) Pendulum results where the neural critic better aligns with an RVI baseline than a linear/RBF critic; (ii) empirical slope $\approx -0.51$ consistent with theory; (iii) MuJoCo ablations show depth/width benefits over linear critics.

4. Assumptions documented and motivated: geometric mixing/ergodicity, an exploration inequality, and smoothness/Lipschitz properties for policy and dynamics, with discussion of when exploration can fail.

**Weaknesses:**

1.  The theory assumes sufficiently wide networks and projects critic updates to remain near initialization. It is unclear how necessary/tight this is or how it maps to common unconstrained training with Adam/weight decay.

2. Guarantees include an $O(\varepsilon_{\text{app}})$ term from critic approximation, but there is limited guidance for architectures/regularization that make $\varepsilon_{\text{app}}$ small in practice; experiments do not quantify this floor or test misspecification.

**Questions:**

Is projection onto a radius constraint essential for the analysis, or could similar guarantees hold for unconstrained (Adam/SGD) updates with weight decay?

---

> ### Author Response · Authors · 2025-11-15
> **Author Response**
>
> **(W1 \& Q1: on projection)**
>
> Thank you for the question. The projection onto a radius-constrained ball is required for the theoretical analysis, as it ensures that the critic iterates remain bounded. We also note that, for sufficiently wide neural networks, the optimal critic typically lies in a neighborhood of the initialization and therefore naturally falls within a constant-radius ball. Importantly, our analysis does not rely on the iterates staying close to the initialization; we only require that the optimal solution lies within some finite-radius ball centered at the initialization, which is a much milder assumption and has been widely adopted in all prior works. Empirically, we observe that the critic parameters stay within a moderate radius throughout training, consistent with the theoretical assumption.
>
> In practice, unconstrained training with Adam or weight decay already enforces an implicit form of radius control: both the adaptive learning-rate normalization in Adam and the shrinkage effect of weight decay prevent the parameters from diverging. Thus, the projection in our analysis can be viewed as a theoretical analogue of the implicit parameter-norm control that commonly emerges in practice with Adam or weight decay.
>
> **(W2: on guidance for approximation error)**
>
> Thank you for your insightful comment. The $O(\epsilon_{\rm app})$ term arises from the critic’s approximation error, which is standard in finite-time analyses of actor-critic methods. Although a full characterization of approximation error versus architecture is still an open problem in actor-critic theory and is beyond the scope of most finite-time theoretical works, in practice, this error can be effectively controlled. Using a sufficiently expressive function class (e.g., 2–3 layer MLPs with width scaling with the state dimension), together with standard regularization and stability techniques such as weight decay, target networks, or gradient clipping, helps keep the approximation floor small. That said, this term naturally depends on many factors, and its sensitivity can vary considerably across tasks. Consequently, the precise architectural and regularization choices are task-dependent, which is consistent with common practice in RL implementations.
>
> Accordingly, our analysis characterizes the algorithm’s sample complexity under the standard assumption that the critic attains a reasonable approximation, in line with prior finite-time actor–critic results. Our empirical evaluations employ established architectures and training protocols. While we do not directly measure the approximation error floor, Fig. 1(a) provides clear qualitative evidence that the neural critic achieves a substantially smaller $\epsilon_{\rm app}$ than the linear critic in approximating the optimal value function.

---

### Meta-Review · Area_Chair_ubm2 · 2026-01-07

**Summary:**

The paper provides a finite-time analysis of single-timescale actor-critic algorithms using deep neural networks (DNN) in continuous state-action spaces. Reviewers generally agreed that the setting is more realistic than that of previous works and bridges a gap between theory and practice. Key concerns centered on the technical novelty compared to prior work (e.g., Tian et al., 2024), the dependence on neural network width ($m$), and the fairness of the sampling scheme definitions. The authors’ rebuttal successfully clarified that their analysis removes the $m \to \infty$ requirement found in previous works, yielding a tighter $O(T^{-1/2})$ convergence rate without pushing the model into the linearized NTK regime.

**Reviewer Concerns:**

Reviewers 8pXP and qM9u raised concerns about the technical advancements compared to Tian et al. (2024). The authors demonstrated that their refined critic-error analysis eliminates the standalone $1/\sqrt{m}$ residual error, enabling the use of finite-width networks. However, based on my understanding, it’s not entirely clear how m depends on T. In particular, when analyzing NTK-based neural network function approximation, m is typically required to be sufficiently large to accommodate the overparameterization scheme (see for example A Finite-Time Analysis of Q-Learning with Neural Network Function Approximation, ICML 2020). While m doesn’t necessarily need to approach infinity, it’s common and intuitive to set m as a function of T because the approximation error of NTK accumulates over time. The authors should thoroughly address this aspect in their final version.

Reviewer 8pXP requested clarification on “Markovian sampling.” The authors explained the distinction between their single-trajectory approach and the random-horizon rollouts employed in previous research, which led the reviewer to increase their score.

**Reviewer Scores:**

Reviewer VBtF: 6 (Marginally above threshold; appreciated the realistic single-timescale setting).

Reviewer Duep: 6 (Marginally above threshold; noted the substantial theoretical advance for continuous spaces).

Reviewer 8pXP: 4 (Mentioned they will raise score in the comment section; major concerns regarding sampling and width were addressed in the rebuttal).

Reviewer qM9u: 6 (Maintained positive score; acknowledged the difficulty of single-timescale analysis but desired more DNN baselines).

---

### Decision · Program_Chairs · 2026-01-26

Accept (Poster)